# DECOUPLED Q-CHUNKING

**Qiyang Li**
UC Berkeley
qcli@berkeley.edu

**Seohong Park**
UC Berkeley
seohong@berkeley.edu

**Sergey Levine**
UC Berkeley
svlevine@berkeley.edu

## ABSTRACT

Temporal-difference (TD) methods learn state and action values efficiently by bootstrapping from their own future value predictions, but such a self-bootstrapping mechanism is prone to *bootstrapping bias*, where the errors in the value targets accumulate across steps and result in biased value estimates. Recent work has proposed to use chunked critics, which estimate the value of short action sequences ("chunks") rather than individual actions, speeding up value backup. However, extracting policies from chunked critics is challenging: policies must output the entire action chunk open-loop, which can be sub-optimal for environments that require policy reactivity and also challenging to model especially when the chunk length grows. Our key insight is to decouple the chunk length of the critic from that of the policy, allowing the policy to operate over shorter action chunks. We propose a novel algorithm that achieves this by optimizing the policy against a distilled critic for partial action chunks, constructed by optimistically backing up from the original chunked critic to approximate the maximum value achievable when a partial action chunk is extended to a complete one. This design retains the benefits of multi-step value propagation while sidestepping both the open-loop sub-optimality and the difficulty of learning action chunking policies for long action chunks. We evaluate our method on challenging, long-horizon offline goal-conditioned tasks and show that it reliably outperforms prior methods.
**Code:** github.com/ColinQiyangLi/dqc.

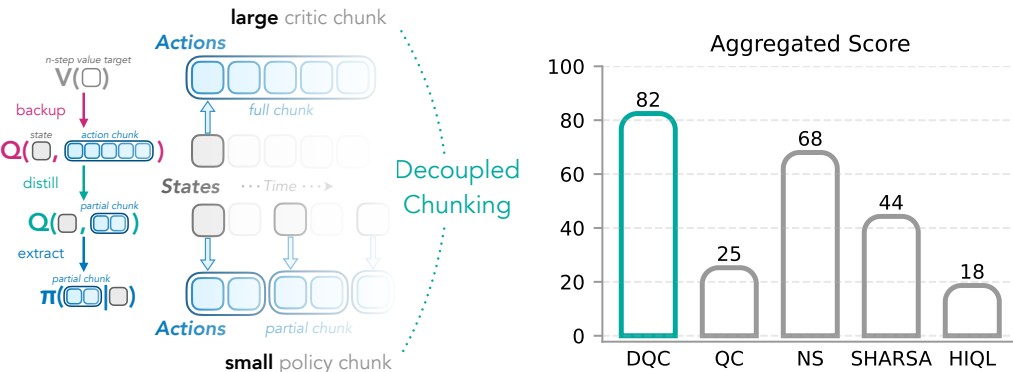

Figure 1: **Decoupled Q-chunking (DQC).** *Left:* The key idea of our method is to 'decouple' the action chunk size of the critic $Q$ from that of the policy $\pi$. A large critic chunk size allows for efficient value learning while a small policy chunk size makes policy learning more tractable and allows for better policy reactivity. *Right:* Our method outperforms all baselines on six hardest environments on OGBench, an offline goal-conditioned RL benchmark with challenging long-horizon tasks.

## 1 INTRODUCTION

Temporal-difference (TD) methods are powerful reinforcement learning (RL) techniques that can directly learn from off-policy prior data without requiring an explicit dynamics model, making them well-suited for offline RL (Levine et al., 2020) and sample-efficient online RL (Chen et al., 2021). Despite their successes, a key challenge remains: bootstrapping bias (Jaakkola et al., 1993; Sutton

et al., 1998; De Asis et al., 2018; Park et al., 2025b). This bias stems from the core design of TD updates, where the value at the current state is learned by regressing towards the learner's own predictions at the next time step. As a result, any prediction error is compounded backward across steps, making learning particularly challenging in long-horizon, sparse-reward tasks.

Multi-step return backups (Sutton et al., 1998) can alleviate bootstrapping bias by shifting the regression target further into the future and effectively reducing the time horizon. However, naïvely applying them introduces additional biases because computing the target involves summing rewards along off-policy trajectories that may deviate from the actions that the agent would take. Although importance sampling can in principle correct such off-policy biases by reweighting the off-policy trajectories (Munos et al., 2016), it often suffers from high variance and thus requires truncation and other heuristics for numerical stability, making it difficult to tune in practice. Recent works (Seo & Abbeel, 2025; Li et al., 2025a; Tian et al., 2025; Li et al., 2025b) leverage chunked value functions, which estimate the value of short action sequences ("chunks") rather than a single action. This formulation allows $n$-step return backup without the pessimistic bias (under some condition we formalize in Section 4). However, theoretical guarantees of action chunking Q-learning, especially on arbitrary off-policy data, are still an open problem as existing analysis (*e.g.*, in Li et al. (2025b)) only considers the case where the data is collected by an action chunking policy. Moreover, on the empirical side, directly optimizing a policy over full action chunks is difficult, particularly as the chunk size grows, and it is still unclear how to best extract a policy from a chunked critic.

In this work, we lay the theoretical foundation of action chunking Q-learning where we identify the key *open-loop consistency* condition (Definition 2) under which Q-learning with action chunking critic is guaranteed to produce a near-optimal action chunking policy. On top of it, we characterize the condition when closed-loop execution (*i.e.*, only executing the first action in the predicted action chunk) of such action chunking policy is expected to be even close to the optimal closed-loop policy. Motivated by our analysis, we develop a simple practical algorithm that builds on top of the idea of closed-loop execution of action chunking policies to address the action chunking policy learning challenge. The key insight is that we can avoid training the policy to predict the full action chunks and instead to only predict shorter, partial action chunks against the chunked critic. To achieve this, we use a 'distilled' chunked critic with a chunk size that matches the policy: it optimistically regresses to the original chunked critic to approximate the maximum value that the partial action chunk can achieve after being extended into a full action chunk. Conceptually, while the action optimization is still done for the longer, complete action chunks, the policy network is only trained to output the partial action chunk of an optimized complete action chunk. This way, the policy only needs to predict a much shorter action chunk (*e.g.*, in the extreme case, only one action), which often admits a much simpler distribution, while enjoying the value learning benefits from the use of chunked critics.

Our main contributions are two-fold. On the theoretical side, we provide the *first* formal analysis of Q-learning with action chunking, focusing on characterizing the value learning bias of the bellman backup of action chunking critic. Specifically, we introduce the *open-loop consistency* condition under which we *exactly* characterize the worst-case value estimation bias (Theorems 1 and 4) and sub-optimality gap (Theorems 2 and 5) at the fixed point of the bellman optimality equations. Moreover, we characterize *(i)* the conditions under which action chunking critic backup is preferable over $n$-step return backup with a single-step critic (Proposition 3), and *(ii)* the conditions under which closed-loop execution of the action chunking policy further mitigates the open-loop bias (Theorem 3). On the empirical side, we propose a new technique, **Decoupled Q-chunking (DQC)**, that addresses the policy learning challenge in action chunking Q-learning by decoupling the policy chunk size from the critic chunk size. DQC trains a policy to only predict a partial action chunk, significantly reducing the policy learning challenge, while retaining the value learning benefits of the chunked critic. We instantiate this technique as a practical offline RL algorithm that outperforms the previous state-of-the-art method on the hardest set of environments in OGBench (Park et al., 2025a), a challenging, long-horizon goal-conditioned RL benchmark.

## 2 RELATED WORK

**Offline and offline-to-online reinforcement learning** methods assume access to an offline dataset to learn a policy without interactions with the environment (offline) (Kumar et al., 2020; Kostrikov et al., 2022; Tarasov et al., 2024) or with as little online interaction with the environment as possible (offline-to-online) (Lee et al., 2022; Ball et al., 2023; Nakamoto et al., 2024). Q-learning or TD-based RL

algorithms have been a popular choice for these problem settings as they naturally handle off-policy data without the need for on-policy rollouts, and also exhibit great online sample-efficiency (Chen et al., 2021; D'Oro et al., 2023). A large body of literature in these two problem settings has been focusing on tackling the distribution shift challenge by appropriately constraining the policies with respect to the prior offline data, and most of them use the standard 1-step TD backup for Q-learning, which has been known to suffer from the bootstrapping bias problem in the RL literature (Jaakkola et al., 1993; Sutton et al., 1998). To tackle this, recent work (Jeong et al., 2023; Park & Lee, 2025; Park et al., 2025b; Li et al., 2025b) has shown that multi-step return backups are effective for improving offline/offline-to-online Q-learning agents. These methods either use a standard single-step critic network (Park et al., 2025b) that suffers from the off-policy bias, or use a 'chunked,' multi-step critic network (Li et al., 2025b) that does not have such bias but poses a huge policy learning challenge when the chunk size is too large. Our method brings the best of both worlds—it uses critic chunking to avoid the off-policy bias while simultaneously avoiding the policy learning challenge by extracting a simpler policy that extracts a shorter action chunk from the full-chunk critic.

**Multi-step return backups** are computed with multi-step off-policy rewards that can lead to systematic value underestimation (Sutton et al., 1998; Peng & Williams, 1994; Konidaris et al., 2011; Thomas et al., 2015), and there has been a rich literature (Precup et al., 2000; Munos et al., 2016; Rowland et al., 2020) dedicated to fix these biases via importance sampling (Kloek & Van Dijk, 1978) with truncation (Ionides, 2008). These approaches often require a careful balance between bias and variance that can be tricky to tune. More recently, Seo & Abbeel (2025); Li et al. (2025a); Tian et al. (2025); Li et al. (2025b) group temporally extended sequences of actions as chunks and directly estimate the value of an action chunk rather than a single action. Such a formulation allows the value backup to operate directly in the chunk space, which allows multi-step return backup without the systematic biases from the sub-optimal off-policy data. Despite their empirical success, we still lack a good theoretical understanding of the convergence of TD-learning with 'chunked' critics, as well as when it should be preferred over the standard $n$-step returns. Our work lays out the theoretical foundation for Q-learning with critic chunking, and identifies an important yet subtle, often overlooked bias in the chunked TD-backup. We quantify such bias and provide the condition under which TD backup using critic chunking is guaranteed to perform better than the standard $n$-step return backup with a single-step critic.

See additional discussions of related work on hierarchical reinforcement learning and theoretical analysis for action chunking and confounding variables in Section E.

## 3 PRELIMINARIES

**Reinforcement learning** can be formalized as a Markov decision process, $\mathcal{M} = (\mathcal{S}, \mathcal{A}, T, r, \rho, \gamma)$, where $\mathcal{S}$ is the state space, $\mathcal{A}$ is the action space, $T : \mathcal{S} \times \mathcal{A} \rightarrow \Delta_{\mathcal{S}}$ is the transition kernel that defines the next state distribution conditioned on the current state and the current action (*e.g.*, $s' \sim T(\cdot \mid s, a)$), $r : \mathcal{S} \times \mathcal{A} \rightarrow [0, 1]$ is the reward function, $\rho \in \Delta_S$ is the initial state distribution, and $\gamma \in [0, 1)$ is the discount factor. We also assume we have access to a prior offline dataset $D = \{(s_0^i, a_0^i, r_0^i, s_1^i, a_1^i, r_1^i, \cdots, s_H^i)\}_{i=1}^{|D|}$ where the goal is to learn a policy, $\pi : \mathcal{S} \rightarrow \Delta_{\mathcal{A}}$ that maximizes its return, $\eta(\pi) = \mathbb{E}_{s_{t+1} \sim T(\cdot \mid s_t, a_t), a_t \sim \pi(\cdot \mid s_t), s_0 \sim \rho} \left[ \sum_{t=0}^{\infty} \gamma^t r(s_t, a_t) \right]$. We call a policy that attains the maximum return as an *optimal policy*, $\pi^\star$.

**Temporal difference learning.** Modern value-based reinforcement learning methods often learn a critic network, $Q : \mathcal{S} \times \mathcal{A} \rightarrow \mathbb{R}$ parameterized by $\phi$ to approximate the expected return starting from state $s$ and action $a$, and $\phi$ is often trained using the temporal-difference (TD) loss:

$$L(\phi) = \mathbb{E}_{s,a,s' \sim \mathcal{D}} \left[ (Q_\phi(s,a) - r(s,a) - \gamma \max_{a'} Q_{\bar{\phi}}(s', a'))^2 \right], \tag{1}$$

where $\bar{\phi}$ is often set to be an exponential moving average of $\phi$.

**Implicit value backup.** Instead of using $\max_{a'} Q(s', a')$ as the TD target, we can use an *implicit maximization* loss function $f_{\text{imp}}$ to learn $V_\xi(s)$ to approximate it (Kostrikov et al., 2022):

$$L(\xi) = \mathbb{E}_{s,a \sim \mathcal{D}} \left[ f_{\text{imp}}^\kappa(\bar{Q}(s,a) - V_\xi(s)) \right]. \tag{2}$$

Two popular choices of $f_{\text{imp}}^\kappa$ are (1) expectile: $f_{\text{expectile}}^\kappa(c) = |\kappa - \mathbb{I}_{c<0}|c^2$, and (2) quantile: $f_{\text{quantile}}^\kappa(c) = |\kappa - \mathbb{I}_{c<0}||c|$, for any real value $\kappa \in [0.5, 1)$. At the optimum of $L(\xi)$, $V_\xi(s)$ approximates the $\kappa$-expectile/quantile of the distribution of the TD target for $Q(s,a)$, induced by the

data distribution $\mathcal{D}$. With such a technique, we no longer need to explicitly find the action $a$ that maximizes $Q(s, a)$ and can use $V_\xi(s)$ as the backup target:

$$L(\phi) = \mathbb{E}_{s,a,s' \sim \mathcal{D}} \left[ (Q_\phi(s, a) - r(s, a) - \gamma V_\xi(s'))^2 \right]. \tag{3}$$

**Multi-step return backup.** TD learning can sometimes struggle because regressing the value network towards its own potentially inaccurate value estimates amplifies the value estimation errors further. To tackle this challenge, we can instead sample a trajectory segment, $(s_t, a_t, s_{t+1}, \cdots, a_{t+n-1}, s_{t+n})$, to construct an $n$-step return backup target from states $h$ steps ahead:

$$L_{\text{ns}}(\phi) = \mathbb{E}_{s_t, a_t, \cdots, s_{t+n}} \left[ \left( Q_\phi(s_t, a_t) - R_{t:t+n} - \gamma^n \bar{Q}(s_{t+n}, a_{t+n}^\star) \right)^2 \right], \tag{4}$$

where $a_{t+n}^\star = \arg\max_{a_{t+n}} Q(s_{t+n}, a_{t+n})$, $R_{t:t+n} := \sum_{t'=t}^{t+n-1} \gamma^{t'-t} r(s_{t'}, a_{t'})$. The $n$-step return backup reduces the effective horizon by a factor of $n$, alleviating the bootstrapping bias problem.

**Action chunking critic.** Alternatively, one may learn an action chunking critic to estimate the value of a short sequence of actions (an *action chunk*), $a_{t:t+h} := (a_t, a_{t+1}, \cdots, a_{t+h-1})$ instead: $Q(s_t, a_{t:t+h})$ (Seo & Abbeel, 2025; Li et al., 2025a; Tian et al., 2025; Li et al., 2025b). The TD backup loss for such a critic is naturally multi-step:

$$L_{\text{QC}}(\phi) = \mathbb{E}_{s_{t:t+h+1}, a_{t:t+h}} \left[ \left( Q_\phi(s_t, a_{t:t+h}) - R_{t:t+h} - \gamma^h \bar{Q}(s_{t+h}, a_{t+h:t+2h}^\star) \right)^2 \right], \tag{5}$$

where again $a_{t+h:t+2h}^\star = \arg\max_{a_{t+h:t+2h}} Q(s_{t+h}, a_{t+h:t+2h})$.

## 4 WHEN AND HOW SHOULD WE USE ACTION CHUNKING FOR Q-LEARNING?

In this section, we build a theoretical foundation for Q-learning with action chunking critics. We start by providing a formal definition of our key *open-loop consistent* condition (Definition 2), quantifying the value estimation bias incurred from backing up on non-action chunking data (Theorem 1) and the optimality of action chunking policy (Theorem 2) using this condition. Finally, we characterize the key *optimality variability* conditions under which the closed-loop execution of a learned action chunking policy is close to the optimal closed-loop policy (Theorem 3).

### 4.1 ASSUMPTIONS AND NOTATIONS

To build the foundation of our analysis, we start by describing the trajectory data distribution that we use for Q-learning and the trajectory distribution induced by an action chunking policy. In particular, we assume that the trajectory data distribution obeys the transition dynamics $T$:

> **Assumption 1** (Data Obeys the Transition Dynamics)   $\mathcal{D} \in \Delta_{\mathcal{T}}$ is a trajectory distribution generated by rolling out a behavior policy from a distribution of $s_t \sim \mu$. The behavior policy can be non-Markovian (*i.e.*, $\pi_\beta(a_{t+k} \mid s_{t:t+k+1}, a_{t:t+k})$). Each subsequent state is generated according to the dynamics of the MDP $\mathcal{M}$: $s_{t+k+1} \sim T(\cdot \mid s_{t+k}, a_{t+k}), \forall k \in \{0, 1, \cdots, h-1\}$. The resulting trajectory is $\{s_t, s_{t+1}, \cdots, s_{t+h}, a_t, a_{t+1}, \cdots, a_{t+h}\} \in \mathcal{T} = \mathcal{S}^h \times \mathcal{A}^h$.

Next, we formally define the open-loop trajectory distribution that we would obtain if we take the same actions in the data and roll them out open-loop for $h$ steps in the MDP.

> **Definition 1** (Open-loop Trajectory)   From any data distribution $\mathcal{D}$, we use $\pi_\mathcal{D}^\circ : \mathcal{S} \to \Delta_{\mathcal{A}^h}$ to denote an action chunking policy that admits the same marginal distribution as $\mathcal{D}$:
>
> $$\pi_\mathcal{D}^\circ(a_{t:t+h} \mid s_t) := P_\mathcal{D}(a_{t:t+h} \mid s_t). \tag{6}$$
>
> Rolling out this action chunking policy by carrying out actions in chunks induces a trajectory distribution $P_\mathcal{D}^\circ \in \Delta_{\mathcal{S}^{h+1}, \mathcal{A}^h}$ that is generally different from $P_\mathcal{D}$:
>
> $$P_\mathcal{D}^\circ(s_{t+1:t+h+1}, a_{t:t+h} \mid s_t) := \pi_\mathcal{D}^\circ(a_{t:t+k} \mid s_t) \prod_{k=0}^{h-1} T(s_{t+k+1} \mid s_{t+k}, a_{t+k}). \tag{7}$$

Finally, we introduce a set of notations and conventions that we use in our theoretical analysis. We use $a_{t:t+h}$ to denote an action chunk of length $h$: $(a_t, a_{t+1}, \cdots, a_{t+h-1})$ (not including $a_{t+h}$). We use the subscript $[\cdot]_{\mathrm{ac}}$ for all action chunking policies or value functions, $\hat{[\cdot]}$ to denote the *nominal* (*i.e.*, estimated) value (in contrast to the *actual* without the '^'), and $[\cdot]^+$ to denote something that is *learned* from the data (usually defaults to $\mathcal{D}$). For example, $\hat{V}_{\mathrm{ac}}^+ : \mathcal{S} \to [0, 1/(1-\gamma)]$ is the *nominal* value (*i.e.*, expected discounted return) of an action chunking policy $\pi_{\mathrm{ac}}^+$ learned from $\mathcal{D}$, whereas $V_{\mathrm{ac}}^+$ is the *actual* value of the same action chunking policy (where the value is obtained by rolling the policy out in the MDP with open-loop action chunks). As we will elucidate in the next section, the *nominal* value and the *actual* value of a policy are usually different, and hence making this differentiation critical in our analysis. We also use $[\cdot]^\star$ to denote the optimal policy or value function under the constraint of the policy class (*e.g.*, $\pi_{\mathrm{ac}}^\star$ for the optimal action chunking policy and $\pi^\star$ for the optimal closed-loop 1-step policy). Finally, we use $H = 1/(1-\gamma)$, $\bar{H} = 1/(1-\gamma^h)$ to denote the effective horizon for 1-step TD backup and $h$-step TD backup respectively.

## 4.2 WEAK AND STRONG OPEN-LOOP CONSISTENCY (OLC)

From the definition above, we have demonstrated that replaying the actions from the trajectory data distribution $P_\mathcal{D}$ in an open-loop manner may result in a different trajectory distribution, $P_\mathcal{D}^\circ$. This discrepancy between $P_\mathcal{D}^\circ$ and $P_\mathcal{D}$ has not been carefully analyzed by prior work but can play a huge role in the optimal policy that action chunking Q-learning converges to. To characterize this discrepancy, we use a notion of consistency as defined below.

> **Definition 2** (Open-Loop Consistency) $\quad \mathcal{D}$ is **weakly** $\varepsilon_h$-open-loop consistent if for every $s_t \in \mathcal{S}$ with $P_\mathcal{D}(s_t) > 0$ (*i.e.*, $s_t \in \mathrm{supp}(P_\mathcal{D}(s_t))$),
>
> $$D_{\mathrm{TV}}(P_\mathcal{D}^\circ(s_{t+h'}, a_{t+h'} \mid s_t) \,\|\, P_\mathcal{D}(s_{t+h'}, a_{t+h'} \mid s_t)) \le \varepsilon_h, \forall h' \in \{1, 2, \cdots, h-1\}, \quad (8)$$
> $$D_{\mathrm{TV}}(P_\mathcal{D}^\circ(s_{t+h} \mid s_t) \,\|\, P_\mathcal{D}(s_{t+h} \mid s_t)) \le \varepsilon_h. \quad (9)$$
>
> $\mathcal{D}$ is **strongly** $\varepsilon_h$-open-loop consistent if additionally for every $a_{t:t+h} \in \mathrm{supp}(P_\mathcal{D}(a_{t:t+h} \mid s_t))$,
>
> $$D_{\mathrm{TV}}(T(s_{t+h'} \mid s_t, a_{t:t+h'}) \,\|\, P_\mathcal{D}(s_{t+h'} \mid s_t, a_{t:t+h})) \le \varepsilon_h, \forall h' \in \{1, 2, \cdots, h\}, \quad (10)$$
>
> where we use $T(s_{t+h'} \mid s_t, a_{t:t+h'})$ to denote the distribution of the future state $s_{t+h'}$ after carrying out the action sequence $a_{t:t+h'}$ in the environment open-loop from the current state $s_t$.

Intuitively, $\mathcal{D}$ is $\varepsilon_h$-open-loop consistent if, when executing the same sequence of actions from it open-loop from $s_t$, the resulting marginal distribution of the state-action $h$ steps into the future (*i.e.*, $s_{t+h}$) deviates from the corresponding distribution in the dataset by at most $\varepsilon_h$ in total variation distance. The strong version (Equation (10)) requires the total variation distance bound to hold for every action sequence in the support, whereas the weak version (Equations (8) and (9)) only requires the bound to hold in expectation. See Section G.2 for examples of weakly open-loop consistent data.

## 4.3 VALUE LEARNING BIAS OF ACTION CHUNKING Q-LEARNING

Next, we show that the *weak* open-loop consistency of $\mathcal{D}$ alone is sufficient to show that *behavior* value iteration of an action chunking critic results in a *nominal* value function (*i.e.*, $\hat{V}_{\mathrm{ac}}$) with a bounded bias from the *true* value (*i.e.*, $V_{\mathrm{ac}}$) of the behavior cloning action chunking policy $\tilde{\pi}_{\mathrm{ac}}$:

> **Theorem 1** (AC Value Bias) $\quad$ Let $\hat{V}_{\mathrm{ac}} : \mathcal{S} \to [0, 1/(1-\gamma)]$ be a solution of
>
> $$\hat{V}_{\mathrm{ac}}(s_t) = \mathbb{E}_{s_{t+1:t+h+1}, a_{t:t+h} \sim P_\mathcal{D}(\cdot \mid s_t)} \left[ R_{t:t+h} + \gamma^h \hat{V}_{\mathrm{ac}}(s_{t+h}) \right], \quad (11)$$
>
> with $R_{t:t+h} = \sum_{t'=t}^{t+h} \gamma^{t'-t} r(s_{t'}, a_{t'})$ and $V_{\mathrm{ac}}$ is the true value of $\tilde{\pi}_{\mathrm{ac}} : s_t \mapsto P_\mathcal{D}(a_{t:t+h} \mid s_t)$. If $\mathcal{D}$ is weakly $\varepsilon_h$-open-loop consistent, then for all $s_t \in \mathrm{supp}(P_\mathcal{D}(s_t))$,
>
> $$\left| V_{\mathrm{ac}}(s_t) - \hat{V}_{\mathrm{ac}}(s_t) \right| \le \frac{\gamma \varepsilon_h}{(1-\gamma)(1-(1-\varepsilon_h)\gamma^h)} \le \varepsilon_h H \bar{H}. \quad (12)$$

Furthermore, we show that this bound is *tight* (Theorem 4 in Section F). The proofs can be found in Section H.2 and Section H.3. A direct consequence of these results is that the *true* value of the optimal action chunking policy is close to that of the optimal closed-loop policy:

**Corollary 1** (Optimality Gap for AC Policy)   Let $\mathcal{D}^\star$ be the data collected by any optimal policy $\pi^\star$. If $\mathcal{D}^\star$ is weakly $\varepsilon_h$-open-loop consistent, then for all $s_t \in \mathrm{supp}(P_{\mathcal{D}^\star}(s_t))$,

$$V^\star(s_t) - V_{\mathrm{ac}}^\star(s_t) \leq V^\star(s_t) - \tilde{V}_{\mathrm{ac}}(s_t) \leq \frac{\gamma \varepsilon_h}{(1-\gamma)(1-(1-\varepsilon_h)\gamma^h)} \leq \varepsilon_h H \bar{H}, \qquad (13)$$

where $V^\star$ is the value of the optimal policy $\pi^\star$, $V_{\mathrm{ac}}^\star$ is the *true* value of the optimal action chunking policy, and $\tilde{V}_{\mathrm{ac}}$ is the *true* value of the action chunking policy from cloning the data $\mathcal{D}^\star$:

$$\tilde{\pi}_{\mathrm{ac}}^{\mathcal{D}^\star}(a_{t:t+h} \mid s_t) : s_t \mapsto P_{\mathcal{D}^\star}(\cdot \mid s_t). \qquad (14)$$

We show that this bound is also *tight* (Corollary 2 in Section F). The proofs can be found in Section H.4 and Section H.5. Next, we analyze the performance of the action chunking policy obtained by Q-learning as a solution of the bellman optimality equation under $\mathrm{supp}(\mathcal{D})$:

$$\hat{Q}_{\mathrm{ac}}^+(s_t, a_{t:t+h}) = \mathbb{E}_{s_{t+1:t+h+1} \sim P_{\mathcal{D}}(\cdot | s_t, a_{t:t+h})} \left[ R_{t:t+h} + \gamma^h \hat{Q}_{\mathrm{ac}}^+(s_{t+h}, \pi_{\mathrm{ac}}^+(s_{t+h})) \right], \qquad (15)$$

where $\pi_{\mathrm{ac}}^+$ is defined as $\pi_{\mathrm{ac}}^+ : s_t \mapsto \arg\max_{a_{t:t+h} \in \mathrm{supp}(P_{\mathcal{D}}(a_{t:t+h}|s_t))} \hat{Q}_{\mathrm{ac}}^+(s_t, a_{t:t+h})$.

With only the weak open-loop consistency condition, the worst-case performance of the action chunking policy may be arbitrarily low (see a formal statement in Proposition 2 with proof available in Section H.6). Intuitively, the chunked critic $Q(s_t, a_{t:t+h})$ has no way of differentiating a low-probability, 'lucky' success from a closed-loop, high-probability success. This can cause the learned policy $\pi_{\mathrm{ac}}^+$ to erroneously prefer very low-value action chunks even when the optimal action chunks are available in the data distribution. Fortunately, the strong open-loop consistency (Equation (10)) is sufficient for optimality of action chunking Q-learning as quantified by the following bound:

**Theorem 2** (AC Q-Learning under Strong OLC)   If $\mathcal{D}$ and $\mathcal{D}^\star$ are strongly $\varepsilon_h$-open-loop consistent and $\mathrm{supp}(P_{\mathcal{D}}(s_t, a_{t:t+h})) \supseteq \mathrm{supp}(P_{\mathcal{D}^\star}(s_t, a_{t:t+h}))$, then for all $s_t \in \mathrm{supp}(P_{\mathcal{D}^\star}(s_t))$,

$$V^\star(s_t) - V_{\mathrm{ac}}^+(s_t) \leq \frac{\varepsilon_h \gamma}{1-\gamma} \left[ \frac{2}{1-(1-2\varepsilon_h)\gamma^h} + \frac{1}{1-(1-\varepsilon_h)\gamma^h} \right] \leq 3\varepsilon_h H \bar{H}, \qquad (16)$$

where $V^\star$ is the value of a closed-loop optimal policy and $V_{\mathrm{ac}}^+$ is the *true* value of $\pi_{\mathrm{ac}}^+$.

Theorem 2 (proof in Section H.7) shows that as long as both $\mathcal{D}$ and $\mathcal{D}^\star$ satisfy the strongly open-loop consistency condition and $\mathcal{D}$ contains the behavior in $\mathcal{D}^\star$, Q-learning with action chunking is guaranteed to converge to a near-optimal action chunking policy. Also, we show this bound is *tight* with Theorem 5 in Section F (with proof in Section H.8). Up to now, none of the bounds that we have shown so far depend on the sub-optimality of the data. Indeed, we can make the data arbitrarily sub-optimal while the action chunking policy learning is still guaranteed to be near optimal wheras naïve $n$-step return may suffer (see Section G.1).

### 4.4   CLOSED-LOOP EXECUTION OF ACTION CHUNKING POLICY

If we reuse the same strongly $\varepsilon_h$-open-loop consistency assumption, we can guarantee that closed-loop execution of the action chunking policy is also near-optimal (proof available in Section H.9):

**Proposition 1** (Optimality of Closed-loop Execution of Action Chunking Policy)   Let $V^\bullet$ be the value of the one-step policy, $\pi^\bullet$, as a result of the closed-loop execution of the action chunking policy $\pi_{\mathrm{ac}}^+$ learned from $\mathcal{D}$. That is, for each $s_t \in \mathrm{supp}(P_{\mathcal{D}}(s_t))$,

$$\pi^\bullet(s_t) = a_t^+, \quad \text{where } a_{t:t+h}^+ = \pi_{\mathrm{ac}}^+(s_t). \qquad (17)$$

If $\mathcal{D}$ and $\mathcal{D}^\star$ are both strongly $\varepsilon_h$-open-loop consistent and $\mathrm{supp}(P_{\mathcal{D}}(s_t, a_{t:t+h})) \supseteq \mathrm{supp}(P_{\mathcal{D}^\star}(s_t, a_{t:t+h}))$, then for all $s_t \in \mathrm{supp}(P_{\mathcal{D}^\star}(s_t))$,

$$V^\star(s_t) - V^\bullet(s_t) \leq \frac{\varepsilon_h \gamma}{(1-\gamma)^2} \left[ \frac{2}{1-(1-2\varepsilon_h)\gamma^h} + \frac{1}{1-(1-\varepsilon_h)\gamma^h} \right] \leq 3\varepsilon_h H^2 \bar{H}. \qquad (18)$$

However, we are paying up to a horizon factor $H$ in sub-optimality gap. Can we do better?

In practical applications, the data distributions that we are dealing with often have more structure. For example, it is common to have a dataset consisting of multiple sources where each data source is collected by either a human expert or a scripted policy that exhibits a somewhat predictable behavior (*e.g.*, after a robot arm picks up a cube, it will always move up rather than dropping it right away). We formalize this kind of structures as a notion of optimality variability as follows:

---

**Definition 3** (Optimality Variability)   $\mathcal{D}$ exhibits $\vartheta_h$-bounded variability in optimality conditioned on an event $X$ if

$$\max_{\mathrm{supp}(P_{\mathcal{D}}(\cdot|X))} \left[ R_{t:t+h} + \gamma^h V^\star(s_{t+h}) \right] - \min_{\mathrm{supp}(P_{\mathcal{D}}(\cdot|X))} \left[ R_{t:t+h} + \gamma^h V^\star(s_{t+h}) \right] \le \vartheta_h. \quad (19)$$

---

It turns out that if (1) the data distribution is a mixture of a bunch of data sources where the optimality variability conditioned on the *current actions* is bounded within each data source, and additionally (2) the optimality variability conditioned on the *current action chunks* is bounded globally across mixture, we can form a much stronger bound on the optimality of $\pi^\bullet$:

---

**Theorem 3** (Closed-loop AC Policy under Bounded OV)   Let $\mathcal{D}^\star$ be the data distribution collected by an optimal policy. Assume $\mathcal{D}$ can be decomposed into a mixture of data distributions $\{\mathcal{D}^\star, \mathcal{D}_1, \mathcal{D}_2, \cdots \mathcal{D}_M\}$ such that each data distribution component satisfies Assumption 1 and for some $\vartheta_h^L, \vartheta_h^G \ge 0$, they satisfy the following two conditions:

**1. Locally bounded optimality variability condition**: every $\mathcal{D}_i$ (including $\mathcal{D}^\star$) exhibits $\vartheta_h^L$-bounded variability in optimality conditioned on $s_t, a_t$ for all $(s_t, a_t) \in \mathrm{supp}(P_{\mathcal{D}_i}(s_t, a_t))$, and

**2. Globally bounded optimality variability condition**: $\mathcal{D}$ as a whole exhibits $\vartheta_h^G$-variability in optimality conditioned on $s_t, a_{t:t+h}$ for all $(s_t, a_{t:t+h}) \in \mathrm{supp}(P_{\mathcal{D}}(s_t, a_{t:t+h}))$.

Then for all $s_t \in \mathrm{supp}(P_{\mathcal{D}^\star}(s_t))$,

$$V^\star(s_t) - V^\bullet(s_t) \le \frac{\vartheta_h^L}{1-\gamma} + \frac{\vartheta_h^G + \gamma^h \min(\vartheta_h^L, \vartheta_h^G)}{(1-\gamma)(1-\gamma^h)} \le \vartheta_h^L H + 2\vartheta_h^G H \bar{H}. \quad (20)$$

---

We also show that Theorem 3 is tight (see Theorem 6 in Section F with proof in Section H.13).

Overall, compared to executing the action chunking policy in open-loop chunks, closed-loop execution attains a similar bound under the strongly $\varepsilon_h$-open-loop consistent assumption, and excels under the bounded optimality variability assumptions. Conceptually, closed-loop execution of the learned action chunking policy decouples the open-loop execution horizon (policy chunk length) from the value-learning horizon (critic chunk length). Such decoupling inherits the strength of action chunking TD and 1-step TD: (1) the value learning speedup of action chunking Q-learning, and (2) the reactivity of a standard, single-step policy. Furthermore, executing the first action (or more generally a partial chunk) of the original action chunk also brings practical benefits: it removes the need to explicitly train a policy to predict the full action chunk all at once, which can be especially challenging when the chunk size grows big. Can we develop a practical method that realizes such potential?

## 5   DECOUPLED Q-CHUNKING

In this section, we propose a new algorithm that enjoys the benefits of value backup speedup of critic chunking while avoiding the difficulty of learning an open-loop action chunking policy with a large chunk size. Our core idea is to decouple the chunk size of the critic from that of the policy where the policy only predicts a partial action chunk. In particular, we train a policy $\pi(a_{t:t+h_a} \mid s_t)$ to output an action chunk (with a size of $h_a \ll h$) using the following objective:

$$L(\pi) := -\mathbb{E}_{a_{t:t+h_a} \sim \pi(\cdot|s_t)}[Q_\phi(s_t, [a_{t:t+h_a}, a^\star_{t+h_a:t+h}])], \quad (21)$$

where $[a_{t:t+h_a}, a^\star_{t+h_a:t+h}]$ represents the concatenation of two partial action chunks (size $h_a$ and size $h - h_a$) into a full action chunk $a_{t:t+h}$ of size $h$, and $a^\star_{t+h_a:t+h}$ is the best 'second-half' of the action chunk that maximizes the critic value under $Q_\phi$:

$$a^\star_{t+h_a:t+h} := \arg\max_{a_{t+h_a:t+h}} Q_\phi(s_t, [a_{t:t+h_a}, a_{t+h_a:t+h}]). \quad (22)$$

Essentially, we want our policy to predict the partial action chunk (of size $h_a$) within an optimal action chunk of size $h$, rather than the entire optimal action chunk. This lowers the policy expressivity requirement and hence the learning challenges associated with it.

However, directly optimizing the objective in Equation (21) does not lead to a new algorithm because taking the maximization over $a_{t+h_a:t+h}$ seemingly requires us to learn a policy of the original chunk size anyways. To address this issue, we learn a separate partial critic $Q_\psi^P$, which only takes in the partial action chunk (of size $h_a$) as input, to approximate the maximum value this partial action chunk can achieve when it is extended to the full action chunk (of size $h$):

$$Q_\psi^P(s_t, a_{t:t+h_a}) \approx Q_\phi(s_t, [a_{t:t+h_a}, a_{t+h_a:t+h}^\star]). \tag{23}$$

To train $Q_\psi^P$, we can use an *implicit maximization* loss function (as described in Equation (2)):

$$L(\psi) := f_{\text{imp}}^{\kappa_d}(\bar{Q}_\phi(s_t, a_{t:t+h}) - Q_\psi^P(s_t, a_{t:t+h_a})), \tag{24}$$

where $s_t, a_{t:t+h}$ are sampled from the offline dataset $D$. As a result, the partial critic, $Q_\psi^P$, is distilled from the original critic via optimistic regression, where its optimum $Q_\psi^\star(s_t, a_{t:t+h_a})$ approximates $Q_\phi(s_t, [a_{t:t+h_a}, a_{t+h_a:t+h}^\star])$ in Equation (21), conveniently removing the need for training a policy to predict the whole optimal action chunk entirely. This allows us to simplify the policy objective as

$$L(\pi) := -\mathbb{E}_{a_{t:t+h_a} \sim \pi(\cdot|s_t)} \left[ Q_\psi^P(s_t, a_{t:t+h_a}) \right]. \tag{25}$$

In summary, DQC trains a policy to predict a partial chunk, $a_{t:t+h_a}$ (of size $h_a$), by hill climbing the value of a partial critic $Q_\psi^P(s_t, a_{t:t+h_a})$ that is distilled from the original chunked critic $Q_\phi(s_t, a_{t:t+h})$ via an implicit maximization loss. This allows our policy to fully leverage the chunked critic $Q_\phi$ (and thus the value speedup benefits associated with Q-chunking) without the need to predict the full action chunk (of size $h$), mitigating the learning challenge of an action chunking policy.

**Practical considerations for offline RL.** Finally, we describe several implementation details that we find to work well in the offline RL setting, which our experiments focus on. Our implementation draws inspiration from a prior method, IDQL (Hansen-Estruch et al., 2023).

We first train a behavior cloning flow policy $\pi_\beta$ using a standard flow-matching objective (Liu et al., 2023) on the offline dataset $D$. Then, we approximate the policy optimization objective in Equation (25) using best-of-N sampling without explicitly modeling $\pi$:

$$a_{t:t+h_a}^\star \leftarrow \arg\max_{\{a_{t:t+h_a}^i\}_{i=1}^N} Q_\psi^P(s_t, a_{t:t+h_a}), \quad \text{where } a_{t:t+h_a}^1, \cdots, a_{t:t+h_a}^N \sim \pi_\beta(\cdot \mid s_t), \tag{26}$$

and $a_{t:t+h_a}^\star$ is output of the policy that we extract from $Q_\psi^P$ for state $s_t$. Essentially, this sampling procedure is a test-time approximation of the objective in Equation (25), where it outputs an action (chunk) that maximizes $Q_\psi^P$, subject to the behavior prior, as modeled by $\pi_\beta$.

For TD learning of $Q_\phi$, directly computing the TD backup target from either $Q_{\bar\phi}$ or $Q_{\bar\psi}^P$ is computationally expensive, as either requires samples from the current policy, which is approximated via the best-of-N sampling procedure as described above. Instead, we use the implicit value backup (Kostrikov et al., 2022) (*i.e.*, as described in Equation (2)) to approximate the target:

$$L(\xi) = f_{\text{quantile}}^{\kappa_b}(\bar{Q}_\psi^P(s_t, a_{t:t+h_a}) - V_\xi(s_t)), \tag{27}$$

where we pick the quantile regression loss as the implicit maximization loss function. This is because the Q-value obtained from best-of-N sampling can be seen as the largest order statistic of a random batch (of size $N$) of the behavior Q-values. Such statistic estimates the behavior Q-value distribution's $\frac{N-1}{N}$-quantile, which is the same as $V_\xi(s)$ at the optimum of $L(\xi)$ if we set $\kappa_b = \frac{N-1}{N}$. In practice, we use a smaller $\kappa_b$ for numerical stability (see Table 7).

Finally, we pick the expectile regression loss for training the distilled partial critic $Q_\psi^P$ because prior work has found it to work the best among all implicit maximization loss functions (Hansen-Estruch et al., 2023). A summary of the algorithm is available in Section A, Algorithm 1.

## 6 EXPERIMENTAL SETUP

We conduct experiments to evaluate the benefits of decoupling the policy chunk size and the critic chunk size on OGBench (Park et al., 2025a)—a challenging long-horizon, goal-conditioned offline

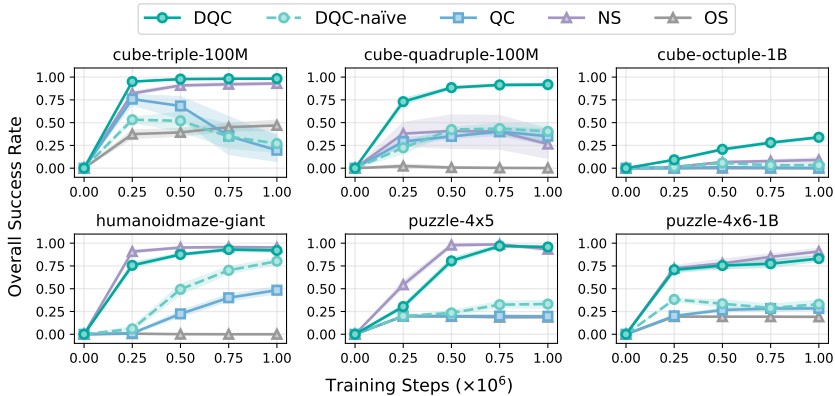

Figure 2: **Offline goal-conditioned RL results (10 seeds).** Our method (*DQC*) uses *decoupled* critic and policy chunk sizes. *QC*: Q-chunking (Li et al., 2025b); *NS*: $n$-step return backup; *OS*: 1-step TD-backup; *DQC-naïve*: same as QC but executes a partial action chunk.

| Task | FBC | HFBC | IQL | HIQL | SHARSA | OS | NS | QC | DQC-naïve | DQC |
|------|-----|------|-----|------|--------|-----|-----|-----|-----------|-----|
| cube-triple-100M | $54_{[51,56]}$ | $56_{[53,59]}$ | $66_{[63,67]}$ | $35_{[31,39]}$ | $83_{[81,85]}$ | $47_{[41,53]}$ | $93_{[91,94]}$ | $20_{[7,36]}$ | $27_{[18,38]}$ | $\mathbf{98}_{[\mathbf{98,99}]}$ |
| cube-quadruple-100M | $34_{[32,37]}$ | $37_{[34,40]}$ | $53_{[52,55]}$ | $24_{[21,28]}$ | $64_{[62,68]}$ | $0_{[0,0]}$ | $27_{[11,43]}$ | $35_{[26,43]}$ | $40_{[29,49]}$ | $\mathbf{92}_{[\mathbf{90,93}]}$ |
| cube-octuple-1B | $0_{[0,0]}$ | $28_{[26,29]}$ | $0_{[0,0]}$ | $20_{[17,23]}$ | $\mathbf{34}_{[\mathbf{31,36}]}$ | $0_{[0,0]}$ | $9_{[6,12]}$ | $0_{[0,0]}$ | $3_{[1,5]}$ | $\mathbf{34}_{[\mathbf{33,35}]}$ |
| humanoidmaze-giant | $1_{[1,2]}$ | $6_{[4,8]}$ | $3_{[2,5]}$ | $24_{[22,26]}$ | $19_{[16,23]}$ | $0_{[0,0]}$ | $\mathbf{95}_{[\mathbf{94,97}]}$ | $48_{[45,52]}$ | $80_{[77,83]}$ | $92_{[90,94]}$ |
| puzzle-4x5 | $0_{[0,0]}$ | $0_{[0,0]}$ | $20_{[19,20]}$ | $0_{[0,0]}$ | $1_{[1,2]}$ | $19_{[18,19]}$ | $93_{[91,95]}$ | $20_{[20,20]}$ | $33_{[29,37]}$ | $\mathbf{96}_{[\mathbf{95,97}]}$ |
| puzzle-4x6-1B | $1_{[0,1]}$ | $4_{[3,5]}$ | $6_{[3,9]}$ | $9_{[5,13]}$ | $64_{[60,68]}$ | $19_{[19,20]}$ | $\mathbf{91}_{[\mathbf{86,94}]}$ | $28_{[27,30]}$ | $33_{[28,38]}$ | $83_{[80,86]}$ |

Table 1: **Comparisons with prior methods (10 seeds).** Our method outperforms SHARSA (Park et al., 2025b) (the previous state-of-the-art method on this benchmark) on most environments.

RL benchmark consisting of a diverse set of environments (from manipulation to locomotion). We now describe our main comparisons. To start with, we consider several direct ablation baselines where the same algorithm backbone is being used (*i.e.*, implicit value backup and best-of-N sampling):

**QC** (Q-chunking (Li et al., 2025b)) uses a single critic that has the same chunk length as that of the policy (*i.e.*, $h = h_a$). This baseline tests whether having *decoupled* chunk sizes is important.

**DQC-naïve** is a naïve attempt at decoupling the critic chunk size from the policy chunk size, where it takes the QC policy to predict full action chunks of size $h$ but only execute the first $h_a$ actions.

**NS**: $n$-step return TD backup. This baseline uses a single one-step critic (*i.e.*, $Q(s_t, a_t)$). Compared to DQC with $h = n$ and $h_a = 1$, this baseline tests whether using a chunked critic is important.

**OS**: Standard 1-step TD backup. This is the same as NS but with $n = 1$.

We also consider a few strong baselines from prior work: **FBC/HFBC** (Park et al., 2025b), **IQL/HIQL** (Kostrikov et al., 2022; Park et al., 2023), and **SHARSA** (Park et al., 2025b).

In our ablations, we also consider an additional baseline, **QC-NS**, that uses the idea of decoupled policy chunking and critic chunking, but without using a distilled critic. This baseline simply uses $n$-step return targets to directly train a critic with a chunk size of $h_a$ without implicit maximization (Equation (24)). For all our experiments, we run 10 seeds for all methods, and report the means and the 95% confidence intervals.

## 7 RESULTS

In this section, we present our experimental results to answer the following three questions:

**(Q1) Does DQC improve upon $n$-step return, Q-chunking?** Figure 2 compares DQC (ours) to both $n$-step and QC across six challenging long-horizon GCRL environments, with our method performing on par or better across the board. Table 1 shows DQC also consistently outperforms the previous state-of-the-art method on this benchmark, SHARSA (Park et al., 2025b), on all environments. For each environment, we tune DQC (ours), QC, NS, and OS (see the tuning range in Table 8) and pick

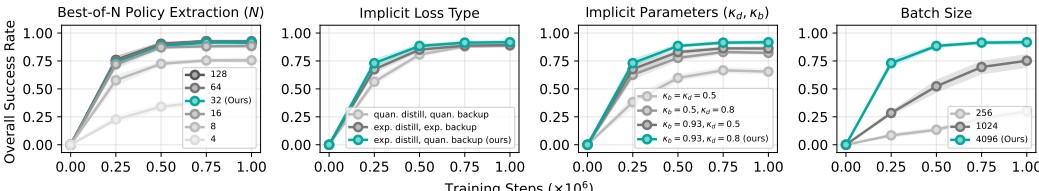

Figure 3: **Distilled critic ablations (10 seeds).** Each group in the legend contains DQC and its non-distilled counterpart with the same configuration. Our method (DQC) performs on par or better than the non-distilled counterpart across all configurations.

Figure 4: **Hyperparameter sensitivity analysis on** `cube-quadruple` **(10 seeds).** *Best-of-N*: the number of action samples drawn from $\pi_\beta(\cdot \mid s)$ during policy evaluation; *Implicit loss type*: the implicit maximization loss function used for distillation and value backup; *Batch size*: the number of examples used in each gradient step.

the best configuration (Table 6) for hyperparameters used in Figure 2 and Table 1. For all baselines from prior work (SHARSA, HIQL, IQL, HFBC, FBC), we directly use their tuned hyperparameters and run with the same batch size (*i.e.*, 4096) as used in our method and other baselines. See the complete result table for all combinations of $h, n, h_a$ in Section B.

**(Q2) Is training a separate distilled critic $Q_\psi^P$ necessary?** In Figure 3, we compare DQC to DQC without using the distilled critic across three different $(h, h_a)$ configurations: $(h = 25, h_a = 5)$, $(h = 25, h_a = 1)$, and $(h = 5, h_a = 1)$. For configurations with $h_a = 1$, the baseline without using the distilled critic is the same as the $n$-step return baseline (with $n = h$) and for the configuration with $h_a = 5$, it is the same as combining Q-chunking and $n$-step return (QC-NS). Across three configurations, DQC performs on par or better than its non-distilled counterpart. This highlights that the a separate distilled critic for the partial action chunk is necessary for the effectiveness of DQC.

**(Q3) How sensitive is DQC to its hyperparameters?** Figure 4 shows that our method is not sensitive to the implicit backup method (quantile or expectile), and somewhat sensitive to the implicit parameters $\kappa_b, \kappa_d$. In particular, DQC is still reasonably effective as long as some form of optimism is employed (*i.e.*, either $\kappa_b \neq 0.5$ or $\kappa_d \neq 0.5$). Using no optimism ($\kappa_b = \kappa_d = 0.5$) results in a big performance drop. The other important hyperparameters are $N$ in the best-of-N policy extraction and the batch size. Having large enough batch size (*i.e.*, 4096) and $N$ (*e.g.*, $N = 32$) is crucial for good performance, although increasing $N$ further (*e.g.*, $N = 128$) does not lead to better performance.

# 8 DISCUSSION

We provide a theoretical foundation for action chunking Q-learning and demonstrate how to effectively extract policies from chunked critics. Theoretically, we provide a formal analysis of action chunking Q-learning, identifying the TD backup bias that arises from open-loop inconsistency and characterizing the conditions under which action chunking Q-learning is preferred over $n$-step return learning and the conditions under which closed-loop execution of the action chunking policy is near-optimal. Empirically, we develop a new technique that enables effective policy extraction from chunked critics with long action chunks, scaling up action chunking Q-learning to much harder environments. Together, these contributions advance the goal of tackling bootstrapping bias in TD-learning. Several challenges remain, indicating promising avenues for future research. For example, our method relies on a fixed policy action chunk size $h_a$ and critic action chunk size $h$ across all states, even though the optimal action chunk size may vary by state. Developing practical methods that can support flexible, state-dependent chunk sizes would be a natural next step.

ACKNOWLEDGMENTS

This work was supported by DARPA ANSR and ONR N00014-25-1-2060. This research used the Savio computational cluster resource provided by the Berkeley Research Computing program at UC Berkeley. We would like to thank William Chen for discussions and inspiration, especially on the proof for Proposition 4. We would also like to thank Andrew Wagenmaker for suggestions and feedback on the theory (Theorems 1 and 2 and Propositions 1 and 3). We would also like to thank Dibya Ghosh for feedback on an early version of the teaser figure and Ameesh Shah for writing feedback on an early draft of the paper.

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

---

**Algorithm 1** Decoupled Q-chunking (DQC).

---

**Given:** $D, Q_\phi(s_t, a_{t:t+h}), Q_\psi^P(s_t, a_{t:t+h_a}), V_\xi(s_t), \pi_\beta(a_{t:t+h_a} \mid s_t)$

**1. Agent Update:**

$(s_{t:t+h+1}, a_{t:t+h}, r_{t:t+h}) \sim D.$       ▷ *sample trajectory chunk from the offline dataset*

Optimize $Q_\phi$ with $L(\phi) = \left( Q_\phi(s_t, a_{t:t+h}) - \sum_{k=0}^{h-1} \gamma^k r_{t+k} - \gamma^h \bar{V}_\xi(s_{t+h}) \right)^2.$

Optimize $Q_\psi^P$ with $L(\psi) = f_{\text{expectile}}^{\kappa_d} \left( \bar{Q}_\phi(s_t, a_{t:t+h}) - Q_\psi^P(s_t, a_{t:t+h_a}) \right).$

Optimize $V_\xi$ with $L(\xi) = f_{\text{quantile}}^{\kappa_b} (\bar{Q}_\psi^P(s_t, a_{t:t+h_a}) - V_\xi(s_t)),$

**2. Policy Extraction:**

$a_{t:t+h_a}^1, a_{t:t+h_a}^2, \cdots, a_{t:t+h_a}^N \sim \pi_\beta(\cdot \mid s_t)$       ▷ *sample N actions from behavior policy*

$a_{t:t+h_a}^\star \leftarrow \arg\max_{\{a_{t:t+h_a}^i\}_{i=1}^N} Q_\psi^P(s_t, a_{t:t+h_a})$       ▷ *take the action with the highest Q-value*

---

## A    ALGORITHM PSEUDOCODE

## B    FULL RESULTS

Table 2 reports the performance of our method (DQC) and baselines for all hyperparameter configurations. All of them use the same hyperparameters in Table 4 with the only exception that SHARSA, HIQL, IQL, FBC, and HFBC handle goal-sampling for training behavior cloning policies differently. We discuss this in more details in Section D.

| Method | | | c3-100M | c4-100M | c8-1B | hg | p45 | p46-1B |
|---|---|---|---|---|---|---|---|---|
| **DQC** | $h = 25$ | $h_a = 1$ | $76_{[73,80]}$ | $45_{[41,49]}$ | $10_{[8,11]}$ | $92_{[90,94]}$ | $91_{[89,92]}$ | $83_{[80,86]}$ |
| **DQC** | $h = 25$ | $h_a = 5$ | $\mathbf{98}_{[\mathbf{98,99}]}$ | $\mathbf{92}_{[\mathbf{90,93}]}$ | $\mathbf{34}_{[\mathbf{33,35}]}$ | $51_{[48,54]}$ | $\mathbf{96}_{[\mathbf{95,97}]}$ | $68_{[66,71]}$ |
| **DQC** | $h = 5$ | $h_a = 1$ | $95_{[94,97]}$ | $84_{[83,86]}$ | $0_{[0,0]}$ | $19_{[15,22]}$ | $90_{[88,92]}$ | $44_{[42,47]}$ |
| **DQC-naïve** | $h = 25$ | $h_a = 1$ | $14_{[8,22]}$ | $16_{[9,23]}$ | $1_{[0,2]}$ | $22_{[20,24]}$ | $32_{[28,36]}$ | $33_{[29,37]}$ |
| **DQC-naïve** | $h = 25$ | $h_a = 5$ | $27_{[18,38]}$ | $27_{[15,39]}$ | $3_{[1,5]}$ | $0_{[0,1]}$ | $33_{[29,37]}$ | $33_{[28,38]}$ |
| **DQC-naïve** | $h = 5$ | $h_a = 1$ | $16_{[7,30]}$ | $40_{[29,49]}$ | $0_{[0,0]}$ | $80_{[77,83]}$ | $20_{[20,20]}$ | $26_{[25,28]}$ |
| **QC** | $h = 25$ | $h_a = 25$ | $21_{[13,31]}$ | $12_{[7,18]}$ | $0_{[0,0]}$ | $0_{[0,0]}$ | $30_{[27,33]}$ | $37_{[33,42]}$ |
| **QC** | $h = 5$ | $h_a = 5$ | $20_{[7,36]}$ | $35_{[26,43]}$ | $0_{[0,0]}$ | $48_{[45,52]}$ | $20_{[20,20]}$ | $28_{[27,30]}$ |
| **QC-NS** | $n = 25$ | $h_a = 5$ | $51_{[22,80]}$ | $53_{[28,77]}$ | $18_{[10,25]}$ | $60_{[58,61]}$ | $\mathbf{95}_{[\mathbf{94,96}]}$ | $\mathbf{95}_{[\mathbf{93,97}]}$ |
| **NS** | $n = 25$ | $h_a = 1$ | $30_{[26,35]}$ | $19_{[11,28]}$ | $9_{[6,12]}$ | $\mathbf{95}_{[\mathbf{94,97}]}$ | $89_{[87,91]}$ | $\mathbf{91}_{[\mathbf{86,94}]}$ |
| **NS** | $n = 5$ | $h_a = 1$ | $93_{[91,94]}$ | $27_{[11,43]}$ | $1_{[0,3]}$ | $89_{[87,91]}$ | $93_{[91,95]}$ | $56_{[48,63]}$ |
| **OS** | $n = 1$ | $h_a = 1$ | $47_{[41,53]}$ | $0_{[0,0]}$ | $0_{[0,0]}$ | $0_{[0,0]}$ | $19_{[18,19]}$ | $19_{[19,20]}$ |
| **FBC** | | | $54_{[51,56]}$ | $34_{[32,37]}$ | $0_{[0,0]}$ | $1_{[1,2]}$ | $0_{[0,0]}$ | $1_{[0,1]}$ |
| **HFBC** | | | $56_{[53,59]}$ | $37_{[34,40]}$ | $28_{[26,29]}$ | $6_{[4,8]}$ | $0_{[0,0]}$ | $4_{[3,5]}$ |
| **IQL** | | | $66_{[63,67]}$ | $53_{[52,55]}$ | $0_{[0,0]}$ | $3_{[2,5]}$ | $20_{[19,20]}$ | $6_{[3,9]}$ |
| **HIQL** | | | $35_{[31,39]}$ | $24_{[21,28]}$ | $20_{[17,23]}$ | $24_{[22,26]}$ | $0_{[0,0]}$ | $9_{[5,13]}$ |
| **SHARSA** | | | $83_{[81,85]}$ | $64_{[62,68]}$ | $\mathbf{34}_{[\mathbf{31,36}]}$ | $19_{[16,23]}$ | $1_{[1,2]}$ | $64_{[60,68]}$ |

Table 2: **Complete results for all hyperparameter configurations across different combinations of** $h$**,** $n$ **and** $h_a$ **(10 seeds).** We adopt the following abbreviations: `c3`=`cube-triple`, `c4`=`cube-quadruple`, `c8`=`cube-octuple`, `hg`=`humanoidmaze-giant`, `p45`=`puzzle-4x5`, `p46`=`puzzle-4x6`. The hyperparameters used are specified in Tables 6 and 7.

## C    ENVIRONMENTS AND DATASETS

To evaluate our method, we consider 8 goal-conditioned environments in OGBench with varying difficulties (Figure 6). The dataset size, episode length, and the action dimension for each environment is available in Table 3. We describe each of the environments and the datasets we use as follows.

**Environment** `cube-*`: We consider three cube environments (`cube-triple`, `cube-quadruple`, `cube-octuple`). As the names suggest, the goal of these environments involve using a robot arm to manipulate 3/4/8 cubes from some initial configuration to some specified goal configuration. We

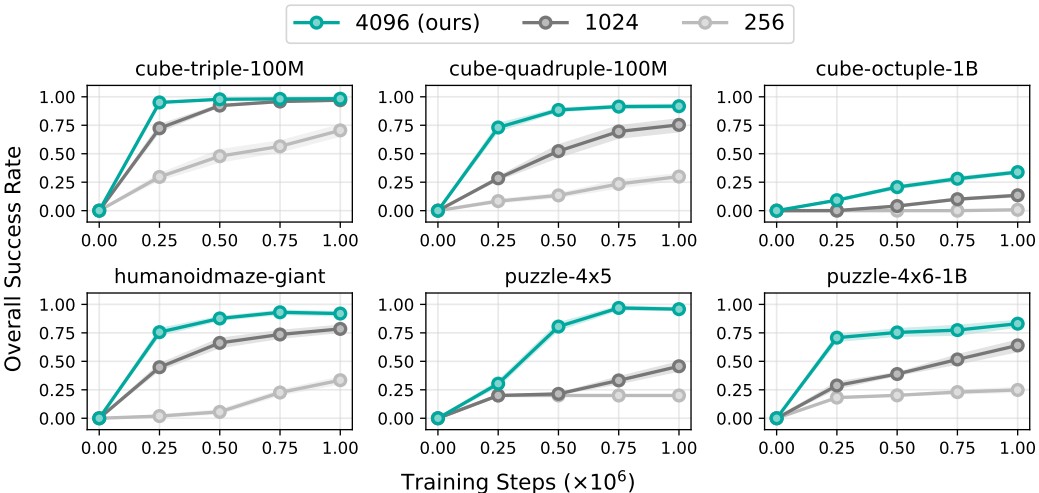

Figure 5: **Batch size sensitivity (10 seeds).** Large batch size is crucial for DQC's performance especially on hard tasks (*e.g.*, `cube-quadruple`, `cube-octuple`, `puzzle-4x5` and `puzzle-4x6`).

use the same five evaluation tasks used in OGBench (Park et al., 2025a) for `cube-triple` and `cube-quadruple` and the same five evaluation tasks used in Park et al. (2025b) for `cube-octuple`. We refer the environment detail to the corresponding references.

| Environment | Dataset Size | Episode Length | Action Dimension ($A$) |
|---|---|---|---|
| `cube-triple-100M` | 100M | 1000 | 5 |
| `cube-quadruple-100M` | 100M | 1000 | 5 |
| `cube-octuple-1B` | 1B | 1500 | 5 |
| `humanoidmaze-giant` | 4M (default) | 4000 | 21 |
| `puzzle-4x5` | 3M (default) | 1000 | 5 |
| `puzzle-4x6-1B` | 1B | 1000 | 5 |

Table 3: **Environment metadata.** For both `humanoidmaze-giant` and `puzzle-4x5`, we use the default dataset that is released in the original OGBench benchmark (Park et al., 2025a). For the other environments, we use larger datasets as we find them to be essential for achieving good performances on these environments.

**Environment `humanoidmaze-*`:** We also consider the hardest locomotion environment available in OGBench. The goal of the environment is to control and navigate a humanoid agent from some initial location to some specified goal location in a $16 \times 12$ maze. This environment also has the longest episode length (4000, more than twice as long as the second longest episode length as used in `cube-octuple`). We refer the environment detail to Park et al. (2025a).

**Environment `puzzle-*`:** Finally, we consider two environments that involve solving a combinatorial puzzle with a robot arm. The puzzle consists of a board of $4 \times 5$ or $4 \times 6$ buttons, organized as a regular grid (4 rows and 5 or 6 columns). Each button has a binary state. Whenever the end-effector of the arm touches a button, the button and all its adjacent four buttons (three or two if the button is on the edge of the grid or in the corner) flip its binary state. The goal of the environment is to transform the board from some initial state to some specified goal state. We refer the environment detail to Park et al. (2025b).

At the test-time/evaluation-time, the goal-conditioned agent is tested on five evaluation tasks for each of the six environments we consider. The overall success rate is the average over 5 tasks with 50 evaluation trials each. For the prior baselines, SHARSA, HIQL, IQL, HFBC and FBC, we run 15 evaluation trials for each task, following Park et al. (2025b).

**Datasets.** We use `play` datasets for all `cube-*` and `puzzle-*` environments and `navigate` dataset for `humanoidmaze-*`. We use the original datasets available for `humanoidmaze-giant` and `puzzle-4x5` because they are sufficient for solving the environments. Using larger datasets on these environments do not help differentiate among different methods/baselines. For each of the other environments, we use the largest dataset available from Park et al. (2025b) as we find it to be necessary to solve these environments (or achieve non-trivial performance on `cube-octuple`).

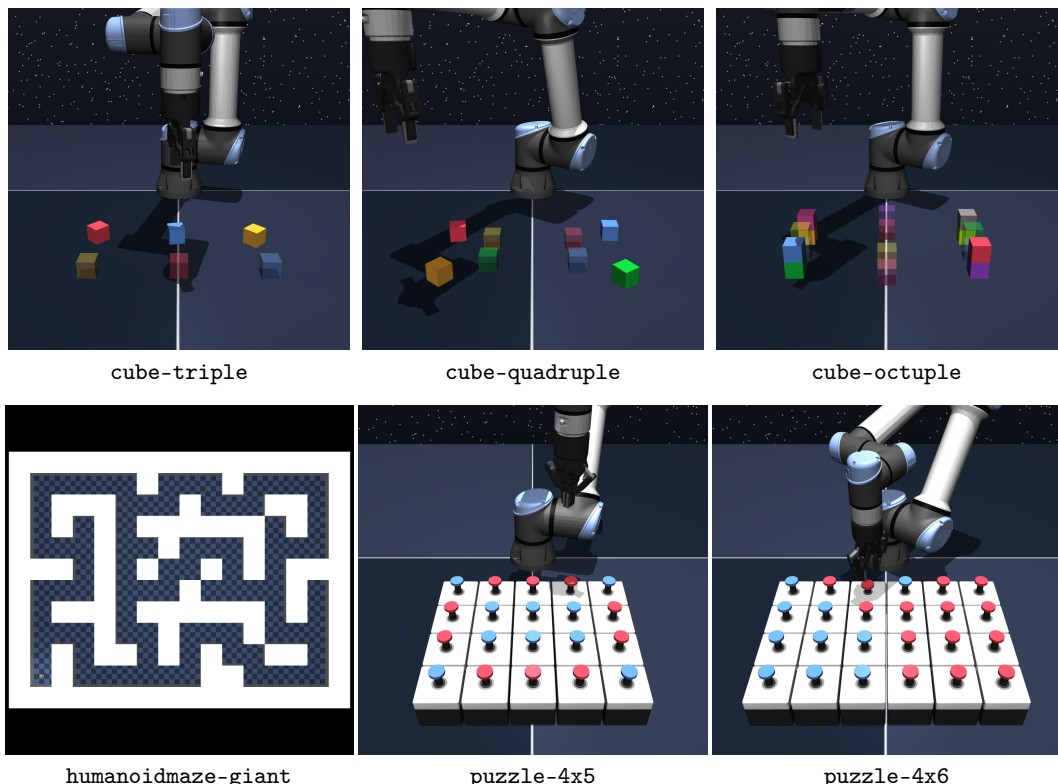

cube-triple          cube-quadruple          cube-octuple

humanoidmaze-giant          puzzle-4x5          puzzle-4x6

Figure 6: **Environments used in our experiments**.

## D   HYPERPARAMETERS AND IMPLEMENTATION DETAILS

**Hyperparameters.** Table 4 describes the common hyperparameters used in all our experiments. Tables 6 and 7 describe the environment-specific hyperparameters and Table 8 describes the range of hyperparameters we use for tuning each method.

**Goal-conditioned RL implementation details.** While we have described in the main body of the paper how DQC works as a general RL algorithm, we have not touched on how DQC and similarly all our baselines works with the goal-conditioned RL (GCRL) setting. We consider the setting where we have access to an oracle goal representation $\Psi : \mathcal{S} \to \mathcal{G}$ where $\mathcal{G}$ is the goal space (see Table 5 for the oracle goal representation description for each environment). The goal-conditioned reward function $r : (s, g) \mapsto \mathbb{I}_{\Psi(s)=g}$ is a binary reward function where its output is 1 if the goal $g$ is reached by the current state $s$. We can treat $g$ as part of an extended state $\tilde{s} = [s, g] \in \tilde{\mathcal{S}} = \mathcal{S} \times \mathcal{G}$ and learn value functions (*e.g.*, $Q_\phi(\tilde{s}, a)$) normally with such extended state.

A common practical trick in the GCRL setting is goal relabeling. That is, during training for each $(s, a)$ pair in the training batch, a goal $g$ is sampled from some distribution (*i.e.*, $p^{\mathcal{D}}(\cdot \mid s, a)$) and the reward of the transition is relabeled with the goal-conditioned reward function. Following Park et al. (2025b), the goal distribution $P^g(\cdot \mid s, a) : \mathcal{S} \times \mathcal{A} \to \Delta_{\mathcal{G}}$ is a mixture of four distributions, conditioned on the training state-action example:

$$P^g = w_{\text{cur}} P^g_{\text{cur}} + w_{\text{geom}} P^g_{\text{geom}} + w_{\text{traj}} P^g_{\text{traj}} + w_{\text{rand}} P^g_{\text{rand}}, \tag{28}$$

| Parameter | Value |
|---|---|
| Batch size | 4096 |
| Discount factor ($\gamma$) | 0.999 |
| Optimizer | Adam |
| Learning rate | $3 \times 10^{-4}$ |
| Target network update rate ($\lambda$) | $5 \times 10^{-3}$ |
| Critic ensemble size ($K$) | 2 |
| Critic target | $\min(Q_1, Q_2)$ for `cube-*` 
 $(Q_1 + Q_2)/2$ for `puzzle-*` and `humanoid-*` |
| Value loss type | binary cross entropy |
| Best-of-N sampling ($N$) | 32 |
| Number of flow steps | 10 |
| Number of training steps | $10^6$ |
| Network width | 1024 |
| Network depth | 4 hidden layers |
| Value goal sampling ($w_{\mathrm{cur}}^{\mathrm{v}}, w_{\mathrm{geom}}^{\mathrm{v}}, w_{\mathrm{traj}}^{\mathrm{v}}, w_{\mathrm{rand}}^{\mathrm{v}}$) | $(0.2, 0, 0.5, 0.3)$ |
| Actor goal sampling ($w_{\mathrm{cur}}^{\mathrm{p}}, w_{\mathrm{geom}}^{\mathrm{p}}, w_{\mathrm{traj}}^{\mathrm{p}}, w_{\mathrm{rand}}^{\mathrm{p}}$) | DQC/QC/NS/OS: $\pi_\beta$ is not goal-conditioned 
 SHARSA (`cube`): $(0, 1, 0, 0)$ 
 SHARSA (`puzzle`): $(0, 0, 1, 0)$ 
 SHARSA (`humanoidmaze`): $(0, 0, 1, 0)$ |

Table 4: **Common hyperparameters.** For the GCRL goal-sampling distribution we follow the same hyperparameters used in Park et al. (2025b).

| Environment | Goal Representation ($\Psi$) | Goal Domain ($\mathcal{G}$) |
|---|---|---|
| `cube-triple` | $(x, y, z)$ of three cubes (rel. to center) | $\mathbb{R}^9$ |
| `cube-quadruple` | $(x, y, z)$ of four cubes (rel. to center) | $\mathbb{R}^{12}$ |
| `cube-octuple` | $(x, y, z)$ of eight cubes (rel. to center) | $\mathbb{R}^{24}$ |
| `humanoidmaze-giant` | $(x, y)$ of the humanoid | $\mathbb{R}^2$ |
| `puzzle-4x5` | the binary state for each button | $\{0, 1\}^{20}$ |
| `puzzle-4x6` | the binary state for each button | $\{0, 1\}^{24}$ |

Table 5: **Oracle goal representation description for each environment.** Following Park et al. (2025b), we assume access to an oracle goal representation for each environment. More detailed definition of these oracle goal representations is available in OGBench (Park et al., 2025a).

where

1. $P_{\mathrm{cur}}^g(\cdot \mid s, a) = \delta_{\Psi(s)}$: the goal is the same as the current state;

2. $P_{\mathrm{geom}}^g(\cdot \mid s, a)$: geometric distribution over the future states in the same trajectory that $(s, a)$ is from;

3. $P_{\mathrm{traj}}^g(\cdot \mid s, a)$: uniform distribution over the future states in the same trajectory that $(s, a)$ is from; and finally

4. $P_{\mathrm{rand}}^g(\cdot \mid s, a) = \Psi(\mathcal{U}_{\mathcal{D}(s)})$: uniform distribution over the dataset ($\mathcal{D}(s)$ is the distribution of states in the dataset).

and $w_{\mathrm{cur}}, w_{\mathrm{geom}}, w_{\mathrm{traj}}, w_{\mathrm{rand}} > 0$ are the corresponding weights for each of the distribution components with $w_{\mathrm{cur}} + w_{\mathrm{geom}} + w_{\mathrm{traj}} + w_{\mathrm{rand}} = 1$.

In practice, it has been found to be beneficial to use a separate set of goal sampling weights for TD backup (Park et al., 2025a) (*i.e.*, $(w_{\mathrm{cur}}^{\mathrm{v}}, w_{\mathrm{geom}}^{\mathrm{v}}, w_{\mathrm{traj}}^{\mathrm{v}}, w_{\mathrm{rand}}^{\mathrm{v}})$) and for policy learning (*i.e.*, $(w_{\mathrm{cur}}^{\mathrm{p}}, w_{\mathrm{geom}}^{\mathrm{p}}, w_{\mathrm{traj}}^{\mathrm{p}}, w_{\mathrm{rand}}^{\mathrm{p}})$). However, in our implementation of DQC/QC/NS/OS, we do not train a goal-conditioned policy, as our policy extraction is done entirely at test-time by best-of-N sampling

| Environment | DQC $(h, h_a, \kappa_b, \kappa_d)$ | DQC-naïve $(h, h_a, \kappa_b)$ | QC-NS $(h, h_a, \kappa_b)$ | QC $(h = h_a, \kappa_b)$ | NS $(n, \kappa_b)$ | OS $(\kappa_b)$ | SHARSA $(n)$ | HIQL $(h, \kappa, \alpha)$ | IQL $(\alpha)$ | HFBC $(h)$ |
|---|---|---|---|---|---|---|---|---|---|---|
| cube-triple-100M | $(25, 5, 0.93, 0.8)$ | $(25, 5, 0.93)$ | $(25, 5, 0.93)$ | $(5, 0.93)$ | $(5, 0.5)$ | $0.5$ | $25$ | $(25, 0.5, 10)$ | $3$ | $25$ |
| cube-quadruple-100M | $(25, 5, 0.93, 0.8)$ | $(5, 1, 0.93)$ | $(25, 5, 0.93)$ | $(5, 0.93)$ | $(5, 0.7)$ | $0.7$ | $25$ | $(25, 0.5, 10)$ | $3$ | $25$ |
| cube-octuple-1B | $(25, 5, 0.93, 0.5)$ | $(25, 5, 0.93)$ | $(25, 5, 0.93)$ | $(25, 0.93)$ | $(25, 0.97)$ | $0.7$ | $25$ | $(50, 0.5, 10)$ | $10$ | $50$ |
| humanoidmaze-giant | $(25, 1, 0.5, 0.8)$ | $(5, 1, 0.9)$ | $(25, 5, 0.5)$ | $(5, 0.5)$ | $(25, 0.7)$ | $0.5$ | $50$ | $(50, 0.5, 3)$ | $0.3$ | $50$ |
| puzzle-4x5 | $(25, 5, 0.9, 0.5)$ | $(25, 5, 0.9)$ | $(25, 5, 0.7)$ | $(5, 0.9)$ | $(25, 0.7)$ | $0.7$ | $50$ | $(25, 0.7, 3)$ | $1$ | $25$ |
| puzzle-4x6-1B | $(25, 1, 0.7, 0.5)$ | $(25, 5, 0.7)$ | $(25, 5, 0.5)$ | $(5, 0.7)$ | $(25, 0.5)$ | $0.7$ | $50$ | $(25, 0.7, 3)$ | $1$ | $25$ |

Table 6: **Environment-specific hyperparameters for DQC, QC, NS, OS, SHARSA, HIQL, IQL, and HFBC.** For SHARSA, HIQL, IQL, and HFBC, we follow the hyperparameters in the original paper (Park et al., 2025b).

| Environment | DQC $h = 25, h_a = 5$ $(\kappa_b, \kappa_d)$ | DQC $h = 25, h_a = 1$ $(\kappa_b, \kappa_d)$ | DQC $h = 5, h_a = 1$ $(\kappa_b, \kappa_d)$ | QC-NS $n = 25, h_a = 5$ $\kappa_b$ | NS $n = 25$ $\kappa_b$ | NS $n = 5$ $\kappa_b$ | QC $h = 25$ $\kappa_b$ | QC $h = 5$ $\kappa_b$ | OS $\kappa_b$ |
|---|---|---|---|---|---|---|---|---|---|
| cube-triple-100M | $(0.93, 0.8)$ | $(0.93, 0.8)$ | $(0.5, 0.8)$ | $0.93$ | $0.5$ | $0.5$ | $0.93$ | $0.93$ | $0.5$ |
| cube-quadruple-100M | $(0.93, 0.8)$ | $(0.93, 0.8)$ | $(0.5, 0.8)$ | $0.93$ | $0.5$ | $0.7$ | $0.93$ | $0.93$ | $0.7$ |
| cube-octuple-1B | $(0.93, 0.5)$ | $(0.93, 0.5)$ | $(0.93, 0.5)$ | $0.93$ | $0.97$ | $0.5$ | $0.93$ | $0.93$ | $0.7$ |
| humanoidmaze-giant | $(0.5, 0.8)$ | $(0.5, 0.8)$ | $(0.5, 0.5)$ | $0.5$ | $0.5$ | $0.5$ | $0.5$ | $0.5$ | $0.5$ |
| puzzle-4x5 | $(0.9, 0.5)$ | $(0.9, 0.5)$ | $(0.5, 0.5)$ | $0.7$ | $0.7$ | $0.5$ | $0.9$ | $0.9$ | $0.7$ |
| puzzle-4x6-1B | $(0.7, 0.5)$ | $(0.7, 0.5)$ | $(0.5, 0.5)$ | $0.5$ | $0.7$ | $0.5$ | $0.7$ | $0.7$ | $0.7$ |

Table 7: **Environment-specific hyperparameters under different $h, n, h_a$ configurations for DQC, QC, NS, OS.** For DQC-naïve, we use the same hyperparameter as the corresponding QC baseline.

from an *unconditional* (*i.e.*, not goal-conditioned) behavior policy $\pi_\beta$. In particular, we use an unconditioned flow policy $\pi_\beta(\cdot \mid s)$ that is parameterized by a velocity field $v_\beta : \mathcal{S} \times \mathbb{R}^A \times [0, 1] \to \mathbb{R}^A$ that is trained with the standard flow-matching objective:

$$L_{\mathrm{FM}}(\beta) = \mathbb{E}_{u \sim \mathcal{U}[0,1], z \sim \mathcal{N}, (s,a) \sim \mathcal{D}} \left[ \| v_\beta(s, (1 - u)z + ua, u) - a + z \|_2^2 \right] \tag{29}$$

For SHARSA, we use the official implementation where both flow policies (high-level and low-level) are goal-conditioned (and thus are trained with the goal distribution mixture specified by $w_{\mathrm{cur}}^{\mathrm{P}}, w_{\mathrm{geom}}^{\mathrm{P}}, w_{\mathrm{traj}}^{\mathrm{P}}, w_{\mathrm{rand}}^{\mathrm{P}}$). The goal sampling distribution for training the value networks (for all methods) and the goal sampling distribution for the policy networks (for SHARSA only) are provided in Table 4.

| Environment | Backup Quantile $(\kappa_b)$ | Distillation Expectile $(\kappa_d)$ | Backup Horizon $(h)$ or $(n)$ | Policy Chunk Size $(h_a)$ |
|---|---|---|---|---|
| cube-* | $\{0.5, 0.7, 0.9, 0.93, 0.95, 0.97, 0.99\}$ | $\{0.5, 0.8\}$ | $\{5, 25\}$ | $\{1, 5, 25\}$ |
| Others | $\{0.5, 0.7, 0.9\}$ | $\{0.5, 0.8\}$ | $\{5, 25\}$ | $\{1, 5, 25\}$ |

Table 8: **Hyperparameter tuning range for all methods**. For NS, we only tune $\kappa_b$ and $n$ because the policy chunk size is always 1 and there is no distilled critic. Similarly, for QC, we only tune $\kappa_b$ and $h = h_a$ because the policy chunk size is the same as the critic chunk size and there is no distilled critic. For OS, we only tune $\kappa_b$.

## E  ADDITIONAL RELATED WORK

**Theory of Action Chunking** Existing analyses for action chunking focus exclusively on the imitation learning setting (Tu et al., 2022; Simchowitz et al., 2025). While they laid out the theoretical foundation of action chunking policies for imitation learning, formal guarantees of action chunking RL are still an open problem. In the adjacent field of stochastic optimal control (SOC), action chunking is related to control under *intermittent observations* where the observation inputs to the controller are either unreliable (*e.g.*, with a Poissonian model (Wang, 2001; Dupuis & Wang, 2002)), or partially missing (Mishra et al., 2020; Yan et al., 2022; Noba & Yamazaki, 2022; Bayer et al., 2024). While conceptually related, these analyses are in the continuous-time setting in contrast to discrete-time transitions. To the best of our knowledge, we are the first to provide a formal analysis of action chunking in Q-learning. In particular, we identify the key open-loop consistency condition under which we quantify the exact worst-case Q-learning sub-optimality.

**Theoretical analysis for reinforcement learning under unobserved confounding variables.** RL with action chunking policies can be seen as a special case of RL under unobserved confounding variables (Kallus & Zhou, 2021) as the action chunking policies ignore the intermediate states during the execution of an action chunk. Prior analyses are based off either causal-inference-inspired sensitivity models (Kallus & Zhou, 2020; Namkoong et al., 2020; Kausik et al., 2024), confounded MDP models (Bennett et al., 2021; Fu et al., 2022; Shi et al., 2024), or more general partially observable MDP (POMDP) models (Tennenholtz et al., 2020; Miao et al., 2022; Shi et al., 2022; Bennett & Kallus, 2024) where the confounding variables are modeled as part of the partially observable states. These models largely focus on characterizing either how much confounding variables affect the policy behavior (*e.g.*, bounded odds-ratio between the policy with or without conditioning on the confounding variables (Kallus & Zhou, 2020)) or how much the observations reveal the confounding variables (*e.g.*, the full-rank emission matrix assumption (Azizzadenesheli et al., 2016) and the weak revealing assumption (Liu et al., 2022) in POMDP). In contrast, our analysis specializes in action chunking policies where the unobserved variables are the intermediate states during an action chunk. This allows us to establish a more specialized (and thus distinct) open-loop consistency condition under which we can identify the exact worst case bias (*i.e.*, with matching lower and upper-bound to the exact value) for both behavioral value estimation and sub-optimality gap of the fixed-point for bellman optimality iteration, which are usually unknown under the more general models/assumptions in the literature.

**Hierarchical reinforcement learning methods** (Dayan & Hinton, 1992; Dietterich, 2000; Peng et al., 2017; Riedmiller et al., 2018; Shankar & Gupta, 2020; Pertsch et al., 2021; Gehring et al., 2021; Xie et al., 2021) solve tasks by typically leveraging a bi-level structure: a set of low-level/skill policies that directly interact with the environment and a high-level policy that selects among low-level policies. The low-level policies can also be learned via online RL (Kulkarni et al., 2016; Vezhnevets et al., 2016; 2017; Nachum et al., 2018) or offline pre-training on a prior dataset (Paraschos et al., 2013; Merel et al., 2019; Ajay et al., 2021; Pertsch et al., 2021; Touati et al., 2022; Nasiriany et al., 2022; Hu et al., 2023; Frans et al., 2024; Chen et al., 2024; Park et al., 2024). In the options framework, these low-level policies are often additionally associated with initiation and termination conditions that specify when and for how long these actions can be used (Sutton et al., 1999; Menache et al., 2002; Chentanez et al., 2004; Şimşek & Barto, 2007; Konidaris, 2011; Daniel et al., 2016; Srinivas et al., 2016; Fox et al., 2017; Bacon et al., 2017; Bagaria & Konidaris, 2019; Bagaria et al., 2024; Koch et al., 2025). A long-lasting challenge in HRL is optimization stability because the high-level policy needs to optimize for an objective that is shaped by the constantly changing low-level policies (Nachum et al., 2018). Prior work (Ajay et al., 2021; Pertsch et al., 2021; Wilcoxson et al.,

2025) avoided this by first pre-training low-level policies and then keeping them frozen during the optimization of the high-level policy. Macro-actions (McGovern & Sutton, 1998; Durugkar et al., 2016), or action chunking (Zhao et al., 2023) is another form of temporally extended action, a special case of the low-level policies often considered in HRL, options literature, where a short horizon of actions is predicted all at once and executed in open loop. Such an approach collapses the bi-level structure, conveniently side-stepping optimization instability, and when combined with Q-learning, has shown great empirical successes in offline-to-online RL setting (Seo et al., 2024; Li et al., 2025b). Action chunking policies need to predict multiple actions open-loop, which can be difficult to learn and sacrifice reactivity. Our approach regains policy reactivity by predicting and executing only a partial action chunk, while still learning with the fully chunked critic for TD-backup. This design preserves the value propagation benefits of chunked critic without relying on fully open-loop action chunking policies, allowing our approach to work well on a wider range of tasks.

# F    WORST-CASE ANALYSIS

## F.1    WORST-CASE ANALYSIS OF THEOREM 1

**Theorem 4** (Worst-case AC Value Bias)    For any $h > 1, \gamma \in [0, 1), \varepsilon_h \in [0, 1/2]$, there exists an MDP $\mathcal{M}$ and a weakly $\varepsilon_h$-open-loop consistent $\mathcal{D}$ such that for some $s_t \in \text{supp}(P_{\mathcal{D}}(s_t))$,

$$V_{\text{ac}}(s_t) - \hat{V}_{\text{ac}}(s_t) = \frac{\gamma \varepsilon_h}{(1 - \gamma)(1 - (1 - \varepsilon_h)\gamma^h)}. \tag{30}$$

Similarly, there exists $\mathcal{M}$ and $\varepsilon_h$-open-loop consistent $\mathcal{D}$ such that for some $s_t \in \text{supp}(P_{\mathcal{D}}(s_t))$,

$$\hat{V}_{\text{ac}}(s_t) - V_{\text{ac}}(s_t) = \frac{\gamma \varepsilon_h}{(1 - \gamma)(1 - (1 - \varepsilon_h)\gamma^h)}. \tag{31}$$

Proof is available in Section H.3.

## F.2    WORST-CASE ANALYSIS OF COROLLARY 1

**Corollary 2** (Worse-case Optimality Gap for Action Chunking Policy)    For any $h > 1, \gamma \in [0, 1), \varepsilon_h \in [0, 1/2]$, there exists an MDP $\mathcal{M}$ whose optimal policy $\pi^\star$ induces a data distribution $\mathcal{D}^\star$ that is weakly $\varepsilon_h$-open-loop consistent, such that for some $s_t \in \text{supp}(P_{\mathcal{D}^\star}(s_t))$,

$$V^\star(s_t) - V_{\text{ac}}^\star(s_t) = \frac{\gamma \varepsilon_h}{(1 - \gamma)(1 - (1 - \varepsilon_h)\gamma^h)}. \tag{32}$$

Proof is available in Section H.5.

## F.3    WORST-CASE ANALYSIS OF ACTION CHUNKING Q-LEARNING UNDER WEAK OLC

**Proposition 2** (AC Q-Learning under Weak OLC)    For any $h > 1, \gamma \in [0, 1), c \in [0, 1/2)$, $\varepsilon_h \in (0, 1/2)$, there exists an MDP $\mathcal{M}$, a weakly $\varepsilon_h$-open-loop consistent $\mathcal{D}$ and $\mathcal{D}^\star$ with $\text{supp}(P_{\mathcal{D}}(s_t, a_{t:t+h})) \supseteq \text{supp}(P_{\mathcal{D}^\star}(s_t, a_{t:t+h}))$, such that for some $s_t \in \text{supp}(P_{\mathcal{D}^\star}(s_t))$,

$$V^\star(s_t) - V_{\text{ac}}^+(s_t) = V_{\text{ac}}^\star(s_t) - V_{\text{ac}}^+(s_t) = \frac{\gamma c}{1 - \gamma}. \tag{33}$$

Proof is available in Section H.6.

## F.4    WORST-CASE ANALYSIS OF THEOREM 2

**Theorem 5** (Worst-case Analysis of Q-Learning with Action Chunking Policy on Off-policy Data)    For any $h > 1, \gamma \in (0, 1), \varepsilon_h \in (0, 1/5), c_1 \in (0, \varepsilon_h/2)$, and $c_2 \in (0, 2\varepsilon_h\gamma)$, there exists an MDP $\mathcal{M}$ and strongly $\varepsilon_h$-open-loop consistent data distributions $\mathcal{D}$ and $\mathcal{D}^\star$ with $\text{supp}(P_{\mathcal{D}}(s_t, a_{t:t+h})) \supseteq \text{supp}(P_{\mathcal{D}^\star}(s_t, a_{t:t+h}))$, such that for some $s_t \in \text{supp}(P_{\mathcal{D}^\star}(s_t))$,

$$V^\star(s_t) - V_{\text{ac}}^+(s_t) = \frac{2\varepsilon_h\gamma - c_2}{(1 - \gamma)(1 - (1 - 2\varepsilon_h)\gamma^h)} + \frac{\varepsilon_h\gamma}{(1 - \gamma)(1 - (1 - \varepsilon_h - c_1)\gamma^h)}, \tag{34}$$

where $V^\star$ is the value of an optimal policy and $V_{\text{ac}}^+$ is the *true* value of $\pi_{\text{ac}}^+$. As $c_1, c_2 \to 0$,

$$V^\star(s_t) - V_{\text{ac}}^+(s_t) \to \frac{\varepsilon_h\gamma}{1 - \gamma} \left[ \frac{2}{1 - (1 - 2\varepsilon_h)\gamma^h} + \frac{1}{1 - (1 - \varepsilon_h)\gamma^h} \right]. \tag{35}$$

Proof is available in Section H.8.

## F.5    WORST-CASE ANALYSIS OF THEOREM 3

It is worth noting that although the global optimality variability condition looks similar to the strong open-loop consistency condition, they have completely different properties. For instance, a nearly

strong open-loop consistent data distribution $\mathcal{D}$ can have unbounded global optimality variability and a data distribution that exhibits zero optimality variability can also have large open-loop inconsistency. The implication of this is that while the closed-loop execution of an action chunking policy can be near-optimal, the same action chunking policy executed in chunks can be sub-optimal. We formalize this intuition as the worse-case result below:

---

**Theorem 6** (Worst-case Closed-loop AC Policy under BOV)  For any $h > 1, \gamma \in (0,1), \vartheta_h^G, \vartheta_h^L \in \left(0, \frac{\gamma - \gamma^h}{4(1-\gamma)}\right], c \in \left[0, \frac{\gamma - \gamma^h}{4(1-\gamma^h)}\right), \sigma \in \left(0, \frac{\min(\vartheta_h^G, \vartheta_h^L)}{1-\gamma}\right)$, there exists $\mathcal{M}$ and $\mathcal{D}$ satisfying the assumptions in Theorem 3 such that there exists $s_t \in \text{supp}(P_{\mathcal{D}^\star}(s_t))$, where

$$V^\star(s_t) - V^\bullet(s_t) = \frac{\vartheta_h^L}{1-\gamma} + \frac{\vartheta_h^G + \gamma^h \min(\vartheta_h^L, \vartheta_h^G)}{(1-\gamma)(1-\gamma^h)} - \sigma, V^\star(s_t) - V_{\text{ac}}^+(s_t) \geq \frac{c}{1-\gamma}. \quad (36)$$

---

The examples in the proof of Theorem 6 (available in Section H.13) serve as a dual purpose—they not only show that our upper-bound in Theorem 3 is *tight* (since we can make $\sigma \to 0$), but also show that the sub-optimality of the action chunking policy can be made arbitrarily large. Furthermore, *both* the local optimality ($\vartheta_h^L$) condition and the global optimality ($\vartheta_h^G$) are *necessary* to guarantee $\pi^\bullet$ being near-optimal. When any of them is large, Theorem 6 implies that there exists an MDP where $\pi^\bullet$ is sub-optimal. As a side note, we can guarantee $\pi^\bullet$ to be near-optimal with an alternative 'stochastic shortcut' assumption (a weaker form of the global optimality variability assumption) and a slightly stronger data mixing assumption. We refer the readers to Section G.4 for the formal results under this alternative assumption.

# G ADDITIONAL THEORETICAL RESULTS

## G.1 COMPARING TO $n$-STEP RETURN Q-LEARNING

We now characterize the condition when action chunking Q-learning should be preferred over the standard $n$-step return backup. We start by introducing a notion of sub-optimality:

---

**Definition 4** (Sub-optimal Data)   $\mathcal{D}$ is $\delta_n$-sub-optimal for a backup horizon length of $n > 1$ if

$$Q^\star(s_t, a_t) - \mathbb{E}_{P_\mathcal{D}(\cdot|s_t, a_t)}[R_{t:t+n} + \gamma^n V^\star(s_{t+n})] \geq \delta_n, \forall s_t, a_t \in \mathrm{supp}(P_\mathcal{D}(s_t, a_t)). \quad (37)$$

---

Intuitively, $\delta_n$ captures how much worse the $n$-step return policy can get compared to the optimal policy incurred by the backup bias. Under such condition, we can show that the action chunking policy is provably better than the $n$-step return policy as long as $\delta_n$ is large.

---

**Proposition 3** (Comparing action chunking backup and $n$-step return backup)   Let $\mathcal{D}$ be strongly $\varepsilon_h$-open-loop consistent and $\delta_n$-sub-optimal, and $\mathrm{supp}(P_\mathcal{D}(s_t)) \supseteq \mathrm{supp}(P_{\mathcal{D}^\star}(s_t))$. Let $\pi_n^+ : s_t \mapsto \arg\max_{a_t} \hat{Q}_n^+(s_t, a_t)$ be the policy learned from $\mathcal{D}$, via $n$-step return backup:

$$\hat{Q}_n^+(s_t, a_t) = \mathbb{E}\left[R_{t:t+n} + \gamma^n \hat{Q}_n^+(s_{t+n}, \pi_n^+(s_{t+n}))\right]. \quad (38)$$

Then, for all $s_t \in \mathrm{supp}(P_{\mathcal{D}^\star}(s_t))$ (and with $\bar{H}_n = 1/(1 - \gamma^n)$),

$$V_{\mathrm{ac}}^+(s_t) - \hat{V}_n^+(s_t) \geq \frac{\delta_n}{1 - \gamma^n} - \frac{\varepsilon_h \gamma}{1 - \gamma}\left[\frac{2}{1 - (1 - 2\varepsilon_h)\gamma^h} + \frac{1}{1 - (1 - \varepsilon_h)\gamma^h}\right], \quad (39)$$

$$\geq \delta_n \bar{H}_n - 3\varepsilon_h H \bar{H}.$$

---

The proof of Proposition 3 is available in Section H.10. Notably, for $n = h$, as long as $\mathcal{D}$ is more than $(3\varepsilon_h H)$-sub-optimal, the value of the action chunking policy is provably better than the value of the $n$-step return policy. It is worth noting that Proposition 3 uses the *nominal* value of the $n$-step return, which may be lower than its *actual* value. We refer the readers to Section G.3 for examples where the $n$-step return policy is provably worse than the action chunking policy.

## G.2 $\varepsilon$-DETERMINISTIC DYNAMICS IS WEAKLY OPEN-LOOP CONSISTENT

To provide some intuitions on what this open-loop consistency implies, we discuss a concrete family of MDPs where any data distribution from these MDPs is (weakly) $\varepsilon_h$-open-loop consistent (Proposition 4, with proof available in Section H.15).

---

**Definition 5** (Near-deterministic Dynamics)   A transition dynamics $T$ is $\varepsilon$-deterministic if there exists a deterministic transition dynamics represented by function $f : \mathcal{S} \times \mathcal{A} \to \mathcal{S}$ and another arbitrary transition dynamics $\tilde{T} : \mathcal{S} \times \mathcal{A} \to \Delta_\mathcal{S}$, and $T$ is a combination of $f$ and $\tilde{T}$:

$$T(s' \mid s, a) = (1 - \varepsilon)\delta_{f(s,a)}(s') + \varepsilon \tilde{T}(s' \mid s, a), \forall s, s' \in \mathcal{S}, a \in \mathcal{A}. \quad (40)$$

---

**Proposition 4** (Deterministic Dynamics are Weakly Open-loop Consistent)   If a transition dynamics $\mathcal{M}$ is $\varepsilon$-deterministic, then any data $\mathcal{D}$ collected from $\mathcal{M}$ is weakly $\varepsilon_h$-open-loop consistent with respect to $\mathcal{M}$ for any $h \in \mathbb{N}^+$ as long as $\varepsilon_h \geq 3(1 - (1 - \varepsilon)^{h-1})$.

---

An $\varepsilon$-deterministic dynamics acts like a deterministic one most of the time (with $1 - \varepsilon$ probability) and a non-deterministic one occasionally (with $\varepsilon$ probability). This bounded stochasticity allows the results of taking an action sequence (of length $h$) open-loop to be deterministically determined in the event that the deterministic dynamics is 'triggered' (with a joint $(1 - \varepsilon)^{h-1}$ probability across $h$ time steps). It is clear that under such event, there is no gap between the 'replayed' open-loop data $P_\mathcal{D}^\circ$ and the original data distribution $P_\mathcal{D}$, and as result there is also no value estimation bias under this event, and thus intuitively we can bound the value estimation error by a function of the probability that the stochastic dynamics is 'triggered' (*i.e.*, with $1 - (1 - \varepsilon)^{h-1}$ probability).

### G.3 CONDITIONS WHEN $n$-STEP RETURN POLICIES ARE PROVABLY SUB-OPTIMAL

**Definition 6** (Near Optimal Data)  We say $\mathcal{D}$ is $\tilde{\delta}_n$-optimal for backup horizon length $n \in \mathbb{N}^+$ if

$$Q^{\star}(s_t, a_t) - \mathbb{E}_{P_{\mathcal{D}}(\cdot|s_t,a_t)} \left[ R_{t:t+n} + \gamma^n V^{\star}(s_{t+n}) \right] \leq \tilde{\delta}_n, \forall s_t, a_t \in \operatorname{supp}(P_{\mathcal{D}}(s_t, a_t)). \quad (41)$$

In Proposition 3, we have shown that the value of the learned action chunking policy is better than the nominal value of $n$-step return policy with a value gap of $3\varepsilon_h H$. However, the actual value of the $n$-step return policy maybe better. Here, we analyze the worst-case performance of $n$-step return policies.

**Proposition 5** (Worst-case analysis of $n$-step return backup)  For any $n \in \mathbb{N}^+$, $\tilde{\delta}_n \in (0, \gamma - \gamma^n)$ and $\sigma \in \left(0, \tilde{\delta}_n/(1-\gamma)\right)$, there exists an MDP $\mathcal{M}$, and a $\tilde{\delta}_n$-optimal data distribution $\mathcal{D}$ with $\operatorname{supp}(P_{\mathcal{D}}(s_t, a_t)) \supseteq \operatorname{supp}(P_{\mathcal{D}^{\star}}(s_t, a_t))$ such that for some $s \in \operatorname{supp}(P_{\mathcal{D}^{\star}}(s_t))$,

$$V_{\mathrm{ac}}^+(s) - V_n^+(s) = \frac{\tilde{\delta}_n}{1 - \gamma} - \sigma, \quad (42)$$

and for all $s \in \operatorname{supp}(P_{\mathcal{D}^{\star}}(s_t))$,

$$V^{\star}(s) = V_{\mathrm{ac}}^+(s). \quad (43)$$

The proof (available in Section H.14) provides concrete examples where $n$-step return policies are worse than action chunking policies. The implication of this result is that the sub-optimality of the data distribution (as characterized by $\delta_n$ and $\tilde{\delta}_n$) is generally independent from the open-loop consistency (as characterized by $\varepsilon_h$).

### G.4 CLOSED-LOOP EXECUTION WITHOUT STOCHASTIC SHORTCUTS

In this section, we provide an alternative way of bounding the sub-optimality of $\pi^{\bullet}$, the closed-loop execution of the learned action chunking policy $\pi_{\mathrm{ac}}^+$. In particular, we characterize two conditions when closed-loop execution of an action chunking policy can help mitigate open-loop biases.

Our first condition is based on the key observation that only a certain type of value overestimation is harmful for closed-loop execution of the action chunking policy. The source of this type of value overestimation comes from *stochastic shortcuts*:

**Definition 7** (Stochastic Shortcuts)  We say $\mathcal{M}$ is free of $\vartheta_h$-stochastic shortcuts for a horizon $h$ if

$$V^{\star}(s_{t+h}) + R_{t:t+h} - V^{\star}(s_t) \leq \vartheta_h,$$

$$\forall s_{t:t+h+1}, a_{t:t+h} : \prod_{k=0}^{h-1} P(s_{t+k+1} \mid s_{t+k}, a_{t+k}) > 0, \quad (44)$$

where $V^{\star}$ is the value function of optimal policy in $\mathcal{M}$.

Intuitively, stochastic shortcuts are low-probability (but plausible) paths (*i.e.*, $s_t, a_t, \cdots, s_{t+h}$) in the MDP that lead to returns that are much higher than the optimal expected value (*i.e.*, $V^{\star}$). These stochastic shortcuts are particularly problematic for action chunking value backup because the chunked critic/Q-function cannot distinguish between a low-probability stochastic shortcut and an optimal (or near-optimal) closed-loop trajectory, leading it to erroneously favor the shortcut.

Our second condition is that our data distribution is a mixture of some data distribution that is collected by some optimal closed-loop policy ($\mathcal{D}^{\star}$) and some data distribution that is collected by an open-loop policy ($\mathcal{D}^{\circ}$, and thus is open-loop consistent). Intuitively, this condition makes sure that any non-optimal trajectory can be accurately estimated by the action chunking value function $\hat{V}_{\mathrm{ac}}^+$ and the bounded mixing ratio restricts the amount of bias that the $\hat{V}_{\mathrm{ac}}^+$ has on the estimation of the optimal trajectories when the open-loop action chunks (*e.g.*, in $\mathcal{D}^{\circ}$) coincide with the action chunks in the optimal data (*e.g.*, in $\mathcal{D}^{\star}$). We formally define the second condition as follows:

**Definition 8** (Open-loop Data Mix)    We say $\mathcal{D}$ is $\alpha$-open-loop mixed if for some $\beta \in [0, 1)$, $\mathcal{D}$ can be decomposed into two data distributions $\mathcal{D}^\star, \mathcal{D}^\circ$ as

$$P_{\mathcal{D}}(\cdot \mid s_t) = \beta P_{\mathcal{D}^\star}(\cdot \mid s_t) + (1 - \beta) P_{\mathcal{D}^\circ}(\cdot \mid s_t), \tag{45}$$

where $\mathcal{D}^\star$ is any data distribution collected by an optimal closed-loop policy $\pi^\star$ and $\mathcal{D}^\circ$ is any strongly open-loop consistent data distribution, and

$$P_{\mathcal{D}^\circ} \left[ a_{t:t+h} \in \mathrm{supp}(P_{\mathcal{D}^\star}(a_{t:t+h} \mid s_t)) \mid s_t \right] \leq \frac{\alpha\beta}{(1 - \alpha)(1 - \beta)}, \quad \forall s_t \tag{46}$$

With such data mixing assumption and in the absence of stochastic shortcuts, we can show that closed-loop execution of the action chunking policy (*i.e.*, only executing the first action of the action chunk) recovers a near-optimal closed-loop policy:

**Theorem 7** (Closed-loop Execution in the Absence of Stochastic Shortcuts)    $\mathcal{D}$ is $\alpha$-open-loop mixed and $\mathcal{M}$ is free of $\vartheta_h$-stochastic shortcut, the value ($V^\bullet$) of the one-step policy ($\pi^\bullet$) as a result of the closed-loop execution of the action chunking policy $\pi_{\mathrm{ac}}^+$ learned from $\mathcal{D}$ admits the following bound for all $s_t \in \mathrm{supp}(P_{\mathcal{D}^\star}(s_t))$:

$$V^\star(s_t) - V^\bullet(s_t) \leq \frac{\alpha}{(1 - \gamma)^2 (1 - \gamma^h (1 - \alpha))} + \frac{\vartheta_h \gamma^h}{(1 - \gamma)(1 - \gamma^h)}. \tag{47}$$

A proof is available in Section H.11. Intuitively, the second condition measures how much percentage of the open-loop data has overlapping support as the optimal data. With some algebraic manipulating, assuming the worst case of Equation (46), we can rewrite the data mixture as

$$\mathcal{D} = \hat{\beta} \left[ (1 - \alpha)\mathcal{D}^\star + \alpha \mathcal{D}_{\mathrm{in}}^\circ \right] + (1 - \hat{\beta}) \mathcal{D}_{\mathrm{out}}^\circ, \tag{48}$$

where $\hat{\beta} = \frac{\beta}{1-\alpha}$, $\mathrm{supp}(P_{\mathcal{D}_{\mathrm{in}}^\circ}(\cdot \mid s_t)) \subseteq \mathrm{supp}(P_{\mathcal{D}^\star}(\cdot \mid s_t))$ and $\mathrm{supp}(P_{\mathcal{D}_{\mathrm{out}}^\circ}(\cdot \mid s_t)) \cap \mathrm{supp}(P_{\mathcal{D}^\star}(\cdot \mid s_t)) = \varnothing$. As the bound is independent of $\hat{\beta}$, it becomes clear that $\mathcal{D}_{\mathrm{out}}^\circ$ plays no contribution to the optimality of action chunking policy learning. The only harmful portion of the open-loop data distribution is $\mathcal{D}_{\mathrm{in}}^\circ$, as the action chunking Q-function cannot differentiate these open-loop actions in $\mathcal{D}_{\mathrm{in}}^\circ$ from the closed-loop optimal actions in $\mathcal{D}^\star$. This is reflected as the first term in our bound. The implication is that even if the data $\mathcal{D}$ is arbitrarily sub-optimal (with $\hat{\beta} \to 0$, and hence arbitrarily bad for $n$-step return policies), $\pi^\bullet$ remains near-optimal as long as the 'in-distribution' open-loop data $\mathcal{D}_{\mathrm{in}}^\circ$ is relatively low in density compared to the optimal closed-loop data $\mathcal{D}^\star$ (*i.e.*, $\alpha$ is small).

Furthermore, our bound is independent of the open-loop consistency of the data $\mathcal{D}$. As $\alpha, \vartheta \to 0$, closed-loop execution of the action chunking policy exactly recovers the optimal policy. In contrast, even when $\alpha, \vartheta \to 0$, open-loop execution of the original action chunking policy (*i.e.*, $\pi_{\mathrm{ac}}^+$) can suffer from the open-loop inconsistency of the data $\mathcal{D}$: its value error can only be bounded by $\frac{\varepsilon_h}{(1-\gamma)(1-\gamma^h)}$ (as shown in Theorem 1), a function of $\varepsilon_h$ (the strong open-loop consistency of $\mathcal{D}$).

# H  PROOFS OF MAIN RESULTS

## H.1  UTILITY LEMMATA

---

**Lemma 1** (Mean value theorem for conditional probabilities)  Let $P_1, P_2 \in \Delta_{\mathcal{X} \times \mathcal{Y}}$ and $P(x, y) := \hat{\alpha}(y) P_1(x, y) + (1 - \hat{\alpha}(y)) P_2(x, y)$ and there exists $\alpha > 0$ such that $\hat{\alpha}(y) \leq \alpha, \forall y \in \mathcal{Y}$. Then, there exists $y \in \mathcal{Y}$ and $\tilde{\alpha} \leq \alpha$ such that

$$P(\cdot \mid y) = \tilde{\alpha} P_1(\cdot \mid y) + (1 - \tilde{\alpha}) P_2(\cdot \mid y) \tag{49}$$

---

*Proof.*

$$\frac{P(x, y)}{P(y)} = \frac{\hat{\alpha}(y) P_1(y) P_1(x \mid y) + (1 - \hat{\alpha}(y)) P_2(x \mid y)}{\hat{\alpha}(y) P_1(y) + (1 - \hat{\alpha}(y)) P_2(y)} \tag{50}$$
$$= \beta(y) P_1(x \mid y) + (1 - \beta(y)) P_2(x \mid y)$$

where $\beta(y) := \frac{\hat{\alpha}(y) P_1(y)}{\hat{\alpha}(y) P_1(y) + (1 - \hat{\alpha}(y)) P_2(y)}$. We now prove $\exists y \in \mathcal{Y}, \tilde{\alpha} \leq \alpha$ for Equation (49) to hold by contradiction.

We first assume $\tilde{\alpha} = \beta(y) > \alpha, \forall y \in \mathcal{Y}$. Now, substitute $\beta(y)$ in and integrate both side by $y$ to obtain

$$\hat{\alpha}(y) P_1(y) > \alpha \hat{\alpha}(y) P_1(y) + \alpha (1 - \hat{\alpha}(y)) P_2(y) \tag{51}$$
$$\hat{\alpha}(y) > \alpha \hat{\alpha}(y) + \alpha - \alpha \hat{\alpha}(y) = \alpha, \tag{52}$$

which is a contradiction to the condition $\hat{\alpha}(y) \leq \alpha$.

Therefore, there must exist $y \in \mathcal{Y}$ with $\tilde{\alpha} \leq \alpha$ such that Equation (49) holds. $\qquad \square$

---

**Lemma 2** (Expectation difference for bounded function and TV)  For two distributions $P, Q \in \Delta_{\mathcal{X}}$ and two bounded functions $f, g : \mathcal{X} \to [0, 1]$, if the TV distance between $P$ and $Q$ is no larger than $\varepsilon$ and $\|f - g\|_\infty \leq \delta$ under $\text{supp}(P) \cap \text{supp}(Q)$, then

$$|\mathbb{E}_{x \sim P}[f(x)] - \mathbb{E}_{x \sim Q}[g(x)]| \leq (1 - \varepsilon)\delta + \varepsilon. \tag{53}$$

---

*Proof.*  Let's decompose the probability mass of $P$ and $Q$ in terms of $d_P, d_{PQ}, d_Q : \mathcal{X} \to \mathbb{R}$ as the following:

$$P(x) = d_P(x) + d_{PQ}(x), \tag{54}$$
$$Q(x) = d_{PQ}(x) + d_Q(x). \tag{55}$$

The $\int d_P(x) \mathrm{d}x$ maximizing solution is

$$d_P(x) = \max(P(x), Q(x)) - Q(x) \tag{56}$$
$$d_Q(x) = \max(P(x), Q(x)) - P(x) \tag{57}$$
$$d_{PQ}(x) = P(x) + Q(x) - \max(P(x), Q(x)). \tag{58}$$

It is clear that under this decomposition,

$$\int d_P(x) \mathrm{d}x = \int d_Q(x) \mathrm{d}x = \hat{\varepsilon} \leq \varepsilon, \tag{59}$$

$$\int d_{PQ}(x) \mathrm{d}x = 1 - \hat{\varepsilon} \geq 1 - \varepsilon. \tag{60}$$

Now we are ready to bound the expectation difference:

$$|\mathbb{E}_{x \sim P}[f(x)] - \mathbb{E}_{x \sim Q}[g(x)]|$$

$$= \left| \left( \int d_P(x)f(x)\mathrm{d}x - \int d_Q(x)g(x)\mathrm{d}x \right) + \left( \int d_{PQ}(x)(f(x) - g(x))\mathrm{d}x \right) \right|$$

$$\leq \left| \int d_P(x)f(x)\mathrm{d}x - \int d_Q(x)g(x)\mathrm{d}x \right| + \left| \int d_{PQ}(x)(f(x) - g(x))\mathrm{d}x \right|$$

$$\leq \max \left( \sup_x f(x) \int d_P(x)\mathrm{d}x - \inf_x g(x) \int d_Q(x)\mathrm{d}x, \sup_x g(x) \int d_Q(x)\mathrm{d}x - \inf_x f(x) \int d_P(x)\mathrm{d}x \right)$$

$$+ \left| \left( \sup_{x:d_{PQ}(x)>0} |f(x) - g(x)| \right) \int d_{PQ}(x)\mathrm{d}x \right|$$

$$\leq \hat{\varepsilon} + \left( \sup_{x \in \mathrm{supp}(P) \cap \mathrm{supp}(Q)} |f(x) - g(x)| \right)(1 - \hat{\varepsilon})$$

$$= \hat{\varepsilon} + \|f - g\|_\infty(1 - \hat{\varepsilon})$$

$$\leq \hat{\varepsilon}(1 - \delta) + \delta$$

$$\leq \varepsilon(1 - \delta) + \delta$$

$$= (1 - \varepsilon)\delta + \varepsilon \tag{61}$$

as desired. $\qquad\square$

---

**Lemma 3** (Total variation under event conditioning)  For two random variables $X \in \Delta_{\mathcal{X}}$ and $Y \in \Delta_{\mathcal{Y}}$ and any $y \in \mathcal{Y}$,

$$D_{\mathrm{TV}}(P(X \mid Y = y) \| P(X)) \leq 1 - P(Y = y) \tag{62}$$

---

*Proof.* Let $p = P(Y = y)$

$$D_{\mathrm{TV}}(P(X \mid Y = y) \| P(X))$$

$$= \frac{1}{2} \int |P(x) - P(x \mid y)|\mathrm{d}x$$

$$= \frac{1}{2} \int |P(x \mid Y = y)(P(Y = y) - 1) + P(x \mid Y \neq y)P(Y \neq y)|\mathrm{d}x \tag{63}$$

$$= \frac{1 - p}{2} \int |(P(x \mid Y \neq y) - P(x \mid Y = y))|\mathrm{d}x$$

$$= (1 - p)D_{\mathrm{TV}}(P(X \mid Y = y) \| P(X \mid Y \neq y))$$

$$\leq 1 - p$$

$\qquad\square$

---

**Lemma 4** (Data Processing Inequality for $f$-divergence (Csiszár, 1967))  For two random variables $A, B \in \Delta_{\mathcal{X}}$ and a deterministic function $f : \mathcal{X} \to \mathcal{Y}$, and $C := g(A), D := g(B)$

$$D_f(P_A \| P_B) \geq D_f(P_C \| P_D). \tag{64}$$

Since TV-distance is a $f$-divergence with $f = |x - 1|$, we have

$$D_{\mathrm{TV}}(P_A \| P_B) \geq D_{\mathrm{TV}}(P_C \| P_D). \tag{65}$$

---

*Proof from Wu (2017).*

$$
\begin{aligned}
D_f(P_A \parallel P_B) &= \mathbb{E}_{x \sim P_B} \left[ f(P_A(x)/P_B(x)) \right] \\
&= \mathbb{E}_{P_{BD}} \left[ f(P_{AC}/P_{BD}) \right] \\
&= \mathbb{E}_{(x,y) \sim P_D} \left[ \mathbb{E}_{P_{B|D}} \left[ f(P_{AC}(x,y)/P_{BD}(x,y)) \right] \right] \\
&\geq \mathbb{E}_{y \sim P_D} \left[ f \left( \mathbb{E}_{x \sim P_{B|D=y}} \left[ P_{AC}(x,y)/P_{BD}(x,y) \right] \right) \right] \\
&= \mathbb{E}_{y \sim P_D} \left[ f \left( \mathbb{E}_{x \sim P_{B|D=y}} \left[ P_C(y)/P_D(y) \right] \right) \right] \\
&= \mathbb{E}_{y \sim P_D} \left[ f \left( P_C(y)/P_D(y) \right) \right] \\
&= D_f(P_C \parallel P_D).
\end{aligned}
\tag{66}
$$

$\square$

## H.2 PROOF OF THEOREM 1

**Theorem 1** (AC Value Bias)  Let $\hat{V}_{\text{ac}} : \mathcal{S} \to [0, 1/(1-\gamma)]$ be a solution of

$$\hat{V}_{\text{ac}}(s_t) = \mathbb{E}_{s_{t+1:t+h+1}, a_{t:t+h} \sim P_{\mathcal{D}}(\cdot|s_t)} \left[ R_{t:t+h} + \gamma^h \hat{V}_{\text{ac}}(s_{t+h}) \right], \tag{11}$$

with $R_{t:t+h} = \sum_{t'=t}^{t+h} \gamma^{t'-t} r(s_{t'}, a_{t'})$ and $V_{\text{ac}}$ is the true value of $\tilde{\pi}_{\text{ac}} : s_t \mapsto P_{\mathcal{D}}(a_{t:t+h} \mid s_t)$. If $\mathcal{D}$ is weakly $\varepsilon_h$-open-loop consistent, then for all $s_t \in \text{supp}(P_{\mathcal{D}}(s_t))$,

$$\left| V_{\text{ac}}(s_t) - \hat{V}_{\text{ac}}(s_t) \right| \leq \frac{\gamma \varepsilon_h}{(1-\gamma)(1-(1-\varepsilon_h)\gamma^h)} \leq \varepsilon_h H \bar{H}. \tag{12}$$

*Proof.* Since $\mathcal{D}$ is $\varepsilon_{h'}$-open-loop consistent in state-action for $h' < h$, the state-action distribution leading up to step $h$ admits the following bound:

$$D_{\text{TV}}(P_{\mathcal{D}}(s_{t+h}, a_{t+h} \mid s_t) \parallel P_{\mathcal{D}}^{\circ}(s_{t+h}, a_{t+h} \mid s_t)) \leq \varepsilon_h \tag{67}$$

Let $R_{t:t+h} = \sum_{k=0}^{h-1} \gamma^k r(s_{t+k}, a_{t+k})$ be the $h$-step reward distribution. Then the difference in $h$-step reward is bounded by

$$\left| \mathbb{E}_{P_{\mathcal{D}}(\cdot|s_t)}[R_{t:t+h}] - \mathbb{E}_{P_{\mathcal{D}}^{\circ}(\cdot|s_t)}[R_{t:t+h}] \right|$$

$$\leq \sum_{h'=1}^{h-1} \left[ \gamma^{h'} \mathbb{E}_{P_{\mathcal{D}}(s_{t+h'}, a_{t+h'}|s_t)}[r(s_{t+h'}, a_{t+h'})] - \mathbb{E}_{P_{\mathcal{D}}^{\circ}(s_{t+h'}, a_{t+h'}|s_t)}[r(s_{t+h'}, a_{t+h'})] \right] \tag{68}$$

$$\leq \sum_{h'=1}^{h-1} \gamma^{h'} \varepsilon_h.$$

where the first inequality uses Lemma 2 and the fact that TV distance is bounded (Equation (67)).

Since $\mathcal{D}$ is $\varepsilon_h$-open-loop consistent for $h$ in state, we have

$$D_{\text{TV}}(P_{\mathcal{D}}(s_{t+h} \mid s_t) \parallel P_{\mathcal{D}}^{\circ}(s_{t+h} \mid s_t)) \leq \varepsilon_h, \tag{69}$$

which can then be used to bound the estimation error using Lemma 2:

$$\left| \mathbb{E}_{s_{t+h} \sim P_{\mathcal{D}}(s_{t+h}|s_t)} \left[ \hat{V}_{\text{ac}}(s_{t+h}) \right] - \mathbb{E}_{s_{t+h} \sim P_{\mathcal{D}}^{\circ}(s_{t+h}|s_t)} \left[ V_{\text{ac}}(s_{t+h}) \right] \right|$$

$$\leq \frac{\varepsilon_h}{1-\gamma} + (1-\varepsilon_h) \sup_{s_{t+h} \in \text{supp}(P_{\mathcal{D}}(s_{t+h}|s_t))} \left[ |\hat{V}_{\text{ac}}(s_{t+h}) - V_{\text{ac}}(s_{t+h})| \right] \tag{70}$$

For all $s_t \in \text{supp}(P_{\mathcal{D}}(s_t))$,

$$\left| \hat{V}_{\text{ac}}(s_t) - V_{\text{ac}}(s_t) \right|$$

$$\leq \left| \mathbb{E}_{P_{\mathcal{D}}(\cdot|s_t)}[R_{t:t+h}] - \mathbb{E}_{P_{\mathcal{D}}^{\circ}(\cdot|s_t)}[R_{t:t+h}] \right|$$

$$+ \gamma^h \left| \mathbb{E}_{s_{t+h} \sim P_{\mathcal{D}}(s_{t+h}|s_t)} \left[ \hat{V}_{\text{ac}}(s_{t+h}) \right] - \mathbb{E}_{s_{t+h} \sim P_{\mathcal{D}}^{\circ}(s_{t+h}|s_t)} \left[ V_{\text{ac}}(s_{t+h}) \right] \right| \tag{71}$$

$$\leq \sum_{h'=0}^{h-1} \left[ \gamma^{h'} \varepsilon_h \right] + \frac{\gamma^h \varepsilon_h}{1-\gamma} + \gamma^h (1-\varepsilon_h) \sup_{s_{t+h} \in \text{supp}(P_{\mathcal{D}}(s_{t+h}|s_t))} \left[ |\hat{V}_{\text{ac}}(s_{t+h}) - V_{\text{ac}}(s_{t+h})| \right].$$

Since the support of $s_{t+h} \mid s_t$ is a subset of the support for $s_t$ by Assumption 1, we can recursively apply the inequality to obtain,

$$\left| \hat{V}_{\text{ac}}(s_t) - V_{\text{ac}}(s_t) \right| \leq \frac{1}{1-(1-\varepsilon_h)\gamma^h} \left( \sum_{h'=1}^{h-1} \left[ \gamma^{h'} \varepsilon_h \right] + \frac{\gamma^h \varepsilon_h}{1-\gamma} \right)$$

$$= \frac{\gamma \varepsilon_h}{(1-\gamma)(1-(1-\varepsilon_h)\gamma^h)}, \tag{72}$$

as desired.  $\square$

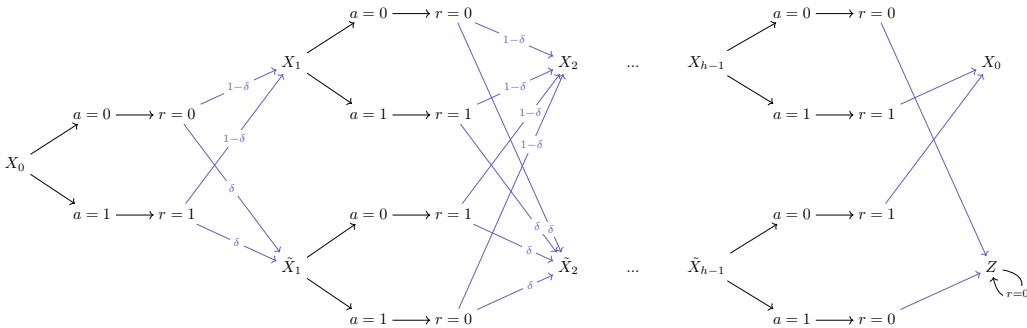

Figure 7: **A $2h$-state MDP that is constructed to meet the upper-bound in Theorem 1.** The data distribution $\mathcal{D}$ that achieves such an upper bound is collected by the optimal policy: $\pi(X_i) = 1, \pi(\tilde{X}_i) = 0$.

### H.3 Proof of Theorem 4

> **Theorem 4** (Worst-case AC Value Bias)  For any $h > 1, \gamma \in [0, 1), \varepsilon_h \in [0, 1/2]$, there exists an MDP $\mathcal{M}$ and a weakly $\varepsilon_h$-open-loop consistent $\mathcal{D}$ such that for some $s_t \in \text{supp}(P_\mathcal{D}(s_t))$,
>
> $$V_{\text{ac}}(s_t) - \hat{V}_{\text{ac}}(s_t) = \frac{\gamma \varepsilon_h}{(1 - \gamma)(1 - (1 - \varepsilon_h)\gamma^h)}. \tag{30}$$
>
> Similarly, there exists $\mathcal{M}$ and $\varepsilon_h$-open-loop consistent $\mathcal{D}$ such that for some $s_t \in \text{supp}(P_\mathcal{D}(s_t))$,
>
> $$\hat{V}_{\text{ac}}(s_t) - V_{\text{ac}}(s_t) = \frac{\gamma \varepsilon_h}{(1 - \gamma)(1 - (1 - \varepsilon_h)\gamma^h)}. \tag{31}$$

*Proof.* Let $\delta \in [0, 1]$ be any value that satisfies $\varepsilon_h = 2\delta(1 - \delta)$. $\delta$ must exist because $\varepsilon_h \in [0, 1/2]$. Let us define a MDP that has $S = 2h$ states, $\mathcal{S} = \{X_0, X_1, \tilde{X}_1, \cdots, X_{h-1}, \tilde{X}_{h-1}, Z\}$, and $A = 2$ actions, $\mathcal{A} = \{0, 1\}$, and the following transition function $T$ and reward function $r$ (see a diagram in Figure 7):

$$
\begin{aligned}
T(\tilde{X}_{i+1} \mid X_i, a) = T(\tilde{X}_{i+1} \mid \tilde{X}_i, a) &= \delta, && \forall a \in \{0, 1\}, i \in \{1, \cdots, h-2\} \\
T(X_{i+1} \mid X_i, a) = T(X_{i+1} \mid \tilde{X}_i, a) &= 1 - \delta, && \forall a \in \{0, 1\}, i \in \{0, \cdots, h-2\} \\
T(Z \mid \tilde{X}_{h-1}, a = 1) = T(Z \mid X_{h-1}, a = 0) &= 1 \\
T(X_0 \mid \tilde{X}_{h-1}, a = 0) = T(X_0 \mid X_{h-1}, a = 1) &= 1 \\
r(\tilde{X}_i, a = 0) = r(X_i, a = 1) &= 1, && \forall i \in \{0, \cdots, h-1\} \\
r(\tilde{X}_i, a = 1) = r(X_i, a = 0) &= 0, && \forall i \in \{0, \cdots, h-1\} \\
r(Z, a = 1) = r(Z, a = 0) &= 0 \\
T(Z \mid Z, a = 0) = T(Z \mid Z, a = 1) &= 1
\end{aligned}
\tag{73}
$$

Now, we assume that the data $\mathcal{D}$ is collected by the optimal closed-loop policy where

$$\pi(X_i) = 1, \pi(\tilde{X}_i) = 0. \tag{74}$$

First, we check $\mathcal{D}$ is $\varepsilon_h$-open-loop consistent.

We can show that by computing the distribution for $P_\mathcal{D}(s_{t+i}, a_{t+i} \mid s_t = X_0)$ and $P_\mathcal{D}^\circ(s_{t+i}, a_{t+i} \mid s_t = X_0)$ as follows:

$$
\begin{aligned}
\begin{bmatrix} P_\mathcal{D}(s_{t+i} = \tilde{X}_i, a_{t+i} = 0 \mid X_0) & P_\mathcal{D}(s_{t+i} = \tilde{X}_i, a_{t+i} = 1 \mid X_0) \\ P_\mathcal{D}(s_{t+i} = X_i, a_{t+i} = 0 \mid X_0) & P_\mathcal{D}(s_{t+i} = X_i, a_{t+i} = 1 \mid X_0) \end{bmatrix} &= \begin{bmatrix} \delta & 0 \\ 0 & 1 - \delta \end{bmatrix} \\[8pt]
\begin{bmatrix} P_\mathcal{D}^\circ(s_{t+i} = \tilde{X}_i, a_{t+i} = 0 \mid X_0) & P_\mathcal{D}^\circ(s_{t+i} = \tilde{X}_i, a_{t+i} = 1 \mid X_0) \\ P_\mathcal{D}^\circ(s_{t+i} = X_i, a_{t+i} = 0 \mid X_0) & P_\mathcal{D}^\circ(s_{t+i} = X_i, a_{t+i} = 1 \mid X_0) \end{bmatrix} &= \begin{bmatrix} \delta^2 & (1-\delta)\delta \\ \delta(1-\delta) & (1-\delta)^2 \end{bmatrix}
\end{aligned}
\tag{75}
$$

From the calculation above, it is clear that

$$D_{\text{TV}}(P_{\mathcal{D}}^{\circ}(s_{t+i}, a_{t+i} \mid s_t) \parallel P_{\mathcal{D}}(s_{t+i}, a_{t+i} \mid s_t)) = \varepsilon_h, \quad \forall i \in \{1, 2, \cdots, h-1\}. \tag{76}$$

From the computed values of $P_{\mathcal{D}}^{\circ}(s_{t+h-1}, a_{t+h-1} \mid s_t)$ and $P_{\mathcal{D}}(s_{t+h-1}, a_{t+h-1} \mid s_t)$, we can derive

$$\begin{aligned} P_{\mathcal{D}}(s_{t+h} = Z \mid s_t = X_0) &= 0, \\ P_{\mathcal{D}}^{\circ}(s_{t+h} = Z \mid s_t = X_0) &= 2(1-\delta)\delta = \varepsilon_h. \end{aligned} \tag{77}$$

From the calculation above, it is clear that

$$D_{\text{TV}}(P_{\mathcal{D}}^{\circ}(s_{t+h} \mid s_t) \parallel P_{\mathcal{D}}(s_{t+h} \mid s_t)) = \varepsilon_h. \tag{78}$$

Up to now, we have checked that $\mathcal{D}$ is $\varepsilon_h$-open-loop consistent. Now, we are left with analyzing $\hat{V}_{\text{ac}}$ and $V_{\text{ac}}$. With some calculations, we can obtain the following:

$$\begin{aligned} \mathbb{E}_{P_{\mathcal{D}}^{\circ}} [R_{t:t+h}] &= 1 + \frac{(1-\varepsilon_h)(\gamma - \gamma^h)}{1-\gamma}, \\ \hat{V}_{\text{ac}}(X_0) &= \frac{1}{1-\gamma}, \\ V_{\text{ac}}(Z) &= 0. \end{aligned} \tag{79}$$

Now, we are ready to compute $V_{\text{ac}}(X_0)$:

$$\begin{aligned} V_{\text{ac}}(X_0) &= \frac{(1-\gamma^h) - \varepsilon_h(\gamma - \gamma^h)}{(1-\gamma)} + \gamma^h \left[(1-\varepsilon_h)V_{\text{ac}}(X_0) + \varepsilon_h V_{\text{ac}}(Z)\right] \\ &= \frac{1 - \gamma^h - \varepsilon_h(\gamma - \gamma^h)}{(1-\gamma)(1 - \gamma^h(1-\varepsilon_h))} \end{aligned} \tag{80}$$

Finally, with $X_0 \in \text{supp}(\mathcal{D})$, we obtain the desired value difference

$$\hat{V}_{\text{ac}}(X_0) - V_{\text{ac}}(X_0) = \frac{\varepsilon_h \gamma}{(1-\gamma)(1 - \gamma^h(1-\varepsilon_h))}. \tag{81}$$

By symmetry, we can flip the reward value (*i.e.*, $0 \to 1$ and $1 \to 0$) to construct the example such that

$$V_{\text{ac}}(X_0) - \hat{V}_{\text{ac}}(X_0) = \frac{\varepsilon_h \gamma}{(1-\gamma)(1 - \gamma^h(1-\varepsilon_h))}. \tag{82}$$

$$\square$$

### H.4 PROOF OF COROLLARY 1

> **Corollary 1** (Optimality Gap for AC Policy)   Let $\mathcal{D}^\star$ be the data collected by any optimal policy $\pi^\star$. If $\mathcal{D}^\star$ is weakly $\varepsilon_h$-open-loop consistent, then for all $s_t \in \mathrm{supp}(P_{\mathcal{D}^\star}(s_t))$,
>
> $$V^\star(s_t) - V^\star_{\mathrm{ac}}(s_t) \le V^\star(s_t) - \tilde{V}_{\mathrm{ac}}(s_t) \le \frac{\gamma \varepsilon_h}{(1-\gamma)(1-(1-\varepsilon_h)\gamma^h)} \le \varepsilon_h H \bar{H}, \qquad (13)$$
>
> where $V^\star$ is the value of the optimal policy $\pi^\star$, $V^\star_{\mathrm{ac}}$ is the *true* value of the optimal action chunking policy, and $\tilde{V}_{\mathrm{ac}}$ is the *true* value of the action chunking policy from cloning the data $\mathcal{D}^\star$:
>
> $$\tilde{\pi}^{\mathcal{D}^\star}_{\mathrm{ac}}(a_{t:t+h} \mid s_t) : s_t \mapsto P_{\mathcal{D}^\star}(\cdot \mid s_t). \qquad (14)$$

*Proof.* Let $\hat{V}_{\mathrm{ac}}$ be the fixed point of the following equation:

$$\hat{V}_{\mathrm{ac}}(s_t) = \mathbb{E}_{s_{t+1:t+h+1}, a_{t:t+h} \sim P_{\mathcal{D}^\star}(\cdot|s_t)} \left[ R_{t:t+h} + \gamma^h \hat{V}_{\mathrm{ac}}(s_{t+h}) \right] \qquad (83)$$

where again $R_{t:t+h} = \sum_{t'=t}^{t+h} \gamma^{t'-t} r(s_{t'}, a_{t'})$. The value of the optimal policy is the fixed point of the following equation:

$$
\begin{aligned}
V^\star(s_t) &= \mathbb{E}_{s_{t+1}, a_t \sim P_{\mathcal{D}^\star}(\cdot|s_t)} \left[ r(s_t, a_t) + \gamma V^\star(s_{t+1}) \right] \\
&= \mathbb{E}_{s_{t:t+2}, a_{t:t+1} \sim P_{\mathcal{D}^\star}(\cdot|s_t)} \left[ r(s_t, a_t) + \gamma r(s_{t+1}, a_{t+1}) + \gamma V^\star(s_{t+2}) \right] \\
&\cdots \\
&= \mathbb{E}_{s_{t+1:t+h+1}, a_{t:t+h} \sim P_{\mathcal{D}^\star}(\cdot|s_t)} \left[ R_{t:t+h} + \gamma^h V^\star(s_{t+h}) \right]
\end{aligned}
\qquad (84)
$$

which is equivalent to fixed-point equation for $\hat{V}_{\mathrm{ac}}$. Therefore $\hat{V}_{\mathrm{ac}} = V^\star$. By Theorem 1, we know that the true value $V_{\mathrm{ac}}$ of the action chunking policy $\tilde{\pi}_{\mathrm{ac}}$ that clones $\mathcal{D}^\star$ is close to $\hat{V}_{\mathrm{ac}}$. More specifically, for all $s_t \in \mathrm{supp}(\mathcal{D}^\star)$,

$$\left| \hat{V}_{\mathrm{ac}}(s_t) - \tilde{V}_{\mathrm{ac}}(s_t) \right| \le \frac{\gamma \varepsilon_h}{(1-\gamma)(1-(1-\varepsilon_h)\gamma^h)}, \qquad (85)$$

which means that

$$V^\star(s_t) - \tilde{V}_{\mathrm{ac}}(s_t) \le \frac{\gamma \varepsilon_h}{(1-\gamma)(1-(1-\varepsilon_h)\gamma^h)}, \qquad (86)$$

where we can remove the absolute value operator because $V^\star(s_t)$ is by definition always at least as large as $\tilde{V}_{\mathrm{ac}}(s_t)$. Since the optimal action chunking policy, by definition, attains equally good or better values (over $\mathcal{S}$) represented by $V_{\mathrm{ac}}$, and the optimal policy $\pi^\star$ also attains equally good or better value (*i.e.*, $V^\star$) compared to that of the optimal action chunking policy $\pi^\star_{\mathrm{ac}}$ (*i.e.*, $V^\star_{\mathrm{ac}}$), the following inequality holds for all $s_t \in \mathrm{supp}(\mathcal{D}^\star)$:

$$V^\star(s_t) \ge V^\star_{\mathrm{ac}}(s_t) \ge \tilde{V}_{\mathrm{ac}}(s_t). \qquad (87)$$

Therefore,

$$V^\star_{\mathrm{ac}}(s_t) - V^\star(s_t) \le \tilde{V}_{\mathrm{ac}}(s_t) - V^\star(s_t) \le \frac{\gamma \varepsilon_h}{(1-\gamma)(1-(1-\varepsilon_h)\gamma^h)}, \qquad (88)$$

as desired. $\qquad \square$

## H.5 PROOF OF COROLLARY 2

> **Corollary 2** (Worse-case Optimality Gap for Action Chunking Policy)  For any $h > 1, \gamma \in [0, 1), \varepsilon_h \in [0, 1/2]$, there exists an MDP $\mathcal{M}$ whose optimal policy $\pi^\star$ induces a data distribution $\mathcal{D}^\star$ that is weakly $\varepsilon_h$-open-loop consistent, such that for some $s_t \in \text{supp}(P_{\mathcal{D}^\star}(s_t))$,
>
> $$V^\star(s_t) - V^\star_{\text{ac}}(s_t) = \frac{\gamma \varepsilon_h}{(1 - \gamma)(1 - (1 - \varepsilon_h)\gamma^h)}. \tag{32}$$

*Proof.* To show this, we need a slightly more complicated MDP (compared to the $2h$-state MDP we use in the proof Section H.3). The MDP we construct for this proof is a $(3h - 1)$-state MDP as illustrated in Figure 8.

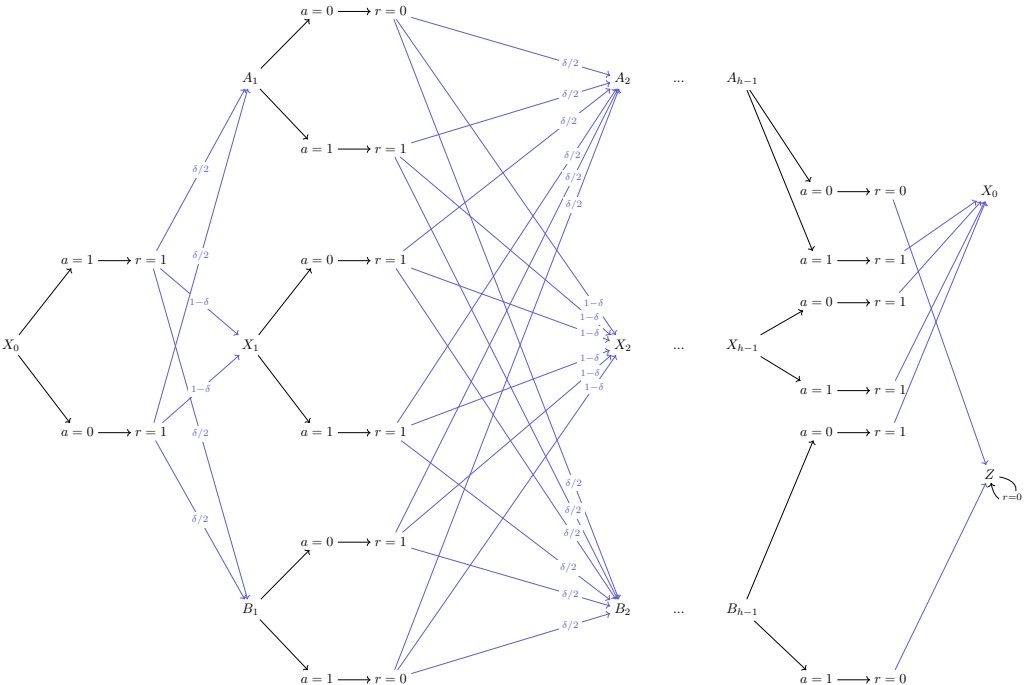

Figure 8: **A $(3h - 1)$-state MDP that is constructed to meet the upper-bound in Corollary 1.**

The optimal policy we pick is described as follows:

$$\begin{aligned}
\pi^\star(a = 0 \mid X_i) &= 1/2 \\
\pi^\star(a = 1 \mid X_i) &= 1/2 \\
\pi^\star(a = 1 \mid A_i) &= 1 \\
\pi^\star(a = 0 \mid B_i) &= 1/2
\end{aligned} \tag{89}$$

This induces the following state distribution,

$$\begin{aligned}
P_{\mathcal{D}^\star}(s_{t+i} = A_i \mid s_t = X_0) &= P_{\mathcal{D}^\star}(s_{t+i} = B_i \mid s_t = X_0) \\
&= P^\circ_{\mathcal{D}^\star}(s_{t+i} = A_i \mid s_t = X_0) = P^\circ_{\mathcal{D}^\star}(s_{t+i} = B_i \mid s_t = X_0) = \delta/2, \\
P_{\mathcal{D}^\star}(s_{t+i} = X_i \mid s_t = X_0) &= P^\circ_{\mathcal{D}^\star}(s_{t+i} = X_i \mid s_t = X_0) = 1 - \delta,
\end{aligned} \tag{90}$$

and a fully factorized distribution for the action chunk,

$$P^\circ_{\mathcal{D}^\star}(a_{t+i} = 0 \mid s_t) = P^\circ_{\mathcal{D}^\star}(a_{t+i} = 0 \mid s_t, a_{t:t+i}) = \frac{1}{2}(\delta_{a=0} + \delta_{a=1}). \tag{91}$$

Now, we derive the condition on $\delta$ when the optimal data $\mathcal{D}^\star$ is $\varepsilon_h$-open-loop consistent. We start by calculating the TV distance discrepancy for the future state-action distribution:

$$D_{\text{TV}}(P_{\mathcal{D}^\star}^{\text{open}}(s_{t+i}, a_{t+i} \mid s_t) \parallel P_{\mathcal{D}^\star}(s_{t+i}, a_{t+i} \mid s_t))$$

$$= \frac{1}{2} \left\| \begin{bmatrix} 0 & \delta/2 \\ (1-\delta)/2 & (1-\delta)/2 \\ \delta/2 & 0 \end{bmatrix} - \begin{bmatrix} \delta/4 & \delta/4 \\ (1-\delta)/2 & (1-\delta)/2 \\ \delta/4 & \delta/4 \end{bmatrix} \right\|_{1,1} \tag{92}$$

$$= \delta/2.$$

In the second line of the equations above, each row in the matrix corresponds to a distinct action $a_{t+i} \in \{0, 1\}$ and each row in the matrix corresponds to a distinct state $s_{t+i} \in \{A_i, X_i, B_i\}$.

Next, we calculate the TV distance discrepancy for $s_{t+h}$:

$$D_{\text{TV}}(P_{\mathcal{D}^\star}^{\text{open}}(s_{t+h} \mid s_t) \parallel P_{\mathcal{D}^\star}(s_{t+h} \mid s_t))$$

$$= \frac{1}{2} \left\| [1 \quad 0] - [1 - \delta/2 \quad \delta/2] \right\|_1 \tag{93}$$

$$= \delta/2.$$

In the second line of the equations above, each element in the vector corresponds to a distinct state $s_{t+h} \in \{X_0, Z\}$. Up to now, we have concluded that $\mathcal{D}^\star$ is $(\delta/2)$-open-loop consistent.

Due to the symmetric structure of this MDP, it is clear that any action chunking policy $\pi_{\text{ac}}(X_0) = a_{t:t+h}$ with $a_{t:t+h} \in \{0, 1\}$ is optimal and achieves the following value:

$$V_{\text{ac}}^\star(X_0) = 1 + (1 - \delta/2) \left[ \frac{\gamma - \gamma^h}{1 - \gamma} + \gamma^h V_{\text{ac}}^\star(X_0) \right]$$

$$= \frac{(1 - \gamma) + (1 - \delta/2)(\gamma - \gamma^h)}{(1 - \gamma)(1 - (1 - \delta/2)\gamma^h)}. \tag{94}$$

The optimal closed-loop policy can achieve the maximum possible return

$$V^\star(X_0) = \frac{1}{1 - \gamma}. \tag{95}$$

Therefore, with $\varepsilon_h = \delta/2$, the optimality gap achieved by this $(3h - 1)$-state MDP is

$$V^\star(X_0) - V_{\text{ac}}^\star(X_0) = \frac{\varepsilon_h \gamma}{(1 - \gamma)(1 - (1 - \varepsilon_h)\gamma^h)}, \tag{96}$$

as desired. $\qquad\qquad\square$

H.6   PROOF OF PROPOSITION 2

> **Proposition 2** (AC Q-Learning under Weak OLC)   For any $h > 1, \gamma \in [0, 1), c \in [0, 1/2)$, $\varepsilon_h \in (0, 1/2)$, there exists an MDP $\mathcal{M}$, a weakly $\varepsilon_h$-open-loop consistent $\mathcal{D}$ and $\mathcal{D}^\star$ with $\text{supp}(P_\mathcal{D}(s_t, a_{t:t+h})) \supseteq \text{supp}(P_{\mathcal{D}^\star}(s_t, a_{t:t+h}))$, such that for some $s_t \in \text{supp}(P_{\mathcal{D}^\star}(s_t))$,
>
> $$V^\star(s_t) - V_{\text{ac}}^+(s_t) = V_{\text{ac}}^\star(s_t) - V_{\text{ac}}^+(s_t) = \frac{\gamma c}{1 - \gamma}. \tag{33}$$

*Proof.* To prove this theorem, we show an example where the optimal action chunking policy can be highly sub-optimal in the absence of the strong open-loop consistency condition.

We define an MDP as follows. Let $\mathcal{S} = \{A, B, C, D, E, Z\}$ and $\mathcal{A} = \{0, 1\}$. Define the transition dynamics and reward function as shown in the diagram below (Figure 9):

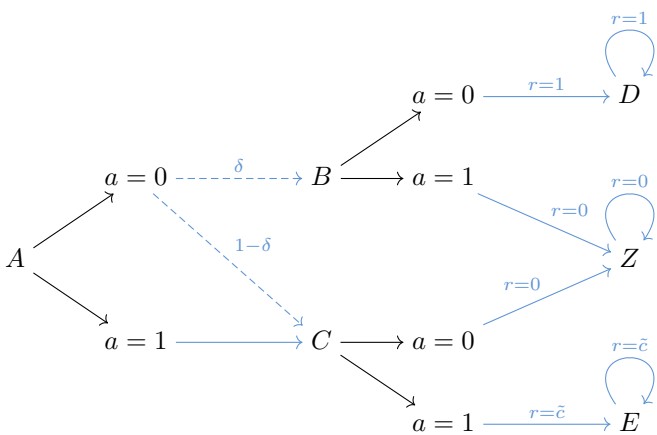

Figure 9: **A 6-state MDP that is constructed to illustrate the pathological failure mode of action chunking Q-learning under weak open-loop consistency.**

where $\delta, \tilde{c} \in [0, 1)$ are real numbers and dotted lines denote stochastic transitions. The reward function depends only on states ($r(A) = r(B) = r(C) = r(F) = 0, r(D) = 1$, and $r(G) = c$). Assume that the dataset is collected by a policy $\pi_\mathcal{D}$ defined as $\pi_\mathcal{D}(A) = 0$ (with probability $\theta$) or 1 (with probability $(1 - \theta)$), $\pi_\mathcal{D}(B) = 0$ (with probability 1), $\pi_\mathcal{D}(C) = 1$ (with probability 1), and $\pi_\mathcal{D}(D) = \pi_\mathcal{D}(Z) = \pi_\mathcal{D}(E) = 0$ (with probability 1).

Let $\pi^\star$ be the deterministic policy with $\pi(A) = 1, \pi(C) = 1, \pi(E) = 0$. Then, it is clear that $\text{supp}(P_\mathcal{D}(s_t, a_{t:t+h})) \supseteq \text{supp}(P_{\mathcal{D}^\star}(s_t, a_{t:t+h})) = \{(s_t = A, a_t = 1, a_{t+1} = 1, a_{t+2:t+h} = 0)\}$. Now we are left to show that $\mathcal{D}$ is $\varepsilon_h$-open-loop consistent.

Since $D, E, Z$ are all self-loops, we only need to analyze the first two actions in the action chunk; the rest is all $a_{t+2:t+h} = 0$ and does not affect the value.

We can compute the following:

$$\begin{aligned}
P_\mathcal{D}(A, (0, 0)) &= D, \ R(A, (0, 0)) = \gamma, \\
P_\mathcal{D}(A, (0, 1)) &= E, \ R(A, (0, 1)) = 2c\gamma, \\
P_\mathcal{D}(A, (1, 1)) &= E, \ R(A, (1, 1)) = 2c\gamma,
\end{aligned} \tag{97}$$

where we denote action chunks as a tuple and slightly abuse notation to denote deterministic outputs of $P_\mathcal{D}(\cdot \mid s_0, a_{0:2})$ (*e.g.*, $P_\mathcal{D}(A, (0, 0)) = D$ indicates that all length-2 trajectories in $\mathcal{D}$ from state $A$ with $a_0 = a_1 = 0$ have $s_2 = D$ with probability 1).

We can similarly compute the marginal state distributions as follows:

$$\begin{bmatrix} P_\mathcal{D}(D \mid A) \\ P_\mathcal{D}(Z \mid A) \\ P_\mathcal{D}(E \mid A) \end{bmatrix} = \begin{bmatrix} \theta\delta \\ 0 \\ 1 - \theta\delta \end{bmatrix}, \tag{98}$$

and

$$\begin{bmatrix} P_{\mathcal{D}}(s_{t+1} = B, a_t = 0 \mid s_t = A) & P_{\mathcal{D}}(s_{t+1} = B, a_t = 1 \mid s_t = A) \\ P_{\mathcal{D}}(s_{t+1} = C, a_t = 0 \mid s_t = A) & P_{\mathcal{D}}(s_{t+1} = C, a_t = 1 \mid s_t = A) \end{bmatrix} = \begin{bmatrix} \theta\delta & 0 \\ 0 & 1 - \theta\delta \end{bmatrix}. \qquad (99)$$

The marginal probability distribution of the action chunks is

$$\begin{aligned} P_{\mathcal{D}}(a_{0,1} = (0,0) \mid A) &= \theta\delta, \\ P_{\mathcal{D}}(a_{0,1} = (0,1) \mid A) &= (1 - \delta)\theta, \\ P_{\mathcal{D}}(a_{0,1} = (1,1) \mid A) &= 1 - \theta. \end{aligned} \qquad (100)$$

The induced $P_{\mathcal{D}}^{\circ}$ is then

$$\begin{bmatrix} P_{\mathcal{D}}^{\circ}(s_{t+h} = D \mid s_t = A) \\ P_{\mathcal{D}}^{\circ}(s_{t+h} = Z \mid s_t = A) \\ P_{\mathcal{D}}^{\circ}(s_{t+h} = E \mid s_t = A) \end{bmatrix} = \begin{bmatrix} \theta\delta^2 \\ (1-\delta)\theta\delta \\ (1-\delta)^2\theta + (1-\theta) \end{bmatrix}, \qquad (101)$$

and

$$\begin{aligned} &\begin{bmatrix} P_{\mathcal{D}}^{\circ}(s_{t+1} = B, a_{t+1} = 0 \mid s_t = A) & P_{\mathcal{D}}^{\circ}(s_{t+1} = B, a_{t+1} = 1 \mid s_t = A) \\ P_{\mathcal{D}}^{\circ}(s_{t+1} = C, a_{t+1} = 0 \mid s_t = A) & P_{\mathcal{D}}^{\circ}(s_{t+1} = C, a_{t+1} = 1 \mid s_t = A) \end{bmatrix} \\ &= \begin{bmatrix} \delta^2\theta & (1-\delta)\delta\theta \\ \delta(1-\delta)\theta & (1-\delta)^2\theta + 1 - \theta \end{bmatrix}. \end{aligned} \qquad (102)$$

We can then compute

$$\begin{aligned} D_{\mathrm{TV}}(P_{\mathcal{D}}^{\circ}(s_{t+1}, a_{t+1} \mid s_t = A) \parallel P_{\mathcal{D}}(s_{t+1}, a_{t+1} \mid s_t = A)) &= 2\theta\delta(1-\delta) \\ D_{\mathrm{TV}}(P_{\mathcal{D}}^{\circ}(s_{t+h} \mid s_t = A) \parallel P_{\mathcal{D}}(s_{t+h} \mid s_t = A)) &= \frac{3}{2}\theta\delta(1-\delta) \end{aligned} \qquad (103)$$

Note that we only need to check $t + 1$ time step for the state-action distribution because

$$\begin{aligned} D_{\mathrm{TV}}(P_{\mathcal{D}}^{\circ}(s_{t+h'}, a_{t+h'} \mid s_t = A) \parallel P_{\mathcal{D}}(s_{t+h'}, a_{t+h'} \mid s_t = A)) = \\ D_{\mathrm{TV}}(P_{\mathcal{D}}^{\circ}(s_{t+h} \mid s_t = A) \parallel P_{\mathcal{D}}(s_{t+h} \mid s_t = A)), \quad \forall h' > 1, \end{aligned} \qquad (104)$$

coming from the fact that the rest of the action chunks are all constant 0 (*i.e.*, $a_{t+2:t+h} = 0$).

Now, we can set $\delta$ to be solution of $\varepsilon_h = 2(1 - \delta)\delta\theta$, such that $\mathcal{D}$ is $\varepsilon_h$-open-loop consistent as required by the assumption.

Next, we analyze the action chunking policy $\pi_{\mathrm{ac}}^+$ learned from this $\varepsilon_h$-open-loop consistent data distribution. We start by calculating $\hat{Q}_{\mathrm{ac}}^+$ as follows:

$$\begin{aligned} \hat{Q}_{\mathrm{ac}}^+(A, (0,0)) &= \frac{\gamma}{1 - \gamma}, \\ \hat{Q}_{\mathrm{ac}}^+(A, (0,1)) &= \frac{\tilde{c}\gamma}{1 - \gamma}, \\ \hat{Q}_{\mathrm{ac}}^+(A, (1,1)) &= \frac{\tilde{c}\gamma}{1 - \gamma}. \end{aligned} \qquad (105)$$

The optimal action chunking policy is then $\hat{\pi}_{\mathrm{ac}}^+(A) = (0,0)$ because $c < 1$.

The true value of this action chunking policy is

$$V_{\mathrm{ac}}^+(A) = \frac{\delta\gamma}{1 - \gamma} \qquad (106)$$

As long as $\tilde{c} > \delta$, the optimal strategy in this MDP is to always choose $(a_0, a_1) = (1, 1)$, in which case the agent receives a constant return:

$$V^{\star}(A) = V_{\mathrm{ac}}^{\star}(A) = \frac{\tilde{c}\gamma}{1 - \gamma}. \qquad (107)$$

The optimality gap in this example is therefore

$$V^\star(A) - V_{\mathrm{ac}}^+(A) = (\tilde{c} - \delta)\frac{\gamma}{1 - \gamma}. \tag{108}$$

Finally, we solve $\varepsilon_h = 2(1 - \delta)\delta\theta$ and pick the smaller solution:

$$\delta = \frac{1 - \sqrt{1 - 2\varepsilon_h/\theta}}{2}. \tag{109}$$

If we set $\theta = 2\varepsilon_h$ and $\tilde{c} = c + \frac{1}{2}$, then we get

$$V^\star(A) - V_{\mathrm{ac}}^+(A) = \frac{c\gamma}{1 - \gamma}, \tag{110}$$

as desired.

As a small extra note, the last step is where the assumption $\varepsilon_h > 0$ becomes necessary, since otherwise the term $2\varepsilon_h/\theta$ (with $\theta = 2\varepsilon_h$) would be undefined. □

## H.7 PROOF OF THEOREM 2

**Theorem 2** (AC Q-Learning under Strong OLC)  If $\mathcal{D}$ and $\mathcal{D}^\star$ are strongly $\varepsilon_h$-open-loop consistent and $\text{supp}(P_{\mathcal{D}}(s_t, a_{t:t+h})) \supseteq \text{supp}(P_{\mathcal{D}^\star}(s_t, a_{t:t+h}))$, then for all $s_t \in \text{supp}(P_{\mathcal{D}^\star}(s_t))$,

$$V^\star(s_t) - V_{\text{ac}}^+(s_t) \leq \frac{\varepsilon_h \gamma}{1-\gamma}\left[\frac{2}{1-(1-2\varepsilon_h)\gamma^h} + \frac{1}{1-(1-\varepsilon_h)\gamma^h}\right] \leq 3\varepsilon_h H\bar{H}, \qquad (16)$$

where $V^\star$ is the value of a closed-loop optimal policy and $V_{\text{ac}}^+$ is the *true* value of $\pi_{\text{ac}}^+$.

*Proof of Theorem 2.*  We start by constructing a bound between $\hat{Q}_{\text{ac}}^+$ and $Q_{\text{ac}}^\star$, the solution of the following bellman equation:

$$Q_{\text{ac}}^\star(s_t, a_{t:t+h}) = \mathbb{E}_{s_{t+1:t+h+1} \sim P_{\mathcal{D}}^\circ(\cdot|s_t, a_{t:t+h})}\left[R_{t:t+h} + \gamma^h \max_{a_{t+h:t+2h}} Q_{\text{ac}}^\star(s_{t+h}, a_{t+h:t+2h})\right].$$
$$(111)$$

Intuitively, $Q_{\text{ac}}^\star$ is the Q-function of the optimal action chunking policy $\pi_{\text{ac}}^\star$ that can be learned from $\mathcal{D}$. Because $\text{supp}(\mathcal{D}) \supseteq \text{supp}(\mathcal{D}^\star)$, $\pi_{\text{ac}}^\star$ is at least as good as $\tilde{\pi}_{\text{ac}}$, the action chunking policy obtained by behavior cloning $\mathcal{D}^\star$. Bounding the difference between $\hat{Q}_{\text{ac}}^+$ and $Q_{\text{ac}}^\star$ allows us to leverage the bound in Corollary 1 to form a bound between $\hat{V}_{\text{ac}}^+$ and $V^\star$.

Since $\mathcal{D}$ is strongly $\varepsilon_h$-open-loop consistent,

$$D_{\text{TV}}(T(s_{t+h'} \mid s_t, a_{t:t+h'}) \parallel P_{\mathcal{D}}(s_{t+h'} \mid s_t, a_{t:t+h})) \leq \varepsilon_h, \forall h' \in \{1, \cdots, h-1\}. \qquad (112)$$

Since $\mathcal{D}^\star$ is also strongly $\varepsilon_h$-open-loop consistent,

$$D_{\text{TV}}(T(s_{t+h'} \mid s_t, a_{t:t+h'}) \parallel P_{\mathcal{D}^\star}(s_{t+h'} \mid s_t, a_{t:t+h})) \leq \varepsilon_h, \forall h' \in \{1, \cdots, h-1\}. \qquad (113)$$

Using the transitive property of TV distance, we have

$$D_{\text{TV}}(P_{\mathcal{D}}(s_{t+h'} \mid s_t, a_{t:t+h}) \parallel P_{\mathcal{D}^\star}(s_{t+h'} \mid s_t, a_{t:t+h})) \leq 2\varepsilon_h, \forall h' \in \{1, \cdots, h-1\}. \qquad (114)$$

Now, for the $h$-step reward, we have

$$\left|\mathbb{E}_{P_{\mathcal{D}}(\cdot|s_t, a_{t:t+h})}\left[R_{t:t+h}\right] - \mathbb{E}_{P_{\mathcal{D}^\star}(\cdot|s_t, a_{t:t+h})}\left[R_{t:t+h}\right]\right|$$
$$\leq \sum_{h'=1}^{h-1}\left[\gamma^{h'} D_{\text{TV}}(P_{\mathcal{D}}(s_{t+h'} \mid s_t, a_{t:t+h}) \parallel P_{\mathcal{D}^\star}(s_{t+h'} \mid s_t, a_{t:t+h}))\right] \qquad (115)$$
$$\leq \frac{2(\gamma - \gamma^h)\varepsilon_h}{1-\gamma}.$$

Similarly, for the value $h$-step into the future, we can use Lemma 2 to obtain the following bound:

$$\left|\mathbb{E}_{s_{t+h} \sim P_{\mathcal{D}}(s_{t+h}|s_t)}\left[V^\star(s_{t+h})\right] - \mathbb{E}_{s_{t+h} \sim P_{\mathcal{D}^\star}(s_{t+h}|s_t)}\left[\hat{V}_{\text{ac}}^+(s_{t+h})\right]\right|$$
$$\leq 2\varepsilon_h + (1-2\varepsilon_h)\sup_{s_{t+h} \in \mathcal{D}^\star}\left|V^\star(s_{t+h}) - \hat{V}_{\text{ac}}^+(s_{t+h})\right|. \qquad (116)$$

We define $Q^\star(s_t, a_{t:t+h})$ to be

$$Q^\star(s_t, a_{t:t+h}) := \mathbb{E}_{P_{\mathcal{D}^\star}(\cdot|s_t, a_{t:t+h})}\left[R_{t:t+h} + \gamma^h V^\star(s_{t+h})\right]. \qquad (117)$$

It is clear that

$$V^\star(s_t) = \mathbb{E}_{a_{t:t+h} \sim P_{\mathcal{D}^\star}}\left[Q^\star(s_t, a_{t:t+h})\right]. \qquad (118)$$

Combining the bound for the $h$-step reward and the bound on the value for $s_{t+h}$, for all $s_t, a_{t:t+h} \in \mathrm{supp}(P_{\mathcal{D}^\star}(s_t, a_{t:t+h}))$,

$$\Delta(s_t, a_{t:t+h}) = Q^\star(s_t, a_{t:t+h}) - \hat{Q}_{\mathrm{ac}}^+(s_t, a_{t:t+h})$$

$$\leq 2\varepsilon_h\gamma^h + \frac{2(\gamma - \gamma^h)\varepsilon_h}{1-\gamma} + (1 - 2\varepsilon_h)\gamma^h \left( V^\star(s_{t+h}) - \hat{V}_{\mathrm{ac}}^+(s_{t+h}) \right)$$

$$\leq \frac{2\varepsilon_h\gamma}{1-\gamma} + (1 - 2\varepsilon_h)\gamma^h \left( \mathbb{E}_{P_{\mathcal{D}^\star}} [Q^\star(s_{t+h}, a_{t+h:t+2h})] - \sup_{a_{t+h:t+2h}} \hat{Q}_{\mathrm{ac}}^+(s_{t+h}, a_{t+h:t+2h}) \right)$$

$$\leq \frac{2\varepsilon_h\gamma}{1-\gamma} + (1 - 2\varepsilon_h)\gamma^h \left( \mathbb{E}_{P_{\mathcal{D}^\star}} \left[ \hat{Q}_{\mathrm{ac}}^+(s_{t+h}, a_{t+h:t+2h}) + \Delta(s_{t+h}, a_{t+h:t+2h}) \right] - \sup_{a_{t+h:t+2h}} \hat{Q}_{\mathrm{ac}}^+(s_{t+h}, a_{t+h:t+2h}) \right)$$

$$\leq \frac{2\varepsilon_h\gamma}{1-\gamma} + (1 - 2\varepsilon_h)\gamma^h \sup_{s_{t+h}, a_{t+h:t+2h}} [\Delta(s_{t+h}, a_{t+h:t+2h})],$$

$$\tag{119}$$

which can be recursively expanded to obtain

$$V^\star(s_t) - \hat{V}_{\mathrm{ac}}^+(s_t) \leq \frac{2\varepsilon_h\gamma}{(1-\gamma)(1 - (1 - 2\varepsilon_h)\gamma^h)}. \tag{120}$$

By Theorem 1, for all $s_t \in \mathrm{supp}(\mathcal{D})$,

$$\left| \hat{V}_{\mathrm{ac}}^+(s_t) - V_{\mathrm{ac}}^+(s_t) \right| \leq \frac{\varepsilon_h\gamma}{(1-\gamma)(1 - (1 - \varepsilon_h)\gamma^h)}. \tag{121}$$

Combining the two inequalities above, for all $s_t \in \mathrm{supp}(\mathcal{D}^\star)$,

$$V^\star(s_t) - V_{\mathrm{ac}}^+(s_t) \leq \frac{\varepsilon_h\gamma}{1-\gamma} \left[ \frac{2}{1 - (1 - 2\varepsilon_h)\gamma^h} + \frac{1}{1 - (1 - \varepsilon_h)\gamma^h} \right]. \tag{122}$$

$\square$

## H.8   Proof of Theorem 5

> **Theorem 5** (Worst-case Analysis of Q-Learning with Action Chunking Policy on Off-policy Data)   For any $h > 1, \gamma \in (0, 1), \varepsilon_h \in (0, 1/5), c_1 \in (0, \varepsilon_h/2)$, and $c_2 \in (0, 2\varepsilon_h\gamma)$, there exists an MDP $\mathcal{M}$ and strongly $\varepsilon_h$-open-loop consistent data distributions $\mathcal{D}$ and $\mathcal{D}^\star$ with $\mathrm{supp}(P_{\mathcal{D}}(s_t, a_{t:t+h})) \supseteq \mathrm{supp}(P_{\mathcal{D}^\star}(s_t, a_{t:t+h}))$, such that for some $s_t \in \mathrm{supp}(P_{\mathcal{D}^\star}(s_t))$,
>
> $$V^\star(s_t) - V_{\mathrm{ac}}^+(s_t) = \frac{2\varepsilon_h\gamma - c_2}{(1 - \gamma)(1 - (1 - 2\varepsilon_h)\gamma^h)} + \frac{\varepsilon_h\gamma}{(1 - \gamma)(1 - (1 - \varepsilon_h - c_1)\gamma^h)}, \quad (34)$$
>
> where $V^\star$ is the value of an optimal policy and $V_{\mathrm{ac}}^+$ is the *true* value of $\pi_{\mathrm{ac}}^+$. As $c_1, c_2 \to 0$,
>
> $$V^\star(s_t) - V_{\mathrm{ac}}^+(s_t) \to \frac{\varepsilon_h\gamma}{1 - \gamma} \left[ \frac{2}{1 - (1 - 2\varepsilon_h)\gamma^h} + \frac{1}{1 - (1 - \varepsilon_h)\gamma^h} \right]. \quad (35)$$

The examples in the following proof of Theorem 5 (available in Section H.8) provide insights on the factor of 3 in $V^\star - V_{\mathrm{ac}}^+ \le 3\varepsilon_h H\bar{H}$ (with $H = 1/(1 - \gamma), \bar{H} = 1/(1 - \gamma^h)$) is necessary. In particular, the worst case can be roughly seen as a combination of the two main results that we have presented so far:

1. $V^\star - V_{\mathrm{ac}}^\star \approx \varepsilon_h H\bar{H}$ (Corollary 1, Corollary 2): the optimal action chunking policy is $(\varepsilon_h H^2)$-sub-optimal due to its inability to react to environment stochasticity, quantified by the strongly-$\varepsilon_h$ open-loop consistency of $\mathcal{D}^\star$.

2. $V_{\mathrm{ac}}^\star - \hat{V}_{\mathrm{ac}}^+ \approx \varepsilon_h H\bar{H}$ (a transformation of Theorem 1 and Theorem 4 on the optimal action chunking policy $\pi_{\mathrm{ac}}^\star$): the value *under-estimation* bias can incur another factor of $\varepsilon_h H\bar{H}$ bringing up the sub-optimality of $\hat{V}_{\mathrm{ac}}^+$ to at most $2\varepsilon_h H\bar{H}$, and finally,

3. $\hat{V}_{\mathrm{ac}}^+ - V_{\mathrm{ac}}^+ \approx \varepsilon_h H\bar{H}$ (Theorem 1, Theorem 4): the action chunking value function may prefer an *overestimated* action chunking policy $\pi_{\mathrm{ac}}^+$ where its actual value is again $\varepsilon_h H\bar{H}$ from its estimated value, resulting in a total sub-optimality of $3\varepsilon_h H\bar{H}$.

Our construction (in the proof of Theorem 5) directly builds on the above insights by using a 2-part MDP where the first part corresponds to an $(\varepsilon_h H\bar{H})$-underestimated action chunking policy that has a $(\varepsilon_h H\bar{H})$-optimality gap from the optimal closed-loop policy and the second part corresponds to an $(\varepsilon_h H\bar{H})$-overestimated action chunking policy that has a $(3\varepsilon_h H\bar{H})$-optimality gap that is preferred by the value function.

Before we start our main proof, we first introduce a Lemma that helps simplifies the inequalities.

> **Lemma 5** (Optimality gap comparator)   For any $\tilde{\gamma} \in [0, 1)$ and $0 < \varepsilon_1 < \varepsilon_2 < 1$,
>
> $$\frac{\varepsilon_1}{1 - (1 - \varepsilon_1)\tilde{\gamma}} < \frac{\varepsilon_2}{1 - (1 - \varepsilon_2)\tilde{\gamma}}. \quad (123)$$

*Proof.*

$$
\begin{aligned}
0 &< (1 - \gamma)(\varepsilon_2 - \varepsilon_1) \\
&= \varepsilon_2 - \varepsilon_2\tilde{\gamma} - \varepsilon_1 + \varepsilon_1\tilde{\gamma} \\
&= \varepsilon_2 - \varepsilon_2\tilde{\gamma} + \varepsilon_1\varepsilon_2\tilde{\gamma} - \varepsilon_1 + \varepsilon_1\tilde{\gamma} - \varepsilon_1\varepsilon_2\tilde{\gamma} \\
&= \varepsilon_2(1 - (1 - \varepsilon_1)\tilde{\gamma}) - \varepsilon_1(1 - (1 - \varepsilon_2)\tilde{\gamma})
\end{aligned}
\quad (124)
$$

Since $1 - (1 - \varepsilon_1)\tilde{\gamma} > 0$ and $1 - (1 - \varepsilon_2)\tilde{\gamma} > 0$, we can divide both sides by $(1 - (1 - \varepsilon_1)\tilde{\gamma})(1 - (1 - \varepsilon_2)\tilde{\gamma})$ to get

$$0 < \frac{\varepsilon_2}{1 - (1 - \varepsilon_2)\tilde{\gamma}} - \frac{\varepsilon_1}{1 - (1 - \varepsilon_1)\tilde{\gamma}}, \quad (125)$$

as desired. $\qquad\square$

Now, we begin the main proof as follows.

*Proof of Theorem 5.* We prove by constructing the following $(2h+4)$-state MDP where the agent can take any of the three actions $\{0, 1, 2\}$ at each state (see a diagram in Figure 10).

**Notations:** we start by introducing some abbreviations for all action chunks that appear in this proof:

$$
\begin{aligned}
a_{t:t+h}^{\star} &= (0, 0, 0, \cdots, 0) \\
a_{t:t+h}^{\diamond} &= (0, 1, 0, \cdots, 0) \\
a_{t:t+h}^{\bullet} &= (0, 2, 0, \cdots, 0) \\
a_{t:t+h}^{\triangle} &= (1, 1, 1, \cdots, 1) \\
a_{t:t+h}^{\circ} &= (1, 0, 1, \cdots, 1) \\
a_{t:t+h}^{\times} &= (1, 2, 1, \cdots, 1)
\end{aligned}
\tag{126}
$$

The first three action chunks $a_{t:t+h}^{\star}, a_{t:t+h}^{\diamond}, a_{t:t+h}^{\bullet}$ are only possible in the top branch and the last three action chunks $a_{t:t+h}^{\triangle}, a_{t:t+h}^{\circ}, a_{t:t+h}^{\times}$ are only possible in the bottom branch because the first action in the action chunk deterministically divides it into the two branches.

Among these action chunks, it is clear by inspection that $\pi_{\mathrm{ac}}(X_0) = (0, 0, \cdots, 0)$ is the optimal action chunking policy, and thus we directly use '$\star$' to denote $a_{t:t+h}^{\star} = (0, 0, \cdots, 0)$. $a_{t:t+h}^{\triangle}$ is also of great importance: as we will show later, $\pi_{\mathrm{ac}}^{+}(X_0) = a_{t:t+h}^{\triangle}$. The *actual* values and *nominal/estimated* values for these action chunks are $(V_{\mathrm{ac}}^{\star}, V_{\mathrm{ac}}^{\diamond}, V_{\mathrm{ac}}^{\bullet}, V_{\mathrm{ac}}^{\triangle}, V_{\mathrm{ac}}^{\circ}, V_{\mathrm{ac}}^{\times})$ and $(\hat{V}_{\mathrm{ac}}^{\star}, \hat{V}_{\mathrm{ac}}^{\diamond}, \hat{V}_{\mathrm{ac}}^{\bullet}, \hat{V}_{\mathrm{ac}}^{\triangle}, \hat{V}_{\mathrm{ac}}^{\circ}, \hat{V}_{\mathrm{ac}}^{\times})$ respectively. Much of the focus of this proof is to calculate the optimality gap, which is the difference between the optimal closed-loop value and the action chunking policy value (either estimated or actual):

$$
\texttt{actual optimality gap:} \quad V^{\star}(X_0) - V_{\mathrm{ac}}^{[\cdot]}(X_0) \tag{127}
$$

$$
\texttt{nominal optimality gap:} \quad V^{\star}(X_0) - \hat{V}_{\mathrm{ac}}^{[\cdot]}(X_0) \tag{128}
$$

**High-level proof sketch:** The MDP contains two branches: a top branch where (as we will show) both the optimal policy $\pi^{\star}$ and the optimal action chunking policy $\pi_{\mathrm{ac}}^{\star}$ take, and a bottom branch where (as we will also show) the learned action chunking policy $\pi_{\mathrm{ac}}^{+}$ takes. The key idea of the construction is that for the top branch, we have

$$
V^{\star}(X_0) - \hat{V}_{\mathrm{ac}}^{\star}(X_0) \approx \frac{2\varepsilon_h \gamma}{(1-\gamma)(1 - (1 - 2\varepsilon_h)\gamma^h)}, \tag{129}
$$

and for the bottom branch, we have

$$
\hat{V}_{\mathrm{ac}}^{\star}(X_0) < \hat{V}_{\mathrm{ac}}^{+}(X_0) \approx V_{\mathrm{ac}}^{+}(X_0) + \frac{\varepsilon_h \gamma}{(1-\gamma)(1 - (1 - \varepsilon_h)\gamma^h)}. \tag{130}
$$

Combining these two together gives

$$
V^{\star}(X_0) - V_{\mathrm{ac}}^{+}(X_0) \approx \frac{\varepsilon_h \gamma}{1-\gamma} \left[ \frac{2}{1 - (1 - 2\varepsilon_h)\gamma^h} + \frac{1}{1 - (1 - \varepsilon_h)\gamma^h} \right]. \tag{131}
$$

We use '$\approx$' because the equalities are not strictly achievable but (as we will show) can be made arbitrarily close.

The proof can be roughly divided into the following steps (we use '$\approx$' to help illustrate the high-level idea below and use more precise argument in the actual proof):

1. MDP description: we formally describe the transition dynamics $T$ and the reward function $r$ for each state-action pair for both the top and the bottom branches.

2. Strong $\varepsilon_h$-open-loop consistency of $\mathcal{D}^{\star}$: we then check the strong open-loop consistency assumption for $\mathcal{D}^{\star}$.

3. Data distribution $\mathcal{D}_{\mathrm{top}}$ for the top branch: we use a mixture data distribution from two policies to construct $\mathcal{D}_{\mathrm{top}}$.

4. **Strong $\varepsilon_h$-open-loop consistency of $\mathcal{D}_{\text{top}}$:** we then check that the constructed data distribution of the top branch satisfies the strongly open-loop consistency assumption. Note that we can do so separately for the top and the bottom because these two distributions have non-overlapping support in $a_{t:t+h}$.

5. **The optimality gap and value estimation error for the top branch:** we prove that $V^\star(X_0) - V^\star_{\text{ac}}(X_0) = \frac{\varepsilon_h \gamma}{(1-\gamma)(1-(1-2\varepsilon_h)\gamma^h)}$ and $V^\star(X_0) - \hat{V}^\star_{\text{ac}}(X_0) = \frac{2\varepsilon_h \gamma}{(1-\gamma)(1-(1-2\varepsilon_h)\gamma^h)}$ and the other two possible action chunks $a^\diamond_{t:t+h} = (0,1,0,\cdots)$ and $a^\bullet_{t:t+h} = (0,2,0,\cdots)$ both admit lower estimated values compared to $a^\star_{t:t+h}$: $\hat{V}^\diamond_{\text{ac}}(X_0) < \hat{V}^\star_{\text{ac}}(X_0)$ and $\hat{V}^\bullet_{\text{ac}}(X_0) < \hat{V}^\star_{\text{ac}}(X_0)$.

6. **Data distribution $\mathcal{D}_{\text{bottom}}$ for the bottom branch:** we again use a mixture data distribution from two different policies to construct $\mathcal{D}_{\text{bottom}}$.

7. **Strong $\varepsilon_h$-open-loop consistency of $\mathcal{D}_{\text{bottom}}$:** we then check that the constructed data distribution of the bottom branch satisfies the strongly open-loop consistency assumption.

8. **The optimality gap and value estimation error for the bottom branch:** we prove that $V^\star(X_0) - \hat{V}^\triangle_{\text{ac}}(X_0) \approx \frac{2\varepsilon_h \gamma}{(1-\gamma)(1-(1-2\varepsilon_h)\gamma^h)}$ and $\hat{V}^\triangle_{\text{ac}}(X_0) - V^\triangle_{\text{ac}}(X_0) = \frac{\varepsilon_h \gamma}{(1-\gamma)(1-(1-\varepsilon_h)\gamma^h)}$, and the other two possible action chunks $a^\diamond_{t:t+h} = (1,0,0,\cdots)$ and $a^\times_{t:t+h} = (1,2,0,\cdots)$ both admit lower estimated values compared to $a^\triangle_{t:t+h}$: $\hat{V}^\diamond_{\text{ac}}(X_0) < \hat{V}^\triangle_{\text{ac}}(X_0)$ and $\hat{V}^\times_{\text{ac}}(X_0) < \hat{V}^\triangle_{\text{ac}}(X_0)$. Moreover $a^\star_{t:t+h}$ also admits a lower estimated value compared to $a^\triangle_{t:t+h}$: $\hat{V}^\star_{\text{ac}}(X_0) < \hat{V}^\triangle_{\text{ac}}(X_0)$ which proves $\pi^+_{\text{ac}}(X_0) = a^\triangle_{t:t+h}$ and thus concluding our proof: $V^\star(X_0) - V^+_{\text{ac}}(X_0) \approx \frac{2\varepsilon_h \gamma}{(1-\gamma)(1-(1-2\varepsilon_h)\gamma^h)} + \frac{\varepsilon_h \gamma}{(1-\gamma)(1-(1-\varepsilon_h)\gamma^h)}$.

Now we begin our proof as follows.

Step 1. MDP description (Figure 10).

The transition function $T$ of the MDP is defined as follows (from left to right):

$$
\begin{aligned}
T(Z \mid Z, a) = T(G \mid G, a) &= 1, \quad \forall a, \\
T(Z \mid s, a = 2) &= 1, \quad \forall a, \forall s : s \neq G \\
T(X_1 \mid X_0, a = 0) &= 1 - 2\varepsilon_h \\
T(\tilde{X}_1 \mid X_0, a = 0) &= \varepsilon_h \\
T(C \mid X_0, a = 0) &= \varepsilon_h \\
T(Y_1 \mid X_0, a = 1) &= 1 - \varepsilon_h - c_1 \\
T(\tilde{Y}_1 \mid X_0, a = 1) &= \varepsilon_h \\
T(G \mid X_0, a = 1) &= c_1 \\
T(X_2 \mid X_1, a = 0) &= 1 \\
T(X_2 \mid \tilde{X}_1, a = 1) &= 1 \\
T(X_2 \mid C, a = 1) &= 1 \\
T(Z \mid X_1, a = 1) &= 1 \\
T(Z \mid C, a = 0) &= 1 \\
T(G \mid \tilde{X}_1, a = 0) &= 1 \\
T(Y_2 \mid Y_1, a = 1) &= 1 \\
T(Y_2 \mid \tilde{Y}_1, a = 0) &= 1 \\
T(Z \mid Y_1, a = 0) &= 1 \\
T(Z \mid \tilde{Y}_1, a = 1) &= 1 \\
T(X_{i+1} \mid X_i, a \in \{0,1\}) = T(Y_{i+1} \mid Y_i, a \in \{0,1\}) &= 1, \quad \forall i \in \{2, \cdots, h-2\} \\
T(X_0 \mid X_{h-1}, a \in \{0,1\}) = T(Y_0 \mid Y_{h-1}, a \in \{0,1\}) &= 1
\end{aligned}
\tag{132}
$$

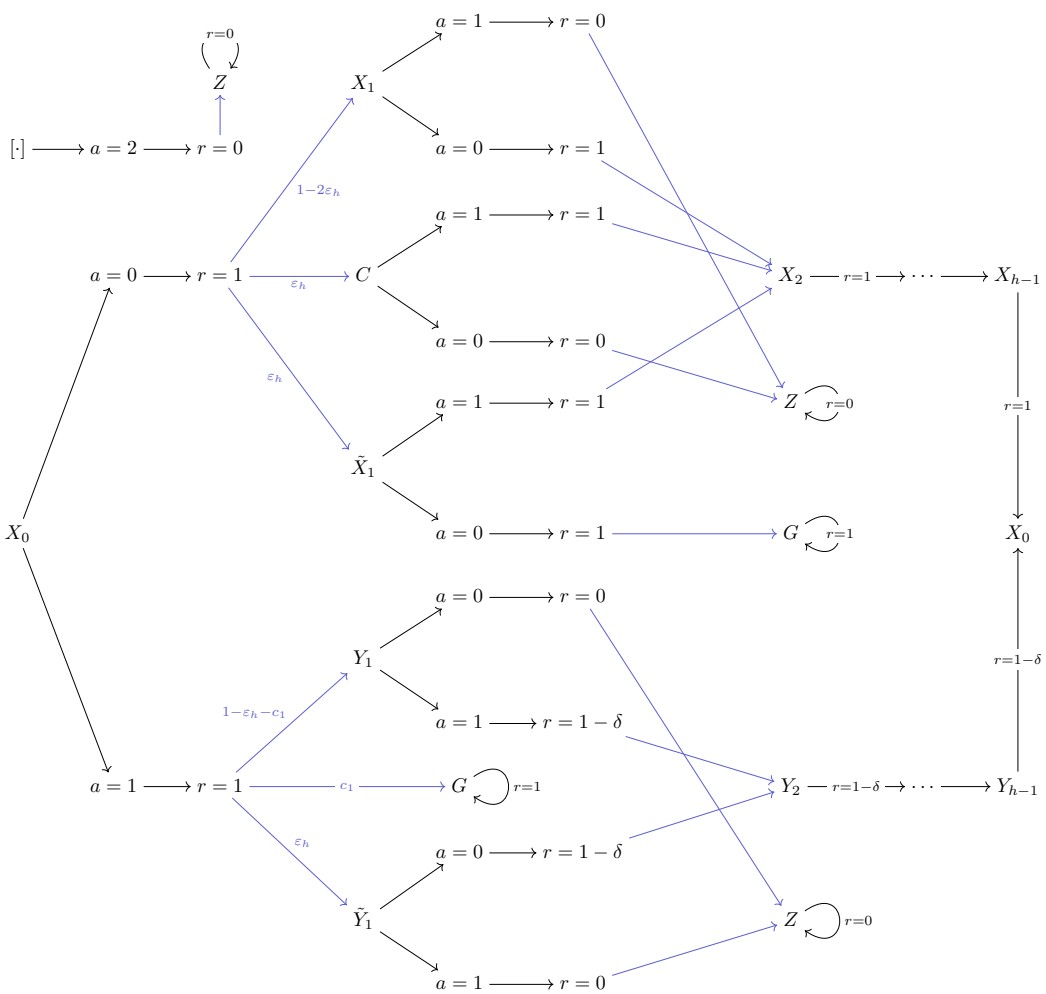

Figure 10: **A $(2h + 4)$-state MDP that is constructed to illustrate the MDP constructed to meet the exact upper-bound optimality gap in Theorem 2.** We redraw the same states $Z$, $G$, $X_0$ in multiple locations in the diagram above for better clarity.

The reward function is defined as

$$
\begin{aligned}
r(Z, a) &= 0, \quad \forall a \\
r(G, a) &= 1, \quad \forall a \\
r(s, a = 2) &= 0, \quad \forall s : s \neq G \\
r(X_0, a = 0) = r(X_0, a = 1) &= 1, \\
r(C, a = 1) = r(X_1, a = 0) = r(\tilde{X}_1, a \in \{0, 1\}) &= 1, \\
r(C, a = 0) = r(X_1, a = 1) &= 0, \\
r(Y_1, a = 1) = r(\tilde{Y}_1, a = 0) &= 1 - \delta, \\
r(Y_1, a = 0) = r(\tilde{Y}_1, a = 1) &= 0, \\
r(X_i, a \in \{0, 1\}) &= 1, \quad \forall i \in \{2, \cdots, h - 1\} \\
r(Y_i, a \in \{0, 1\}) &= 1 - \delta, \quad \forall i \in \{2, \cdots, h - 1\}
\end{aligned}
\tag{133}
$$

Notably, there are some special states:

- State $Z$: a self-looping "black hole" state that always gets $0$ reward at each time step and thus has a constant value of $0$.

- State $G$: a self-looping "black hole" state that always gets 1 reward at each time step and thus has a constant value of $1/(1-\gamma)$.

- State $X_0$: the special state that branches out based on the action taken. The agent periodically visit this state every $h$ steps unless it has been trapped in either $Z$ or $G$. As we proceed in the proof, we will encounter factors in the form of $\frac{1}{1-b\gamma^h}$ in the calculation of the optimality gap. These factors come from the agent looping around and revisiting $X_0$ with $b$-probability each cycle.

These two absorbing states are important because their values sit at the boundary of the value range of our value function $V(s) \in [0, 1/(1-\gamma)]$. Shifting the reaching probability from $Z$ to $G$ or the other way around results in the biggest possible difference in the policy value. Our construction hinges on the constructing $\mathcal{D}$ such that

1. $P_{\mathcal{D}}(\cdot \mid s_t, \pi^\star(s_t))$ and $T(\cdot \mid s_t, \pi^\star(s_t))$ differs by only $\varepsilon_h$ (in TV distance as required by the strongly open-loop consistency assumption) where precisely $\varepsilon_h$ probability mass is moved from reaching state $Z$ to reaching state $G$. This causes the $\hat{V}_{\mathrm{ac}}^\star$ to precisely underestimates the value of $V_{\mathrm{ac}}^\star$ by $\frac{\varepsilon_h \gamma^h}{(1-\gamma)(1-(1-2\varepsilon_h)\gamma^h)}$. It is worth noting that we cannot make the $2\varepsilon_h$ in the denominator $\varepsilon_h$ because $V_{\mathrm{ac}}^\star$ needs to simultaneously maintain a value gap with $V^\star$. If we were to construct an example where $\hat{V}_{\mathrm{ac}}^\star(X_0) - V_{\mathrm{ac}}^\star(X_0) = V_{\mathrm{ac}}^\star$ be $\frac{\varepsilon_h \gamma^h}{(1-\gamma)(1-(1-\varepsilon_h)\gamma^h)}$, it would enforce $V_{\mathrm{ac}}^\star(X_0) = V^\star(X_0)$ because there would be no probability mass left to create the gap between $V^\star$ and $V_{\mathrm{ac}}^\star$. With an extra $\varepsilon_h$ in the denominator, we can also make the optimality gap of $V_{\mathrm{ac}}^\star$ precisely $\frac{\varepsilon_h \gamma^h}{(1-\gamma)(1-(1-2\varepsilon_h)\gamma^h)}$, bringing the combined value gap (between $V^\star$ and $\hat{V}_{\mathrm{ac}}^\star$) up to $\frac{2\varepsilon_h \gamma^h}{(1-\gamma)(1-(1-2\varepsilon_h)\gamma^h)}$.

2. $P_{\mathcal{D}}(\cdot \mid s_t, \pi^+(s_t))$ and $T(\cdot \mid s_t, \pi^+(s_t))$ differs by only $\varepsilon_h$ (again in TV distance as required by the strongly open-loop consistency assumption) where precisely $\varepsilon_h$ probability mass is moved from reaching state $G$ to reaching state $Z$. This causes the $\hat{V}_{\mathrm{ac}}^+$ to precisely overestimates the value of $V_{\mathrm{ac}}^+$ by $\frac{\varepsilon_h \gamma^h}{(1-\gamma)(1-(1-\varepsilon_h)\gamma^h)}$.

We use a special action $a = 2$ where upon taking the action the agent immediately transitions to $Z$ and receives a reward of $0$ (except in $G$). As we will see soon, this action is useful for constructing a data distribution with an easily 'controllable' probability of reaching $Z$ for the top branch and an easily 'controllable' probability of reaching $G$ for the bottom branch. Before we start constructing $\mathcal{D}$, we first check the condition that $\mathcal{D}^\star$ is strongly $\varepsilon_h$-open-loop consistent.

Step 2.  Strong $\varepsilon_h$-open-loop consistency of $\mathcal{D}^\star$: It is clear that one possible $\pi^\star$ that achieves $1/(1-\gamma)$ value is

$$
\begin{aligned}
\pi^\star(X_i) &= 0 \\
\pi^\star(C) &= 1 \\
\pi^\star(\tilde{X}) &= 0
\end{aligned}
\tag{134}
$$

We can easily check that $\mathcal{D}^\star$ collected by $\pi^\star$ is strongly $\varepsilon_h$-open-loop consistent by observing that the only path that $\pi^\star$ outputs $(0, 1, 0, 0, \cdots)$ has $\varepsilon_h$ probability, which causes the state distribution of $s_{t+1}$ to differ by at most $\varepsilon_h$ under the TV distance (subject to $a = (0, 1, 0, 0, \cdots)$ or $a = (0, 0, 0, \cdots)$ conditioning). This concludes that $\mathcal{D}^\star$ generated by $\pi^\star$ above is strongly $\varepsilon_h$-open-loop consistent.

Now, depending on the first action $a_t$, the MDP can be decomposed into two parts: *the top* ($a = 0$) and *the bottom* ($a = 1$). We construct the data distribution for each branch and analyze the *actual* and *nominal* optimality gap for each branch in the following steps.

Step 3.  Data distribution $\mathcal{D}_{\mathrm{top}}$ for the top branch: we use a mixture of the following two policies to construct a strongly $\varepsilon_h$-open-loop consistent $\mathcal{D}_{\mathrm{top}}$.

*Policy $\pi_{\mathrm{top}}^1$:*

$$
\begin{aligned}
\pi_{\mathrm{top}}^1(X_0) = \pi_{\mathrm{top}}^1(C) = \pi_{\mathrm{top}}^1(Z) &= 0, \\
\pi_{\mathrm{top}}^1(X_1) = \pi_{\mathrm{top}}^1(\tilde{X}_1) &= 2.
\end{aligned}
\tag{135}
$$

$\pi^1_{\text{top}}$ always take $a = 2$ unless it is in state $X_0$, $C$ or $Z$ where it always takes $a = 0$. It is clear that this policy only produces two possible action chunks: $a_{t:t+h} = (0, 0, \cdots, 0)$ or $a_{t:t+h} = a^\bullet_{t:t+h} := (0, 2, 0, \cdots)$. We note that the $a_{t:t+h}$ policy always leads to state $Z$:

$$P_{\mathcal{D}_{\pi^1_{\text{top}}}}(s_{t+h} = Z \mid s_t, a_{t:t+h} = 0) = 1. \tag{136}$$

*Policy $\pi^2_{\text{top}}$:*

$$
\begin{aligned}
\pi^2_{\text{top}}(X_0) &= 0, \\
\pi^2_{\text{top}}(\tilde{X}_1) = \pi^2_{\text{top}}(\tilde{C}) &= 1, \\
\pi^2_{\text{top}}(a = 0 \mid X_1) &= 1 - \delta_G, \\
\pi^2_{\text{top}}(a = 1 \mid X_1) &= \delta_G,
\end{aligned}
\tag{137}
$$

with some $\delta_G \in (0, 1)$. $\pi^2_{\text{top}}$ can also only produce two possible action chunks: $a_{t:t+h} = (0, 0, \cdots, 0)$ or $a_{t:t+h} = a^\diamond_{t:t+h} := (0, 1, 0, \cdots, 0)$.

The distribution of $s_{t+h}$ conditioned on $a_{t:t+h} = 0$ is

$$
\begin{aligned}
P_{\mathcal{D}_{\pi^2_{\text{top}}}}(s_{t+h} = Z \mid s_t, a_{t:t+h} = 0) &= 0, \\
P_{\mathcal{D}_{\pi^2_{\text{top}}}}(s_{t+h} = G \mid s_t, a_{t:t+h} = 0) &= 0, \\
P_{\mathcal{D}_{\pi^2_{\text{top}}}}(s_{t+h} = X_0 \mid s_t, a_{t:t+h} = 0) &= 1.
\end{aligned}
\tag{138}
$$

*Mixing $\pi^1_{\text{top}}$ and $\pi^2_{\text{top}}$:* Let $\mathcal{D}_{\text{top}}$ be a mixture of $\mathcal{D}_{\pi^1_{\text{top}}}$ and $\mathcal{D}_{\pi^2_{\text{top}}}$:

$$P_{\mathcal{D}_{\text{top}}} = (1 - \varsigma)P_{\mathcal{D}^1_{\text{top}}} + \varsigma P_{\mathcal{D}^2_{\text{top}}}, \tag{139}$$

where

$$\varsigma = \frac{1}{2(1 - \delta_G) + 1}. \tag{140}$$

It is clear that $0 < \varsigma < 1$ (because $\delta_G \in (0, 1)$), making it valid mixing ratio.

We now compute the marginal state distribution of the mixture by first analyzing the action probability:

$$
\begin{aligned}
P_{\mathcal{D}^1_{\text{top}}}(a^\star_{t:t+h} \mid s_t) &= \varepsilon_h, \\
P_{\mathcal{D}^2_{\text{top}}}(a^\star_{t:t+h} \mid s_t) &= (1 - 2\varepsilon_h)(1 - \delta_G).
\end{aligned}
\tag{141}
$$

The state marginals are then

$$
\begin{aligned}
P_{\mathcal{D}_{\text{top}}}(s_{t+h} = Z \mid s_t, a^\star_{t:t+h}) &= \frac{P_{\mathcal{D}_{\text{top}}}(s_{t+h} = Z, a^\star_{t:t+h} \mid s_t)}{P_{\mathcal{D}_{\text{top}}}(a^\star_{t:t+h} \mid s_t)} \\
&= \frac{(1 - \varsigma)P_{\mathcal{D}^1_{\text{top}}}(a^\star_{t:t+h} \mid s_t)}{(1 - \varsigma)P_{\mathcal{D}^1_{\text{top}}}(a^\star_{t:t+h} \mid s_t) + \varsigma P_{\mathcal{D}^2_{\text{top}}}(a^\star_{t:t+h} \mid s_t)} \\
&= \frac{\varepsilon_h(1 - \varsigma)}{\varepsilon_h(1 - \varsigma) + (1 - 2\varepsilon_h)(1 - \delta_G)\varsigma} \\
&= 2\varepsilon_h.
\end{aligned}
\tag{142}
$$

Therefore,

$$
\begin{aligned}
P_{\mathcal{D}_{\text{top}}}(s_{t+h} = Z \mid s_t, a^\star_{t:t+h}) &= 2\varepsilon_h, \\
P_{\mathcal{D}_{\text{top}}}(s_{t+h} = X_0 \mid s_t, a^\star_{t:t+h}) &= 1 - 2\varepsilon_h, \\
P_{\mathcal{D}_{\text{top}}}(s_{t+h} = G \mid s_t, a^\star_{t:t+h}) &= 0.
\end{aligned}
\tag{143}
$$

**Step 4. Strong $\varepsilon_h$-open-loop consistency of $\mathcal{D}_{\text{top}}$:** Now, we check for strong open-loop consistency for the three possible action chunks on the top branch:

$$
\begin{aligned}
a^\star_{t:t+h} &= (0, 0, 0, \cdots) \\
a^\diamond_{t:t+h} &= (0, 1, 0, \cdots) \\
a^\bullet_{t:t+h} &= (0, 2, 0, \cdots)
\end{aligned}
\tag{144}
$$

For $a_{t:t+h}^\star = 0$, we can compute open-loop marginal state distribution as follows:

$$
\begin{aligned}
T(s_{t+h} = Z \mid s_t, a_{t:t+h}^\star) &= \varepsilon_h, \\
T(s_{t+h} = X_0 \mid s_t, a_{t:t+h}^\star) &= 1 - 2\varepsilon_h, \\
T(s_{t+h} = G \mid s_t, a_{t:t+h}^\star) &= \varepsilon_h.
\end{aligned}
\tag{145}
$$

Combining this with the data distribution calculated in Equation (143), it is clear that

$$
D_{\mathrm{TV}}\big(T(s_{t+h} \mid s_t, a_{t:t+h} = 0) \,\big\|\, P_{\mathcal{D}_{\mathrm{top}}}(s_{t+h} \mid s_t, a_{t:t+h} = 0)\big) = \varepsilon_h.
\tag{146}
$$

We can repeat the same procedure to show that

$$
D_{\mathrm{TV}}\big(T(s_{t+h'} \mid s_t, a_{t:t+h'} = 0) \,\big\|\, P_{\mathcal{D}_{\mathrm{top}}}(s_{t+h'} \mid s_t, a_{t:t+h} = 0)\big) = \varepsilon_h, \quad \forall h' \in \{1, \cdots, h-1\}
\tag{147}
$$

because the only difference in these distributions is that they occupy $s_{t+h'} = X_{h'}$ with $2\varepsilon_h$ probability instead of $s_{t+h} = X_0$ with $2\varepsilon_h$ probability.

For $a_{t:t+h}^\bullet = (0, 2, 0, \cdots)$, it is clear that

$$
D_{\mathrm{TV}}\big(T(s_{t+h'} \mid s_t, a_{t:t+h} = a_{t:t+h}^\bullet) \,\big\|\, P_{\mathcal{D}_{\mathrm{top}}}(s_{t+h'} \mid s_t, a_{t:t+h} = a_{t:t+h}^\bullet)\big) = \varepsilon_h
\tag{148}
$$

holds for any $h' \in \{1, 2, \cdots, h\}$ since the only difference between these two distributions is the $\varepsilon_h$-probability path (*i.e.*, $X_0 \to C \to Z$ where the probability is under $T(\cdot \mid s_t, a_{t:t+h}^\bullet)$).

For $a_{t:t+h}^\diamond = (0, 1, 0, \cdots)$, we first compute the marginal state distributions:

$$
\begin{aligned}
P_{\mathcal{D}_{\mathrm{top}}}(s_{t+h} = Z \mid s_t, a_{t:t+h}^\diamond) &= \frac{(1 - 2\varepsilon_h)\delta_G}{2\varepsilon_h + (1 - 2\varepsilon_h)\delta_G}, \\
P_{\mathcal{D}_{\mathrm{top}}}(s_{t+h} = X_0 \mid s_t, a_{t:t+h}^\diamond) &= \frac{2\varepsilon_h}{2\varepsilon_h + (1 - 2\varepsilon_h)\delta_G}, \\
P_{\mathcal{D}_{\mathrm{top}}}(s_{t+h} = G \mid s_t, a_{t:t+h}^\diamond) &= 0. \\
P_{\mathcal{D}_{\mathrm{top}}}(s_{t+1} = X_1 \mid s_t, a_{t:t+h}^\diamond) &= \frac{(1 - 2\varepsilon_h)\delta_G}{2\varepsilon_h + (1 - 2\varepsilon_h)\delta_G}. \\
P_{\mathcal{D}_{\mathrm{top}}}(s_{t+1} = \tilde{X}_1 \mid s_t, a_{t:t+h}^\diamond) &= \frac{\varepsilon_h}{2\varepsilon_h + (1 - 2\varepsilon_h)\delta_G}. \\
P_{\mathcal{D}_{\mathrm{top}}}(s_{t+1} = C \mid s_t, a_{t:t+h}^\diamond) &= \frac{\varepsilon_h}{2\varepsilon_h + (1 - 2\varepsilon_h)\delta_G}.
\end{aligned}
\tag{149}
$$

We can also compute the open-loop marginal state distribution as follows:

$$
\begin{aligned}
T(s_{t+h} = Z \mid s_t, a_{t:t+h}^\diamond) &= 1 - 2\varepsilon_h \\
T(s_{t+h} = X_0 \mid s_t, a_{t:t+h}^\diamond) &= 2\varepsilon_h \\
T(s_{t+h} = G \mid s_t, a_{t:t+h}^\diamond) &= 0. \\
T(s_{t+1} = X_1 \mid s_t, a_{t:t+h}^\diamond) &= 1 - 2\varepsilon_h. \\
T(s_{t+1} = \tilde{X}_1 \mid s_t, a_{t:t+h}^\diamond) &= \varepsilon_h. \\
T(s_{t+1} = C \mid s_t, a_{t:t+h}^\diamond) &= \varepsilon_h.
\end{aligned}
\tag{150}
$$

Let $c_3$ be any value that satisfies $c_3 \in (0, \varepsilon_h/2)$, we can set

$$
\delta_G = \frac{\varepsilon_h(1 - 2\varepsilon_h - 2c_3)}{(\varepsilon_h + c_3)(1 - 2\varepsilon_h)},
\tag{151}
$$

such that

$$
\begin{aligned}
P_{\mathcal{D}_{\mathrm{top}}}(s_{t+h} = Z \mid s_t, a_{t:t+h}^\diamond) &= 1 - 2\varepsilon_h - 2c_3, \\
P_{\mathcal{D}_{\mathrm{top}}}(s_{t+h} = X_0 \mid s_t, a_{t:t+h}^\diamond) &= 2\varepsilon_h + 2c_3, \\
P_{\mathcal{D}_{\mathrm{top}}}(s_{t+h} = G \mid s_t, a_{t:t+h}^\diamond) &= 0, \\
P_{\mathcal{D}_{\mathrm{top}}}(s_{t+1} = X_1 \mid s_t, a_{t:t+h}^\diamond) &= 1 - 2\varepsilon_h - 2c_3, \\
P_{\mathcal{D}_{\mathrm{top}}}(s_{t+1} = \tilde{X}_1 \mid s_t, a_{t:t+h}^\diamond) &= \varepsilon_h + c_3, \\
P_{\mathcal{D}_{\mathrm{top}}}(s_{t+1} = C \mid s_t, a_{t:t+h}^\diamond) &= \varepsilon_h + c_3.
\end{aligned}
\tag{152}
$$

It is easy to check that $0 < \delta_G < 1$ (a valid probability) because in Equation (151), each term in the numerator has a larger term in the denominator (*i.e.*, $\varepsilon_h < \varepsilon_h + c_3$ and $1 - 2\varepsilon_h - 2c_3 < 1 - 2\varepsilon_h$).

Now, for all $h' \in \{1, 2, \cdots, h\}$, using the values calculated in Equations (150) and (152), we have

$$D_{\text{TV}}\big(T(s_{t+h'} \mid s_t, a_{t:t+h'} = a^\diamond_{t:t+h'}) \, \big\| \, P_{\mathcal{D}_{\text{top}}}(s_{t+h'} \mid s_t, a_{t:t+h} = a^\diamond_{t:t+h})\big) = 2c_3. \quad (153)$$

Since $c_3 < \varepsilon_h/2$, the strong open-loop consistency assumption holds for $a^\diamond_{t:t+h}$ as well.

Step 5. The optimality gap and value estimation error for the top branch:
Now we can compute the optimality gap for the estimated value for $a^\diamond_{t:t+h}$:

$$V^\star(X_0) - \hat{V}^\diamond_{\text{ac}}(X_0) = \frac{(1 - 2\varepsilon_h - 2c_3)\gamma}{(1 - \gamma)(1 - 2(\varepsilon_h + c_3)\gamma^h)}, \quad (154)$$

where the $h$-step reward sub-optimality gap is due to the agent reaching $Z$ with $(1 - 2\varepsilon_h - 2c_3)$ probability, and the $h$-step distribution gap is reflected in the $(1 - 2(\varepsilon_h + c_3)\gamma^h)$ term at bottom because the probability of reaching $X_0$ after $h$ steps is $2(\varepsilon_h + c_3)$.

Similarly, we can compute the optimality gap for $V^\star_{\text{ac}}$ and $\hat{V}^\star_{\text{ac}}$:

$$\begin{aligned} V^\star(X_0) - V^\star_{\text{ac}}(X_0) &= \varepsilon_h \frac{\gamma - \gamma^h}{1 - \gamma} + \frac{\varepsilon_h \gamma^h}{1 - \gamma} + \gamma^h(1 - 2\varepsilon_h)(V^\star - \hat{V}_{\text{ac}}) \\ &= \frac{\varepsilon_h \gamma}{(1 - \gamma)(1 - (1 - 2\varepsilon_h)\gamma^h)}, \end{aligned} \quad (155)$$

$$\begin{aligned} V^\star(X_0) - \hat{V}^\star_{\text{ac}}(X_0) &= \frac{2\varepsilon_h(\gamma - \gamma^h)}{1 - \gamma} + \frac{2\varepsilon_h \gamma^h}{1 - \gamma}\gamma^h(1 - 2\varepsilon_h)(V^\star - \hat{V}_{\text{ac}}) \\ &= \frac{2\varepsilon_h \gamma}{(1 - \gamma)(1 - (1 - 2\varepsilon_h)\gamma^h)}. \end{aligned} \quad (156)$$

Now, we observe that

$$1 - 2\varepsilon_h - 2c_3 > 1 - 3\varepsilon_h > 2\varepsilon_h, \quad (157)$$

where the first inequality is due to $c_3 \in (0, \varepsilon_h/2)$ and the second inequality is due to $\varepsilon_h \in (0, 1/5)$ in our assumption. This allows us to lower-bound the estimated optimality gap for $a^\diamond_{t:t+h}$ as follows:

$$\begin{aligned} V^\star(X_0) - \hat{V}^\diamond_{\text{ac}}(X_0) &= \frac{(1 - 2\varepsilon_h - 2c_3)\gamma}{(1 - \gamma)(1 - 2(\varepsilon_h + c_3)\gamma^h)} \\ &> \frac{2\varepsilon_h \gamma}{(1 - \gamma)(1 - (1 - 2\varepsilon_h)\gamma^h)} \\ &= V^\star(X_0) - \hat{V}^\star_{\text{ac}}(X_0), \end{aligned} \quad (158)$$

where the inequality is obtained by triggering Lemma 5 (*e.g.*, by setting $\varepsilon_1 = 2\varepsilon_h, \varepsilon_2 = (1 - 2\varepsilon_h - 2c_3), \tilde{\gamma} = \gamma^h$). The bound above rules out the possibility of $a^\diamond_{t:t+h}$ being picked by $\hat{\pi}^+_{\text{ac}}$ because it has a lower estimated value compared to $a^\star_{t:t+h}$.

Finally, for $a^\bullet_{t:t+h}$, since it is correlated with $s_{t+h} = Z$ and receives no reward except the first step in $\mathcal{D}_{\text{top}}$, the estimated value is just 1, being trivially smaller than $\hat{V}^\star_{\text{ac}}(X_0)$ and would never get picked by $\hat{\pi}^+_{\text{ac}}$. Up to now, we have finished our data distribution construction and analysis for the top branch. We summarize the key intermediate results as the remark below:

---

**Remark 1** (Intermediate results from Step 1-4)   The optimal action chunk is $a^\star_{t:t+h}$ and the estimated values for the two other possible action chunks $a^\bullet_{t:t+h}, a^\diamond_{t:t+h}$ are smaller than that of $a^\star_{t:t+h}$:

$$\hat{V}^\bullet_{\text{ac}}(X_0) < \hat{V}^\diamond_{\text{ac}}(X_0) < \hat{V}^\star_{\text{ac}}(X_0) = V^\star(X_0) - \frac{2\varepsilon_h \gamma}{(1 - \gamma)(1 - (1 - 2\varepsilon_h)\gamma^h)}. \quad (159)$$

In addition, both $\mathcal{D}_{\text{top}}$ and $\mathcal{D}^\star$ are strongly $\varepsilon_h$-open-loop consistent.

---

Next, we move on to the bottom branch.

Step 6.  Data distribution $\mathcal{D}_{\text{bottom}}$ for the bottom branch: For the bottom, we again use two policies.

*Policy $\pi_{\text{bottom}}^1$:*

$$
\pi_{\text{bottom}}^1(X_0) = \pi_{\text{bottom}}^1(G) = \pi_{\text{bottom}}^1(Z) = 1,
$$
$$
\pi_{\text{bottom}}^1(Y_1) = \pi_{\text{bottom}}^1(\tilde{Y}_1) = 2. \tag{160}
$$

$\pi_{\text{bottom}}^1$ takes $a = 1$ at $X_0$ and $G$ and $Z$, and takes $a = 2$ otherwise (at $Y_1, \tilde{Y}_1$). It is clear that this policy only produces two possible action chunks: $a_{t:t+h}^{\triangle} = (1, 1, 1, \cdots)$ or $a_{t:t+h}^{\times} = (1, 2, 1, \cdots)$.

*Policy $\pi_{\text{bottom}}^2$:*

$$
\pi_{\text{bottom}}^2(X_0) = 1,
$$
$$
\pi_{\text{bottom}}^2(a = 0 \mid Y_1) = \delta_Z,
$$
$$
\pi_{\text{bottom}}^2(a = 1 \mid Y_1) = 1 - \delta_Z, \tag{161}
$$
$$
\pi_{\text{bottom}}^2(\tilde{Y}_1) = 0,
$$
$$
\pi_{\text{bottom}}^2(Y_i) = 1, \quad \forall i \in \{2, \cdots, h - 1\},
$$

where $\delta_Z \in (0, 1)$ and we shall specify the exact value of $\delta_Z$ shortly.

$\pi_{\text{bottom}}^2$ takes $a = 1$ when it is at $Y_i$ and takes either $a = 0$ (with $\delta_Z$ probability) or $a = 1$ (with $1 - \delta_Z$ probability) when it is at $\tilde{Y}_1$. It is clear that this policy only produces two possible action chunks: $a_{t:t+h}^{\triangle} = (1, 1, 1, \cdots)$ or $a_{t:t+h}^{\circ} = (1, 0, 1, \cdots)$.

Now, we observe that the marginal state distributions for both policies conditioned on $a_{t:t+h}^{\triangle}$ are independent of $c_1$ and $\delta_Z$ because the action chunk only appears when $\pi_{\text{bottom}}^1$ reaches $G$ and when $\pi_{\text{bottom}}^2$ reaches $X_0$. More specifically,

$$
P_{\mathcal{D}_{\text{bottom}}^1}(s_{t+1} = G \mid s_t, a_{t:t+h}^{\triangle}) = P_{\mathcal{D}_{\text{bottom}}^1}(s_{t+h} = G \mid s_t, a_{t:t+h}^{\triangle}) = 1, \tag{162}
$$
$$
P_{\mathcal{D}_{\text{bottom}}^2}(s_{t+i} = X_i \mid s_t, a_{t:t+h}^{\triangle}) = P_{\mathcal{D}_{\text{bottom}}^2}(s_{t+h} = X_0 \mid s_t, a_{t:t+h}^{\triangle}) = 1, \forall i \in \{1, \cdots, h - 1\}. \tag{163}
$$

We can now mix $\mathcal{D}_{\text{bottom}}^1$ and $\mathcal{D}_{\text{bottom}}^2$ with an appropriate ratio to control the state marginals for $s_{t:t+h} = G$ and $s_{t:t+h} = X_0$ arbitrarily ($s_{t:t+h} = Z$ stays at 0 because none of the policies take/have taken $a_{t:t+h}^{\triangle}$ when they reach $Z$).

*Mixing $\pi_{\text{bottom}}^1$ and $\pi_{\text{bottom}}^2$:* Let $\mathcal{D}_{\text{bottom}}$ be a mixture of $\mathcal{D}_{\text{bottom}}^1$ and $\mathcal{D}_{\text{bottom}}^2$:

$$
P_{\mathcal{D}_{\text{bottom}}} = (1 - \vartheta)P_{\mathcal{D}_{\text{bottom}}^1} + \vartheta P_{\mathcal{D}_{\text{bottom}}^2}, \tag{164}
$$

where we set the mixing ratio to be

$$
\vartheta = \frac{c_1}{c_1 + (1 - \delta_Z)(\varepsilon_h + c_1)}. \tag{165}
$$

This mixing ratio helps the calculations to be simpler later on.

We can now compute the marginal state distribution of the mixture. We start by analyzing the action probability:

$$
P_{\mathcal{D}_{\text{bottom}}^1}(a_{t:t+h}^{\triangle} \mid s_t) = c_1,
$$
$$
P_{\mathcal{D}_{\text{bottom}}^2}(a_{t:t+h}^{\triangle} \mid s_t) = (1 - \varepsilon_h - c_1)(1 - \delta_Z). \tag{166}
$$

The state marginal is then

$$
\begin{aligned}
P_{\mathcal{D}_{\text{bottom}}}(s_{t+h} = X_0 \mid s_t, a^{\triangle}_{t:t+h}) &= \frac{P_{\mathcal{D}_{\text{bottom}}}(s_{t+h} = X_0, a^{\triangle}_{t:t+h} \mid s_t)}{P_{\mathcal{D}_{\text{bottom}}}(a^{\triangle}_{t:t+h} \mid s_t)} \\
&= \frac{\vartheta P_{\mathcal{D}^2_{\text{bottom}}}(a^{\triangle}_{t:t+h} \mid s_t)}{(1 - \vartheta) P_{\mathcal{D}^1_{\text{bottom}}}(a^{\triangle}_{t:t+h} \mid s_t) + \vartheta P_{\mathcal{D}^2_{\text{bottom}}}(a^{\triangle}_{t:t+h} \mid s_t)} \\
&= \frac{(1 - \varepsilon_h - c_1)(1 - \delta_Z)\vartheta}{c_1(1 - \vartheta) + (1 - \varepsilon_h - c_1)(1 - \delta_Z)\vartheta} \\
&= 1 - \varepsilon_h - c_1.
\end{aligned}
\tag{167}
$$

We can use it to deduce the rest of the marginals as follows:

$$
\begin{aligned}
P_{\mathcal{D}_{\text{bottom}}}(s_{t+h} = G \mid s_t, a^{\triangle}_{t:t+h}) &= \varepsilon_h + c_1, \quad \forall h' \in \{1, \cdots, h-1\}, \\
P_{\mathcal{D}_{\text{bottom}}}(s_{t+h} = X_0 \mid s_t, a^{\triangle}_{t:t+h}) &= 1 - \varepsilon_h - c_1, \\
P_{\mathcal{D}_{\text{bottom}}}(s_{t+h} = Z \mid s_t, a^{\triangle}_{t:t+h}) &= 0, \\
P_{\mathcal{D}_{\text{bottom}}}(s_{t+h'} = Y_{h'} \mid s_t, a^{\triangle}_{t:t+h}) &= 1 - \varepsilon_h - c_1, \quad \forall h' \in \{1, \cdots, h-2\}, \\
P_{\mathcal{D}_{\text{bottom}}}(s_{t+1} = \tilde{Y}_1 \mid s_t, a^{\triangle}_{t:t+h}) &= 0.
\end{aligned}
\tag{168}
$$

Up to now, we have established $\mathcal{D}_{\text{bottom}}$ and we are ready to check the strong open-loop consistency.

**Step 7. Strong $\varepsilon_h$-open-loop consistency of $\mathcal{D}_{\text{bottom}}$:**

*For $a^{\triangle}_{t:t+h} = (1, 1, \cdots)$*, we can compute the open-loop marginals as follows:

$$
\begin{aligned}
T(s_{t+h'} = G \mid s_t, a^{\triangle}_{t:t+h}) &= c_1, \quad \forall h' \in \{1, \cdots, h-1\}, \\
T(s_{t+h} = X_0 \mid s_t, a^{\triangle}_{t:t+h}) &= 1 - \varepsilon_h - c_1, \\
T(s_{t+h} = Z \mid s_t, a^{\triangle}_{t:t+h}) &= \varepsilon_h. \\
T(s_{t+h'} = Y_{h'} \mid s_t, a^{\triangle}_{t:t+h}) &= 1 - \varepsilon_h - c_1, \quad \forall h' \in \{1, \cdots, h-2\} \\
T(s_{t+1} = \tilde{Y}_1 \mid s_t, a^{\triangle}_{t:t+h}) &= \varepsilon_h.
\end{aligned}
\tag{169}
$$

Combining it with the marginals calculated in Equation (168), it is clear that for all $h' \in \{1, \cdots, h-1\}$,

$$
D_{\text{TV}}\big(T(s_{t+h'} \mid s_t, a_{t:t+h'} = a^{+}_{t:t+h'}) \,\|\, P_{\mathcal{D}_{\text{bottom}}}(s_{t+h'} \mid s_t, a_{t:t+h} = a^{+}_{t:t+h})\big) = \varepsilon_h, \tag{170}
$$

satisfying the open-loop consistency.

*For $a^{\times}_{t:t+h} = (1, 2, 1, \cdots)$*, the data and open-loop state marginals are

$$
\begin{aligned}
P_{\mathcal{D}_{\text{bottom}}}(s_{t+h} = Z \mid s_t, a^{\times}_{t:t+h}) &= 1, \\
P_{\mathcal{D}_{\text{bottom}}}(s_{t+1} = Y_1 \mid s_t, a^{\times}_{t:t+h}) &= \frac{1 - \varepsilon_h - c_1}{1 - c_1}, \\
P_{\mathcal{D}_{\text{bottom}}}(s_{t+1} = \tilde{Y}_1 \mid s_t, a^{\times}_{t:t+h}) &= \frac{\varepsilon_h}{1 - c_1}, \\
T(s_{t+h} = Z \mid s_t, a^{\times}_{t:t+h}) &= 1 - c_1, \\
T(s_{t+h} = G \mid s_t, a^{\times}_{t:t+h}) &= c_1, \\
T(s_{t+1} = Y_1 \mid s_t, a^{\times}_{t:t+h}) &= 1 - \varepsilon_h - c_1, \\
T(s_{t+1} = \tilde{Y}_1 \mid s_t, a^{\times}_{t:t+h}) &= \varepsilon_h, \\
T(s_{t+1} = G \mid s_t, a^{\times}_{t:t+h}) &= c_1.
\end{aligned}
\tag{171}
$$

This allows us to bound the TV distance for all $h' \in \{1, \cdots, h-1\}$ as

$$
D_{\text{TV}}\big(T(s_{t+h'} \mid s_t, a_{t:t+h'} = a^{\times}_{t:t+h'}) \,\|\, P_{\mathcal{D}_{\text{bottom}}}(s_{t+h'} \mid s_t, a_{t:t+h} = a^{\times}_{t:t+h})\big) \le \frac{c_1}{1 - c_1}. \tag{172}
$$

Since $c_1 < \varepsilon_h/2 < 1/10$,

$$\frac{c_1}{1 - c_1} < \frac{10}{9}c_1 < 5\varepsilon_h/9 < \varepsilon_h, \tag{173}$$

satisfying the strong open-loop consistency assumption.

*For $a_{t:t+h}^\circ = (1, 0, 1, \cdots)$*, we first compute the state marginals in $\mathcal{D}_{\text{bottom}}$ as follows:

$$
\begin{aligned}
P_{\mathcal{D}_{\text{bottom}}}(s_{t+h} = Z \mid s_t, a_{t:t+h}^\circ) &= \frac{(1 - \varepsilon_h - c_1)\delta_Z}{\varepsilon_h + (1 - \varepsilon_h - c_1)\delta_Z}, \\
P_{\mathcal{D}_{\text{bottom}}}(s_{t+h} = X_0 \mid s_t, a_{t:t+h}^\circ) &= \frac{\varepsilon_h}{\varepsilon_h + (1 - \varepsilon_h - c_1)\delta_Z}, \\
P_{\mathcal{D}_{\text{bottom}}}(s_{t+1} = Y_1 \mid s_t, a_{t:t+h}^\circ) &= \frac{(1 - \varepsilon_h - c_1)\delta_Z}{\varepsilon_h + (1 - \varepsilon_h - c_1)\delta_Z}. \\
P_{\mathcal{D}_{\text{bottom}}}(s_{t+1} = \tilde{Y}_1 \mid s_t, a_{t:t+h}^\circ) &= \frac{\varepsilon_h}{\varepsilon_h + (1 - \varepsilon_h - c_1)\delta_Z}.
\end{aligned} \tag{174}
$$

We can also compute the open-loop marginal state distribution as follows:

$$
\begin{aligned}
T(s_{t+h} = Z \mid s_t, a_{t:t+h}^\circ) &= 1 - \varepsilon_h - c_1, \\
T(s_{t+h} = X_0 \mid s_t, a_{t:t+h}^\circ) &= \varepsilon_h, \\
T(s_{t+h} = G \mid s_t, a_{t:t+h}^\circ) &= c_1, \\
T(s_{t+1} = Y_1 \mid s_t, a_{t:t+h}^\circ) &= 1 - \varepsilon_h - c_1, \\
T(s_{t+1} = \tilde{Y}_1 \mid s_t, a_{t:t+h}^\circ) &= \varepsilon_h, \\
T(s_{t+1} = G \mid s_t, a_{t:t+h}^\circ) &= c_1.
\end{aligned} \tag{175}
$$

Let $c_4 \in (c_1, \varepsilon_h)$, and we set

$$\delta_Z = \frac{\varepsilon_h(1 - \varepsilon_h - c_4)}{(\varepsilon_h + c_4)(1 - \varepsilon_h - c_1)}. \tag{176}$$

Then, we have

$$
\begin{aligned}
P_{\mathcal{D}_{\text{bottom}}}(s_{t+h} = Z \mid s_t, a_{t:t+h}^\circ) &= 1 - \varepsilon_h - c_4, \\
P_{\mathcal{D}_{\text{bottom}}}(s_{t+h} = X_0 \mid s_t, a_{t:t+h}^\circ) &= \varepsilon_h + c_4, \\
P_{\mathcal{D}_{\text{bottom}}}(s_{t+h} = G \mid s_t, a_{t:t+h}^\circ) &= 0, \\
P_{\mathcal{D}_{\text{bottom}}}(s_{t+1} = Y_1 \mid s_t, a_{t:t+h}^\circ) &= 1 - \varepsilon_h - c_4, \\
P_{\mathcal{D}_{\text{bottom}}}(s_{t+1} = \tilde{Y}_1 \mid s_t, a_{t:t+h}^\circ) &= \varepsilon_h + c_4.
\end{aligned} \tag{177}
$$

The TV distance is then

$$D_{\text{TV}}\big(T(s_{t+h'} \mid s_t, a_{t:t+h'} = a_{t:t+h'}^\circ) \,\big\|\, P_{\mathcal{D}_{\text{bottom}}}(s_{t+h'} \mid s_t, a_{t:t+h} = a_{t:t+h}^\circ)\big) = c_4. \tag{178}$$

Since $c_4 < \varepsilon_h$, the strong open-loop consistency is also satisfied for $a_{t:t+h}^\circ$.

Up to now, we have checked that all three possible action chunks in the bottom branch satisfy the strong open-loop consistency assumption. Since $\mathcal{D}_{\text{top}}$ and $\mathcal{D}_{\text{bottom}}$ have non-overlapping supports for $a_{t:t+h}$, and they are both strongly $\varepsilon_h$-open-loop consistent on their own, we can construct $\mathcal{D}$ as

$$P_{\mathcal{D}}(\cdot \mid s_t) = (1 - \varrho)P_{\mathcal{D}_{\text{top}}}(\cdot \mid s_t) + \varrho P_{\mathcal{D}_{\text{bottom}}}(\cdot \mid s_t), \tag{179}$$

for any $\varrho \in (0, 1)$, and conclude that

---

**Remark 2** (Intermediate result from Step 5-7)  $\mathcal{D}$ is strongly $\varepsilon_h$-open-loop consistent.

---

Up to now, we have constructed and checked both $\mathcal{D}$ and $\mathcal{D}^\star$ are strongly $\varepsilon_h$-open-loop consistent.

As the final step, we calculate the optimality gap and value estimation error for these action chunks.

Step 8.  The optimality gap and value estimation error for the bottom branch:

We first note that similar to $a^{\bullet}_{t:t+h}$, $a^{\times}_{t:t+h}$ is correlated with $s_{t+h} = Z$ and always receives $0$ reward except the first step in $\mathcal{D}$. Thus, the estimated value $\hat{V}^{\times}$ is just 1, being trivially smaller than $\hat{V}^{\star}_{\mathrm{ac}}$ and would never get picked by $\hat{\pi}^{\triangle}_{\mathrm{ac}}$. The only top contenders are $a^{+}_{t:t+h}$, $a^{\circ}_{t:t+h}$ and $a^{\star}_{t:t+h}$ (which we already analyzed in Step 5 above).

We start with $a^{\circ}_{t:t+h}$ where we can compute optimality gap as follows:

$$V^{\star}(X_0) - \hat{V}^{\circ}_{\mathrm{ac}}(X_0) = \frac{(1 - \varepsilon_h - c_4)\gamma + \delta(1 - \gamma) + (\varepsilon_h + c_4)\delta(\gamma - \gamma^h)}{(1 - \gamma)(1 - (\varepsilon_h + c_4)\gamma^h)}. \tag{180}$$

Now, observe that

$$\varepsilon_h + c_4 < 2\varepsilon_h < 1 - 2\varepsilon_h, \tag{181}$$

where again the last inequality comes from the fact that $\varepsilon_h < 1/4$.

We can now lower-bound the optimality gap as follows:

$$\begin{aligned}
V^{\star}(X_0) - \hat{V}^{\circ}_{\mathrm{ac}}(X_0) &> \frac{2\varepsilon_h\gamma + \delta(1 - \gamma) + (\varepsilon_h + c_4)\delta(\gamma - \gamma^h)}{(1 - \gamma)(1 - (1 - 2\varepsilon_h)\gamma^h)} \\
&> \frac{2\varepsilon_h\gamma}{(1 - \gamma)(1 - (1 - 2\varepsilon_h)\gamma^h)} \\
&= V^{\star}(X_0) - \hat{V}^{\star}_{\mathrm{ac}}(X_0).
\end{aligned} \tag{182}$$

where the first inequality is obtained by triggering Lemma 5 (*e.g.*, by setting $\varepsilon_1 = 2\varepsilon_h, \varepsilon_2 = (1 - \varepsilon_h - c_4), \tilde{\gamma} = \gamma^h$).

With this lower-bound, we can conclude that $a^{\circ}_{t:t+h}$ would not be picked by $\pi^{+}_{\mathrm{ac}}$ as well because $\hat{V}^{\circ}_{\mathrm{ac}}(X_0) < \hat{V}^{\star}_{\mathrm{ac}}(X_0)$.

Up to now, we have eliminated both $a^{\circ}_{t:t+h}$ and $a^{\times}_{t:t+h}$ (for the possibility of being picked by $\pi^{+}_{\mathrm{ac}}$) and the only remaining contender left is $a^{\triangle}_{t:t+h}$.

We can also compute the estimated and the actual values for $a_{t:t+h} = a^{\triangle}_{t:t+h} = 1$ in terms of their optimality gaps:

$$V^{\star}(X_0) - \hat{V}^{\triangle}_{\mathrm{ac}}(X_0) = \frac{\delta(1 - \varepsilon_h - c_1)\gamma}{(1 - \gamma)(1 - (1 - \varepsilon_h - c_1)\gamma^h)}, \tag{183}$$

$$V^{\star}(X_0) - V^{\triangle}_{\mathrm{ac}}(X_0) = \frac{[\delta(1 - \varepsilon_h - c_1) + \varepsilon_h]\gamma}{(1 - \gamma)(1 - (1 - \varepsilon_h - c_1)\gamma^h)}. \tag{184}$$

Let

$$\delta = \frac{2\varepsilon_h\gamma - c_2}{(1 - \varepsilon_h - c_1)\gamma} \frac{1 - (1 - \varepsilon_h - c_1)\gamma^h}{1 - (1 - 2\varepsilon_h)\gamma^h}. \tag{185}$$

We first check $1 - \delta$ is a valid reward value (within $[0, 1]$):

$$\begin{aligned}
\delta &< \frac{2\varepsilon_h}{1 - \varepsilon_h - c_1} \frac{1 - (1 - \varepsilon_h - c_1)\gamma^h}{1 - (1 - 2\varepsilon_h)\gamma^h} \\
&< \frac{2\varepsilon_h}{1 - 2\varepsilon_h} \frac{1 - (1 - 2\varepsilon_h)\gamma^h}{1 - (1 - 2\varepsilon_h)\gamma^h} \\
&= \frac{2\varepsilon_h}{1 - 2\varepsilon_h} \\
&\le 1,
\end{aligned} \tag{186}$$

where the first inequality is because $c_2 > 0$, the second inequality is due to $c_1 < \varepsilon_h$, and the final inequality is due to $\varepsilon_h < 1/4$.

It is also clear that $\delta > 0$ because all terms are positive in the fraction (Equation (185)).

Next, we substitute $\delta$ in to obtain

$$V^\star(X_0) - \hat{V}_{\mathrm{ac}}^\triangle(X_0) = \frac{2\varepsilon_h\gamma - c_2}{(1-\gamma)(1-(1-2\varepsilon_h)\gamma^h)}, \tag{187}$$

$$V^\star(X_0) - V_{\mathrm{ac}}^\triangle(X_0) = \frac{2\varepsilon_h\gamma - c_2}{(1-\gamma)(1-(1-2\varepsilon_h)\gamma^h)} + \frac{\varepsilon_h\gamma}{(1-\gamma)(1-(1-\varepsilon_h-c_1)\gamma^h)}, \tag{188}$$

where intuitively the second term in $V^\star(X_0) - V_{\mathrm{ac}}^\triangle(X_0)$ is due to the fact that from $P_{\mathcal{D}}(\cdot \mid s_t, a_{t:t+h}^\triangle)$ to $T(\cdot \mid s_t, a_{t:t+h}^\triangle)$, there is a shift in $\varepsilon_h$ probability mass from $s_{t:t+h} = (X_0, G, \cdots)$ to $s_{t:t+h} = (X_0, \tilde{Y}_1, Z, \cdots)$ incurring an additional $\frac{\varepsilon_h\gamma}{1-\gamma}$ sub-optimality in terms of the $h$-step reward, and then amplified by the value recursion by an additional factor of $\frac{1}{1-(1-\varepsilon_h-c_1)\gamma^h}$ (where $1 - \varepsilon_h - c_1$ is the probability that $a_{t:t+h}^\triangle$ reaches $X_0$ for the value recursion to occur).

Since $c_2 > 0$, we can now show that $a_{t:t+h}^\triangle$ achieves the highest estimated value among six possible action chunks:

$$V^\star - \hat{V}_{\mathrm{ac}}^\triangle < V^\star - \hat{V}_{\mathrm{ac}}^\star = \frac{2\varepsilon_h\gamma}{(1-\gamma)(1-(1-2\varepsilon_h)\gamma^h)}, \tag{189}$$

which means that $\pi_{\mathrm{ac}}^+(X_0) = a_{t:t+h}^\triangle = (1, 1, \cdots)$, or equivalently $\hat{V}_{\mathrm{ac}}^\triangle = \hat{V}_{\mathrm{ac}}^+$!

Finally, putting everything together, we have

$$V^\star(X_0) - V_{\mathrm{ac}}^+(X_0) = \frac{2\varepsilon_h\gamma - c_2}{(1-\gamma)(1-(1-2\varepsilon_h)\gamma^h)} + \frac{\varepsilon_h\gamma}{(1-\gamma)(1-(1-\varepsilon_h-c_1)\gamma^h)}, \tag{190}$$

as desired. $\qquad\square$

## H.9   PROOF OF PROPOSITION 1

> **Proposition 1** (Optimality of Closed-loop Execution of Action Chunking Policy)   Let $V^\bullet$ be the value of the one-step policy, $\pi^\bullet$, as a result of the closed-loop execution of the action chunking policy $\pi_{\mathrm{ac}}^+$ learned from $\mathcal{D}$. That is, for each $s_t \in \mathrm{supp}(P_{\mathcal{D}}(s_t))$,
>
> $$\pi^\bullet(s_t) = a_t^+, \quad \text{where } a_{t:t+h}^+ = \pi_{\mathrm{ac}}^+(s_t). \tag{17}$$
>
> If $\mathcal{D}$ and $\mathcal{D}^\star$ are both strongly $\varepsilon_h$-open-loop consistent and $\mathrm{supp}(P_{\mathcal{D}}(s_t, a_{t:t+h})) \supseteq \mathrm{supp}(P_{\mathcal{D}^\star}(s_t, a_{t:t+h}))$, then for all $s_t \in \mathrm{supp}(P_{\mathcal{D}^\star}(s_t))$,
>
> $$V^\star(s_t) - V^\bullet(s_t) \le \frac{\varepsilon_h \gamma}{(1-\gamma)^2} \left[ \frac{2}{1-(1-2\varepsilon_h)\gamma^h} + \frac{1}{1-(1-\varepsilon_h)\gamma^h} \right] \le 3\varepsilon_h H^2 \bar{H}. \tag{18}$$

*Proof.* We observe that

$$V_{\mathrm{ac}}^+(s_t) = Q_{\mathrm{ac}}^+(s_t, a_{t:t+h}^+)$$
$$\le Q^\star(s_t, a_t^+). \tag{191}$$

Combining this with Theorem 2, we get

$$Q^\star(s_t, a_t^+) \ge V^\star(s_t) - \Delta, \tag{192}$$

where $\Delta = \frac{\varepsilon_h \gamma}{1-\gamma} \left[ \frac{2}{1-(1-2\varepsilon_h)\gamma^h} + \frac{1}{1-(1-\varepsilon_h)\gamma^h} \right]$.

Now, we can bound $V^\bullet$ as follows:

$$\begin{aligned}
V^\star(s_t) - V^\bullet(s_t) &\le Q^\star(s_t, a_t^+) - Q^\bullet(s_t, a_t^+) + \Delta \\
&\le \gamma \mathbb{E}_{T(\cdot|s_t, a_t^+)} \left[ V^\star(s_{t+1}) - V^\bullet(s_{t+1}) \right] + \Delta \\
&\le \frac{\varepsilon_h \gamma}{(1-\gamma)^2} \left[ \frac{2}{1-(1-2\varepsilon_h)\gamma^h} + \frac{1}{1-(1-\varepsilon_h)\gamma^h} \right].
\end{aligned} \tag{193}$$

$\square$

## H.10   PROOF OF PROPOSITION 3

> **Proposition 3** (Comparing action chunking backup and $n$-step return backup)   Let $\mathcal{D}$ be strongly $\varepsilon_h$-open-loop consistent and $\delta_n$-sub-optimal, and $\mathrm{supp}(P_{\mathcal{D}}(s_t)) \supseteq \mathrm{supp}(P_{\mathcal{D}^\star}(s_t))$. Let $\pi_n^+ : s_t \mapsto \arg\max_{a_t} \hat{Q}_n^+(s_t, a_t)$ be the policy learned from $\mathcal{D}$, via $n$-step return backup:
>
> $$\hat{Q}_n^+(s_t, a_t) = \mathbb{E}\left[R_{t:t+n} + \gamma^n \hat{Q}_n^+(s_{t+n}, \pi_n^+(s_{t+n}))\right]. \tag{38}$$
>
> Then, for all $s_t \in \mathrm{supp}(P_{\mathcal{D}^\star}(s_t))$ (and with $\bar{H}_n = 1/(1 - \gamma^n)$),
>
> $$V_{\mathrm{ac}}^+(s_t) - \hat{V}_n^+(s_t) \geq \frac{\delta_n}{1 - \gamma^n} - \frac{\varepsilon_h \gamma}{1 - \gamma}\left[\frac{2}{1 - (1 - 2\varepsilon_h)\gamma^h} + \frac{1}{1 - (1 - \varepsilon_h)\gamma^h}\right], \tag{39}$$
> $$\geq \delta_n \bar{H}_n - 3\varepsilon_h H \bar{H}.$$

To prove Proposition 3, we first prove the following helper Lemma 6 to quantify sub-optimality for $n$-step return policy.

> **Lemma 6**   Let $Q_n^\star$ be the solution of the uncorrected $n$-step return backup equation:
>
> $$Q_n^\star(s_t, a_t) = \mathbb{E}_{P_{\mathcal{D}}(\cdot | s_t, a_t)}\left[R_{t:t+n} + \gamma^n \max_{a_{t+n}} Q_n^\star(s_{t+n}, a_{t+n})\right] \tag{194}$$
>
> The following inequality holds as long as $\mathcal{D}$ is $\delta_n$-sub-optimal:
>
> $$Q^\star(s_t, a_t) \geq Q_n^\star(s_t, a_t) + \frac{\delta_n}{1 - \gamma^n}, \forall s_t \in \mathcal{S}, a_t \in \mathcal{A} \tag{195}$$
>
> where $Q^\star$ is the Q-function of the optimal policy in $\mathcal{M}$. For the $n$-step return policy
>
> $$\pi_n^\star : s_t \mapsto \arg\max_{a_t} Q_n^\star(s_t, a_t), \tag{196}$$
>
> its corresponding value admits a similar bound:
>
> $$V^\star(s_t) \geq V_n^\star(s_t) + \frac{\delta_n}{1 - \gamma^n}, \forall s_t \tag{197}$$

*Proof.* Using the definition of sub-optimal data (Definition 4), we have

$$
\begin{aligned}
Q_n^\star(s_t, a_t) &= \mathbb{E}_{P_{\mathcal{D}}(\cdot | s_t, a_t)}\left[R_{t:t+n} + \gamma^n \max_{a_{t+n}} Q_n^\star(s_{t+n}, a_{t+n})\right] \\
&\leq Q^\star(s_t, a_t) - \delta_n + \gamma^h \mathbb{E}_{P_{\mathcal{D}}(\cdot | s_t, a_t)}\left[\max_{a_{t+n}} Q_n^\star(s_{t+n}, a_{t+n}) - V^\star(s_{t+h})\right]
\end{aligned}
\tag{198}
$$

Rearranging the inequality above yields

$$Q_n^\star(s_t, a_t) - Q^\star(s_t, a_t) \leq -\delta_n + \gamma^n \mathbb{E}_{P_{\mathcal{D}}(\cdot | s_t)}[V_n^\star(s_{t+n}) - V^\star(s_{t+n})], \forall s_t \in \mathcal{S}, a_t \in \mathcal{A} \tag{199}$$

By recursively applying the inequality above, we have

$$Q^\star(s_t, a_t) \geq Q_n^\star(s_t, a_t) + \frac{\delta_n}{1 - \gamma^n}, \forall s_t \in \mathcal{S}, a_t \in \mathcal{A} \tag{200}$$

By choosing $a_t^\star = \pi_n^\star(s_t)$, we see that

$$
\begin{aligned}
V^\star(s_t) &\geq Q^\star(s_t, a_t) \\
&\geq Q_n^\star(s_t, a_t^\star) + \frac{\delta_n}{1 - \gamma^n} \\
&= V_n^\star(s_t) + \frac{\delta_n}{1 - \gamma^n}
\end{aligned}
\tag{201}
$$

$\square$

Now we are ready to prove the main Proposition 3.

*Proof of Proposition 3.* From Lemma 6 and Theorem 2, we have

$$V_n^\star(s) + \frac{\delta_n}{1-\gamma^n} \leq V^\star(s) \leq V_{ac}^+(s) + \frac{\varepsilon_h\gamma}{1-\gamma}\left[\frac{2}{1-(1-2\varepsilon_h)\gamma^h} + \frac{1}{1-(1-\varepsilon_h)\gamma^h}\right]. \quad (202)$$

Rearranging the terms give

$$V_{ac}^+(s) - V_n^\star(s) \geq \frac{\delta_n}{1-\gamma^n} - \frac{\varepsilon_h\gamma}{1-\gamma}\left[\frac{2}{1-(1-2\varepsilon_h)\gamma^h} + \frac{1}{1-(1-\varepsilon_h)\gamma^h}\right]. \quad (203)$$

$\square$

## H.11    PROOF OF THEOREM 7

**Theorem 7** (Closed-loop Execution in the Absence of Stochastic Shortcuts)    $\mathcal{D}$ is $\alpha$-open-loop mixed and $\mathcal{M}$ is free of $\vartheta_h$-stochastic shortcut, the value ($V^\bullet$) of the one-step policy ($\pi^\bullet$) as a result of the closed-loop execution of the action chunking policy $\pi_{ac}^+$ learned from $\mathcal{D}$ admits the following bound for all $s_t \in \text{supp}(P_{\mathcal{D}^\star}(s_t))$:

$$V^\star(s_t) - V^\bullet(s_t) \leq \frac{\alpha}{(1-\gamma)^2(1-\gamma^h(1-\alpha))} + \frac{\vartheta_h\gamma^h}{(1-\gamma)(1-\gamma^h)}. \quad (47)$$

Before we start proving the main theorem, we first prove the following Lemma, which shows that the overestimation of $\hat{V}_{ac}^+$ is bounded when the advantage of the stochastic shortcut (as defined in Definition 7) is bounded.

**Lemma 7** (Lack of stochastic shortcut bounds overestimation)    If $\mathcal{M}$ is free of $\vartheta_h$-advantageous stochastic shortcuts for a horizon $h$, then

$$V_{ac}^+(s_t) - V^\star(s_t) \leq \frac{\vartheta_h}{1-\gamma^h}, \quad (204)$$

where $V_{ac}^+$ is the value function of the action chunking policy learned from any data distribution $\mathcal{D}$ from $\mathcal{M}$ and $V^\star$ is the optimal value function in $\mathcal{M}$.

*Proof.*

$$\hat{V}_{ac}^+(s_t) - V^\star(s_t) = \mathbb{E}_{P_{\mathcal{D}}}\left[R_{t:t+h} + \gamma^h\hat{V}_{ac}^+(s_{t+h})\right] - V^\star$$

$$\leq \mathbb{E}_{P_{\mathcal{D}}}\left[\gamma^h(\hat{V}_{ac}^+(s_{t+h}) - V^\star(s_{t+h}))\right] + \vartheta_h \quad (205)$$

$$\leq \frac{\vartheta_h}{1-\gamma^h}.$$

$\square$

**Lemma 8** (Monotonicity of optimality)    Let $\mathcal{D}^\circ$ be any data distribution that is collected by an open-loop policy. Then, for all $s_t, a_{t:t+h}$,

$$V^\star(s_t) \geq \mathbb{E}_{s_{t+1}\sim T(\cdot|s_t,a_t)}\left[r_t + \gamma V^\star(s_{t+1})\right] \geq \mathbb{E}_{P_{\mathcal{D}^\circ}(\cdot|s_t,a_{t:t+h})}\left[R_{t:t+h} + \gamma^h V^\star(s_{t+h})\right]$$

$$(206)$$

*Proof.* The first inequality is clear from the definition of $Q^\star(s_t, a_t)$ and $V^\star(s_t)$:

$$Q^\star(s_t, a_t) := \mathbb{E}_{s_{t+1}\sim T(\cdot|s_t,a_t)}\left[r_t + \gamma V^\star(s_{t+1})\right] \quad (207)$$

$$V^\star(s_t) := \max_{a_t^\star} Q^\star(s_t, a_t^\star) \geq Q^\star(s_t, a_t) \quad (208)$$

For the second inequality, we observe that

$$
\mathbb{E}_{P_{\mathcal{D}^\circ}(\cdot|s_t,a_{t:t+h-1})} \left[ R_{t:t+h-1} + \gamma^{h-1} V^\star(s_{t+h-1}) \right]
$$

$$
= \mathbb{E}_{P_{\mathcal{D}^\circ}(\cdot|s_t,a_{t:t+h-1})} \left[ R_{t:t+h-1} + \gamma^{h-1} \max_{a_{t+h-1}^\star} Q^\star(s_{t+h-1}, a_{t+h-1}^\star) \right]
$$

$$
\geq \mathbb{E}_{P_{\mathcal{D}^\circ}(\cdot|s_t,a_{t:t+h-1})} \left[ R_{t:t+h-1} + \gamma^{h-1} Q^\star(s_{t+h-1}, a_{t+h-1}) \right] \tag{209}
$$

$$
= \mathbb{E}_{P_{\mathcal{D}^\circ}(\cdot|s_t,a_{t:t+h})} \left[ R_{t:t+h-1} + \gamma^{h-1} \left[ r(s_{t+h-1}, a_{t+h-1}) + \gamma V^\star(s_{t+h}) \right] \right]
$$

$$
= \mathbb{E}_{P_{\mathcal{D}^\circ}(\cdot|s_t,a_{t:t+h})} \left[ R_{t:t+h} + \gamma^h V^\star(s_{t+h}) \right]
$$

By induction,

$$
\mathbb{E}_{T(\cdot|s_t,a_t)} \left[ r_t + \gamma V^\star(s_{t+1}) \right] \geq \mathbb{E}_{P_{\mathcal{D}^\circ}(\cdot|s_t,a_{t:t+h})} \left[ R_{t:t+h} + \gamma^h V^\star(s_{t+h}) \right], \tag{210}
$$

as desired. $\square$

With these two lemmata, we are now ready to present our main proof.

*Proof of Theorem 7.* The key idea of our proof is to analyze the action chunk taken by the policy $\pi_{\mathrm{ac}}^+$ (*i.e.,* $\pi_{\mathrm{ac}}^+ : s_t \mapsto \arg\max_a \hat{Q}(s_t, a_{t:t+h})$). Due to our construction of $\mathcal{D}$, the action chunk learned by $\pi_{\mathrm{ac}}^+$ either comes from $\mathcal{D}^\star$ or $\mathcal{D}^\circ$. We can $\mathcal{D}^\circ$ express as the aggregation of two open-loop data distributions, $\mathcal{D}_{\mathrm{in}}^\circ$ and $\mathcal{D}_{\mathrm{out}}^\circ$:

$$
P_{\mathcal{D}^\circ}(\cdot \mid s_t) := \hat{\alpha} P_{\mathcal{D}_{\mathrm{in}}^\circ}(\cdot \mid s_t) + (1 - \hat{\alpha}) P_{\mathcal{D}_{\mathrm{out}}^\circ}(\cdot \mid s_t) \tag{211}
$$

where $\mathrm{supp}(P_{\mathcal{D}_{\mathrm{out}}^\circ}(a_{t:t+h} \mid s_t)) \cap \mathrm{supp}(P_{\mathcal{D}^\star}(a_{t:t+h} \mid s_t)) = \varnothing$ and

$$
\hat{\alpha} \leq \frac{\alpha\beta}{(1-\alpha)(1-\beta)}. \tag{212}
$$

**Case 1, from $\mathcal{D}^\star$:** If $\pi_{\mathrm{ac}}^+(\cdot \mid s_t) \cap \mathrm{supp}(P_{\mathcal{D}^\circ}(a_{t:t+h} \mid s_t)) = \varnothing$, then we know that

$$
a_{t:t+h}^\diamond \in \mathrm{supp}(P_{\mathcal{D}^\star}(a_{t:t+h} \mid s_t)), \tag{213}
$$

for any $a_{t:t+h}^\diamond \in \pi_{\mathrm{ac}}^+$ Thus, the closed-loop execution policy $\pi^\bullet$ takes the optimal action at $s_t$. This leads to the following equalities:

$$
V^\star(s_t) - V^\bullet(s_t) = V^\star(s_t) - Q^\bullet(s_t, a_t^\diamond)
$$

$$
= \gamma \mathbb{E}_{T(\cdot|s_t,a_t^\diamond)} \left[ (V^\star(s_{t+1}) - V^\bullet(s_{t+1})) \right] \tag{214}
$$

$$
\hat{V}_{\mathrm{ac}}^+(s_t) - V^\star(s_t) = \mathbb{E}_{P_{\mathcal{D}^\star}} \left[ \hat{Q}_{\mathrm{ac}}^+(s_t, a_{t:t+h}^\diamond) \right] - V^\star(s_t)
$$

$$
= \gamma^h \mathbb{E}_{P_{\mathcal{D}^\star}} \left[ \hat{V}_{\mathrm{ac}}^+(s_{t+h}) - V^\star(s_{t+h}) \right] \tag{215}
$$

**Case 2, from $\mathcal{D}^\circ$:** If $\pi_{\mathrm{ac}}^+(\cdot \mid s_t) \cap \mathrm{supp}(P_{\mathcal{D}^\circ}(a_{t:t+h} \mid s_t)) \neq \varnothing$, then there exists $a_{t:t+h}^\circ \in \pi_{\mathrm{ac}}^+(\cdot \mid s_t)$ such that

$$
a_{t:t+h}^\circ \in \mathrm{supp}(P_{\mathcal{D}^\circ}(a_{t:t+h} \mid s_t)). \tag{216}
$$

For any $a_{t:t+h}^+ \in \pi_{\mathrm{ac}}^+(\cdot \mid s_t)$ such that $a_{t:t+h}^+ \notin \mathrm{supp}(P_{\mathcal{D}^\circ}(a_{t:t+h} \mid s_t))$, we know from the previous case that $V^\star(s_t) - Q^\bullet(s_t, a_t^+) = \gamma \mathbb{E}_{T(\cdot|s_t,a_t^\diamond)} \left[ (V^\star(s_{t+1}) - V^\bullet(s_{t+1})) \right]$. We are left with analyzing $V^\star - Q^\bullet(s_t, a_t^\circ)$ for the second case.

Since $P_{\mathcal{D}}(\cdot \mid s_t) = \beta P_{\mathcal{D}^\star}(\cdot \mid s_t) + (1 - \beta)\hat{\alpha} P_{\mathcal{D}_{\mathrm{in}}^\circ}(\cdot \mid s_t) + (1 - \beta)(1 - \hat{\alpha}) P_{\mathcal{D}_{\mathrm{out}}^\circ}(\cdot \mid s_t)$ with $\mathrm{supp}(P_{\mathcal{D}_{\mathrm{out}}^\circ}(a_{t:t+h} \mid s_t)) \cap \mathrm{supp}(P_{\mathcal{D}^\star}(a_{t:t+h} \mid s_t)) = \varnothing$, we can isolate the contribution of $\mathcal{D}_{\mathrm{out}}^\circ$ in $\mathcal{D}^\circ$ by an event $I$ that is 1 when $a_{t:t+h} \in \mathrm{supp}(P_{\mathcal{D}^\star}(a_{t:t+h} \mid s_t))$ and 0 otherwise. Now, we can remove $\mathcal{D}_{\mathrm{out}}^\circ$ when conditioned on $I = 1$ as follows:

$$
P_{\mathcal{D}}(\cdot \mid s_t, I = 1) = \frac{\beta}{\beta + (1 - \beta)\hat{\alpha}} P_{\mathcal{D}^\star}(\cdot \mid s_t, I = 1) + \frac{(1 - \beta)\hat{\alpha}}{\beta + (1 - \beta)\hat{\alpha}} P_{\mathcal{D}^\circ}(\cdot \mid s_t, I = 1)
$$

$$
= (1 - \bar{\alpha}) P_{\mathcal{D}^\star}(\cdot \mid s_t, I = 1) + \bar{\alpha} P_{\mathcal{D}^\circ}(\cdot \mid s_t, I = 1) \tag{217}
$$

Since $\hat{\alpha} \leq \frac{\alpha\beta}{(1-\alpha)(1-\beta)}$, with some algebraic manipulation, we can obtain $\bar{\alpha} \leq \alpha$.

By Lemma 1, there exists $a^{\diamond}_{t:t+h} \in \mathrm{supp}(P_{\mathcal{D}^{\star}}(a_{t:t+h} \mid s_t))$ such that

$$P_{\mathcal{D}}(\cdot \mid s_t, a^{\diamond}_{t:t+h}) = (1 - \tilde{\alpha})P_{\mathcal{D}^{\star}}(\cdot \mid s_t, a^{\diamond}_{t:t+h}) + \tilde{\alpha}P_{\mathcal{D}^{\circ}}(\cdot \mid s_t, a^{\diamond}_{t:t+h}) \tag{218}$$

with $\tilde{\alpha} \leq \bar{\alpha} \leq \alpha$.

We are now ready to bound $\hat{V}^{+}_{\mathrm{ac}}$ as follows:

$$
\begin{aligned}
\hat{V}^{+}_{\mathrm{ac}}(s_t) - V^{\star}(s_t) &\geq \hat{Q}^{+}_{\mathrm{ac}}(s_t, a^{\diamond}_{t:t+h}) - V^{\star}(s_t) \\
&= (1 - \tilde{\alpha})\mathbb{E}_{P_{\mathcal{D}^{\star}}}\left[\hat{Q}^{+}_{\mathrm{ac}}(s_t, a^{\diamond}_{t:t+h})\right] + \tilde{\alpha}\mathbb{E}_{P_{\mathcal{D}^{\circ}}}\left[\hat{Q}^{+}_{\mathrm{ac}}(s_t, a^{\diamond}_{t:t+h})\right] - V^{\star}(s_t) \\
&\geq (1 - \alpha)\mathbb{E}_{P_{\mathcal{D}^{\star}}}\left[\gamma^h(\hat{V}^{+}_{\mathrm{ac}}(s_{t+h}) - V^{\star}(s_{t+h}))\right] - \alpha V^{\star}(s_t) \\
&\geq (1 - \alpha)\gamma^h\mathbb{E}_{P_{\mathcal{D}^{\star}}}\left[\hat{V}^{+}_{\mathrm{ac}}(s_{t+h}) - V^{\star}(s_{t+h})\right] - \frac{\alpha}{1 - \gamma}
\end{aligned}
\tag{219}
$$

where we drop the term $\tilde{\alpha}\mathbb{E}_{\mathcal{D}^{\circ}}\left[\hat{Q}^{+}_{\mathrm{ac}}(s_t, a^{\diamond}_{t:t+h})\right] \geq 0$ for the second inequality.

By combining Equation (219) (when $\pi^{+}_{\mathrm{ac}}(\cdot \mid s_t) \cap \mathrm{supp}(P_{\mathcal{D}^{\circ}}(\cdot \mid s_t)) \neq \varnothing$) with Equation (215))(when $\pi^{+}_{\mathrm{ac}}(\cdot \mid s_t) \cap \mathrm{supp}(P_{\mathcal{D}^{\circ}}(\cdot \mid s_t)) = \varnothing$), we can now recursively bound $\hat{V}^{+}_{\mathrm{ac}}(s_t) - V^{\star}(s_t)$ as follows:

$$\hat{V}^{+}_{\mathrm{ac}}(s_t) - V^{\star}(s_t) \geq -\frac{\alpha}{(1-\gamma)(1-\gamma^h(1-\alpha))}. \tag{220}$$

Combining this with the result from Lemma 7, we have both a lower-bound and an upper-bound on $\hat{V}^{+}_{\mathrm{ac}}$ in terms of $V^{\star}$:

$$V^{\star}(s_t) - C_{\alpha} \leq \hat{V}^{+}_{\mathrm{ac}}(s_t) \leq V^{\star}(s_t) + C_{\vartheta}. \tag{221}$$

where $C_{\alpha} := \frac{\alpha}{(1-\gamma)(1-\gamma^h(1-\alpha))}, C_{\vartheta} := \frac{\vartheta_h}{1-\gamma^h}$.

Now, we are ready to bound $V^{\star} - Q^{\bullet}(s_t, a^{\circ}_t)$ for the second case (when $\pi^{+}_{\mathrm{ac}}(\cdot \mid s_t) \cap \mathrm{supp}(P_{\mathcal{D}^{\circ}}(\cdot \mid s_t)) \neq \varnothing$). For notation convenience, for the following equation, we use $P_{\mathcal{D}^{\circ}}$ as an abbreviation for $P_{\mathcal{D}^{\circ}}(\cdot \mid s_t, a^{\circ}_{t:t+h})$.

$$
\begin{aligned}
&V^{\star}(s_t) - Q^{\bullet}(s_t, a^{\circ}_t) \\
&= V^{\star} - \mathbb{E}_{P_{\mathcal{D}^{\circ}}}\left[r_t + \gamma V^{\bullet}(s_{t+1})\right] \\
&\leq V^{\star} - \mathbb{E}_{P_{\mathcal{D}^{\circ}}}\left[R_{t:t+h} + \gamma^h V^{\star}(s_{t+h}) - \gamma(V^{\star}(s_{t+1}) - V^{\bullet}(s_{t+1}))\right] \\
&\leq \hat{V}^{+}_{\mathrm{ac}}(s_t) + C_{\alpha} - \mathbb{E}_{P_{\mathcal{D}^{\circ}}}\left[R_{t:t+h} + \gamma^h(\hat{V}^{+}_{\mathrm{ac}}(s_{t+h}) - C_{\vartheta}) - \gamma(V^{\star}(s_{t+1}) - V^{\bullet}(s_{t+1}))\right] \\
&= \hat{V}^{+}_{\mathrm{ac}}(s_t) - \mathbb{E}_{P_{\mathcal{D}^{\circ}}}\left[R_{t:t+h} + \gamma^h(\hat{V}^{+}_{\mathrm{ac}}(s_{t+h}))\right] + C_{\alpha} + \gamma^h C_{\vartheta} + \mathbb{E}_{T(\cdot|s_t,a^{\circ}_t)}\left[\gamma(V^{\star}(s_{t+1}) - V^{\bullet}(s_{t+1}))\right] \\
&= C_{\alpha} + \gamma^h C_{\vartheta} + \gamma\mathbb{E}_{T(\cdot|s_t,a^{\circ}_t)}[V^{\star}(s_{t+1}) - V^{\bullet}(s_{t+1})],
\end{aligned}
\tag{222}
$$

where the first inequality uses Lemma 8,

$$\mathbb{E}_{T(\cdot|s_t,a^{\circ}_t)}\left[r_t + \gamma V^{\star}(s_{t+1})\right] \geq \mathbb{E}_{P_{\mathcal{D}^{\circ}}(\cdot|s_t,a^{\circ}_{t:t+h})}[R_{t:t+h} + \gamma^h V^{\star}(s_{t+h})], \tag{223}$$

and the second inequality uses the lowerbound for $\hat{V}^{+}_{\mathrm{ac}}(s_t)$ and the upperbound for $\hat{V}^{+}_{\mathrm{ac}}(s_{t+h})$.

Combining the bounds of $V^{\star} - Q^{\bullet}(s_t, a^{\circ}_t)$ for both cases (Equation (214) and Equation (222)), we have

$$
\begin{aligned}
V^{\star} - V^{\bullet}(s_t) &\leq \max\Big(\gamma\mathbb{E}_{T(\cdot|s_t,a^{\circ}_t)}\left[(V^{\star}(s_{t+1}) - V^{\bullet}(s_{t+1}))\right], \\
&\qquad\qquad C_{\alpha} + \gamma^h C_{\vartheta} + \gamma\mathbb{E}_{T(\cdot|s_t,a^{\circ}_t)}[V^{\star}(s_{t+1}) - V^{\bullet}(s_{t+1})]\Big) \\
&= C_{\alpha} + \gamma^h C_{\vartheta} + \gamma\mathbb{E}_{T(\cdot|s_t,a^{\circ}_t)}[V^{\star}(s_{t+1}) - V^{\bullet}(s_{t+1})] \\
&\leq \frac{C_{\alpha} + \gamma^h C_{\vartheta}}{1 - \gamma} \\
&= \frac{\alpha}{(1-\gamma)^2(1-\gamma^h(1-\alpha))} + \frac{\vartheta\gamma^h}{(1-\gamma)(1-\gamma^h)}.
\end{aligned}
\tag{224}
$$

$\square$

### H.12 PROOF THEOREM 3

> **Theorem 3** (Closed-loop AC Policy under Bounded OV)   Let $\mathcal{D}^\star$ be the data distribution collected by an optimal policy. Assume $\mathcal{D}$ can be decomposed into a mixture of data distributions $\{\mathcal{D}^\star, \mathcal{D}_1, \mathcal{D}_2, \cdots \mathcal{D}_M\}$ such that each data distribution component satisfies Assumption 1 and for some $\vartheta_h^L, \vartheta_h^G \geq 0$, they satisfy the following two conditions:
>
> **1. Locally bounded optimality variability condition**: every $\mathcal{D}_i$ (including $\mathcal{D}^\star$) exhibits $\vartheta_h^L$-bounded variability in optimality conditioned on $s_t, a_t$ for all $(s_t, a_t) \in \mathrm{supp}(P_{\mathcal{D}_i}(s_t, a_t))$, and
>
> **2. Globally bounded optimality variability condition**: $\mathcal{D}$ as a whole exhibits $\vartheta_h^G$-variability in optimality conditioned on $s_t, a_{t:t+h}$ for all $(s_t, a_{t:t+h}) \in \mathrm{supp}(P_{\mathcal{D}}(s_t, a_{t:t+h}))$.
>
> Then for all $s_t \in \mathrm{supp}(P_{\mathcal{D}^\star}(s_t))$,
>
> $$V^\star(s_t) - V^\bullet(s_t) \leq \frac{\vartheta_h^L}{1-\gamma} + \frac{\vartheta_h^G + \gamma^h \min(\vartheta_h^L, \vartheta_h^G)}{(1-\gamma)(1-\gamma^h)} \leq \vartheta_h^L H + 2\vartheta_h^G H \bar{H}. \tag{20}$$

*Proof of Theorem 3.*   Consider any $s_t \in \mathrm{supp}(P_{\mathcal{D}^\star}(s_t))$. Let $a_{t:t+h}^+ = \pi_{\mathrm{ac}}^+(s_t)$ and

$$a_{t:t+h}^\circ := \arg\max_{a_{t:t+h} \in \mathrm{supp}(P_{\mathcal{D}^\star}(a_{t:t+h}|s_t))} \left[ \mathbb{E}_{P_{\mathcal{D}^\star}(\cdot|s_t, a_{t:t+h})} \left[ R_{t:t+h} + \gamma^h V^\star(s_{t+h}) \right] \right]. \tag{225}$$

We first observe that

$$\mathbb{E}_{P_{\mathcal{D}^\star}(\cdot|s_t, a_{t:t+h}^\circ)} \left[ R_{t:t+h} + \gamma^h V^\star(s_{t+h}) \right] \geq V^\star(s_t), \tag{226}$$

because

$$V^\star(s_t) = \mathbb{E}_{a_{t:t+h} \sim P_{\mathcal{D}^\star}(\cdot|s_t)} \left[ \mathbb{E}_{P_{\mathcal{D}^\star}(\cdot|s_t, a_{t:t+h})} \left[ R_{t:t+h} + \gamma^h V^\star(s_{t+h}) \right] \right], \tag{227}$$

and the maximum value of a random variable is no less than its expectation.

Let

$$\tilde{Q}_{\min}(s_t, a_{t:t+h}^\circ) := \min_{\mathrm{supp}(P_{\mathcal{D}}(\cdot|s_t, a_{t:t+h}^\circ))} \left[ R_{t:t+h} + V^\star(s_{t+h}) \right], \tag{228}$$

$$\tilde{Q}_{\max}(s_t, a_{t:t+h}^\circ) := \max_{\mathrm{supp}(P_{\mathcal{D}}(\cdot|s_t, a_{t:t+h}^\circ))} \left[ R_{t:t+h} + V^\star(s_{t+h}) \right]. \tag{229}$$

Since $\mathcal{D}$ exhibits $\vartheta_h^G$-variability in optimality, we have

$$\tilde{Q}_{\min}(s_t, a_{t:t+h}^\circ) \geq \tilde{Q}_{\max}(s_t, a_{t:t+h}^\circ) - \vartheta_h^G. \tag{230}$$

$$
\begin{aligned}
&V^\star(s_t) - Q^\star(s_t, a_t^+) \\
&= V^\star(s_t) - \hat{V}_{\mathrm{ac}}^+(s_t) + \hat{V}_{\mathrm{ac}}^+(s_t) - Q^\star(s_t, a_t^+) \\
&= V^\star(s_t) - \hat{Q}_{\mathrm{ac}}^+(s_t, a_{t:t+h}^+) + \hat{Q}_{\mathrm{ac}}^+(s_t, a_{t:t+h}^+) - Q^\star(s_t, a_t^+) \\
&\leq V^\star(s_t) - \hat{Q}_{\mathrm{ac}}^+(s_t, a_{t:t+h}^\circ) + \vartheta_h^L + \gamma^h \mathbb{E}_{P_{\mathcal{D}}(\cdot|s_t, a_{t:t+h}^+)} \left[ \hat{V}_{\mathrm{ac}}^+(s_{t+h}) - V^\star(s_{t+h}) \right] \\
&= V^\star(s_t) - \hat{Q}_{\mathrm{ac}}^+(s_t, a_{t:t+h}^\circ) + \vartheta_h^L + \gamma^h \mathbb{E}_{P_{\mathcal{D}}(\cdot|s_t, a_{t:t+h}^+)} \left[ \hat{V}_{\mathrm{ac}}^+(s_{t+h}) - Q^\star(s_{t+h}, a_{t+h}^+) \right] - \\
&\quad \gamma^h \mathbb{E}_{P_{\mathcal{D}}(\cdot|s_t, a_{t:t+h}^+)} \left[ V^\star(s_{t+h}) - Q^\star(s_{t+h}, a_{t+h}^+) \right].
\end{aligned}
\tag{231}
$$

We can use it to lower-bound $\hat{V}_{\text{ac}}^+(s_t)$ as follows:

$$
\begin{aligned}
\hat{V}_{\text{ac}}^+(s_t) &= \hat{Q}_{\text{ac}}^+(s_t, a_{t:t+h}^+) \\
&\geq \hat{Q}_{\text{ac}}^+(s_t, a_{t:t+h}^\circ) \\
&= \mathbb{E}_{P_{\mathcal{D}}(\cdot|s_t, a_{t:t+h}^\circ)}\left[ R_{t:t+h} + \gamma^h \hat{V}_{\text{ac}}^+(s_{t+h}) \right] \\
&= \mathbb{E}_{P_{\mathcal{D}}(\cdot|s_t, a_{t:t+h}^\circ)}\left[ R_{t:t+h} + \gamma^h V^\star(s_{t+h}) \right] + \mathbb{E}_{P_{\mathcal{D}}(\cdot|s_t, a_{t:t+h}^\circ)}\left[ \gamma^h (\hat{V}_{\text{ac}}^+(s_{t+h}) - V^\star(s_{t+h})) \right] \\
&\geq \tilde{Q}_{\min}(s_t, a_{t:t+h}^\circ) + \mathbb{E}_{P_{\mathcal{D}}(\cdot|s_t, a_{t:t+h}^\circ)}\left[ \gamma^h (\hat{V}_{\text{ac}}^+(s_{t+h}) - V^\star(s_{t+h})) \right] \\
&\geq \tilde{Q}_{\max}(s_t, a_{t:t+h}^\circ) - \vartheta_h^G + \mathbb{E}_{P_{\mathcal{D}}(\cdot|s_t, a_{t:t+h}^\circ)}\left[ \gamma^h (\hat{V}_{\text{ac}}^+(s_{t+h}) - V^\star(s_{t+h})) \right] \\
&\geq \mathbb{E}_{P_{\mathcal{D}^\star}(\cdot|s_t, a_{t:t+h}^\circ)}\left[ R_{t:t+h} + \gamma^h V^\star(s_{t+h}) \right] - \vartheta_h^G + \gamma^h \mathbb{E}_{P_{\mathcal{D}}(\cdot|s_t, a_{t:t+h}^\circ)}\left[ (\hat{V}_{\text{ac}}^+(s_{t+h}) - V^\star(s_{t+h})) \right] \\
&\geq V^\star(s_t) - \vartheta_h^G + \gamma^h \mathbb{E}_{P_{\mathcal{D}}(\cdot|s_t, a_{t:t+h}^\circ)}\left[ (\hat{V}_{\text{ac}}^+(s_{t+h}) - V^\star(s_{t+h})) \right] \\
&\geq V^\star(s_t) - \frac{\vartheta_h^G}{1 - \gamma^h}.
\end{aligned}
\tag{232}
$$

Let $\mathbb{M}^+ = \{\tilde{\mathcal{D}}_1, \cdots \tilde{\mathcal{D}}_{M^+}\}$ be all data distributions from $\{\mathcal{D}^\star, \mathcal{D}_1, \mathcal{D}_2, \cdots, \mathcal{D}_M\}$ where $(s_t, a_{t:t+h}^+)$ is in the support. Let $\tilde{\mathcal{D}}^+$ be any mixture of $\mathbb{M}$ where each mixture component has non-zero weight:

$$
P_{\tilde{\mathcal{D}}^+} = \sum_{i=1}^{M} w_i P_{\tilde{\mathcal{D}}_i},
\tag{233}
$$

where $w_i > 0, \sum_i w_i = 1$.

Let

$$
\tilde{Q}_{\min}^\star(s_t, a_t) := \min_{\text{supp}(P_{\mathcal{D}^\star}(\cdot|s_t, a_t))} \left[ R_{t:t+h} + V^\star(s_{t+h}) \right],
\tag{234}
$$

$$
\tilde{Q}_{\max}^\star(s_t, a_t) := \max_{\text{supp}(P_{\mathcal{D}^\star}(\cdot|s_t, a_t))} \left[ R_{t:t+h} + V^\star(s_{t+h}) \right],
\tag{235}
$$

$$
\tilde{Q}_{\min}^i(s_t, a_t) := \min_{\text{supp}(P_{\mathcal{D}^i}(\cdot|s_t, a_t))} \left[ R_{t:t+h} + V^\star(s_{t+h}) \right],
\tag{236}
$$

$$
\tilde{Q}_{\max}^i(s_t, a_t) := \max_{\text{supp}(P_{\mathcal{D}^i}(\cdot|s_t, a_t))} \left[ R_{t:t+h} + V^\star(s_{t+h}) \right],
\tag{237}
$$

$$
\tilde{Q}_{\max}^+(s_t, a_t^+) := \max_{\text{supp}(P_{\tilde{\mathcal{D}}^+}(\cdot|s_t, a_t^+))} \left[ R_{t:t+h} + V^\star(s_{t+h}) \right],
\tag{238}
$$

$$
\tilde{Q}_{\max}^+(s_t, a_{t:t+h}^+) := \max_{\text{supp}(P_{\tilde{\mathcal{D}}^+}(\cdot|s_t, a_{t:t+h}^+))} \left[ R_{t:t+h} + V^\star(s_{t+h}) \right].
\tag{239}
$$

The minimum and the maximum is over the remaining trajectory conditioned on $s_t, a_t$ or $s_t, a_{t:t+h}$ that is still in the support of the corresponding data distribution.

From the $\vartheta_h^L$-bounded variability in optimality and the Assumption 1 of each data mixture, we observe that

$$
Q^\star(s_t, a_t) \geq \tilde{Q}_{\min}^i(s_t, a_t) \geq \tilde{Q}_{\max}^i(s_t, a_t) - \vartheta_h^L, \quad \forall i \in \{1, 2, \cdots, M\}
\tag{240}
$$

$$
Q^\star(s_t, a_t) \geq \tilde{Q}_{\min}^\star(s_t, a_t) \geq \tilde{Q}_{\max}^\star(s_t, a_t) - \vartheta_h^L.
\tag{241}
$$

We can then derive that

$$
\begin{aligned}
\tilde{Q}_{\max}^+(s_t, a_t^+) &= \max(\tilde{Q}_{\max}^\star(s_t, a_t^+), \tilde{Q}_{\max}^1(s_t, a_t^+), \cdots, \tilde{Q}_{\max}^M(s_t, a_t^+)) \\
&\leq Q^\star(s_t, a_t) + \vartheta_h^L.
\end{aligned}
\tag{242}
$$

With this, we can now upper-bound $\hat{V}_{\text{ac}}^+(s_t)$ as follows:

$$
\begin{aligned}
\hat{V}_{\text{ac}}^+(s_t) &= \hat{Q}_{\text{ac}}^+(s_t, a_{t:t+h}^+) \\
&= \mathbb{E}_{P_{\mathcal{D}}(\cdot|s_t, a_{t:t+h}^+)} \left[ R_{t:t+h} + \gamma^h \hat{V}_{\text{ac}}^+(s_{t+h}) \right] \\
&= \mathbb{E}_{P_{\tilde{\mathcal{D}}^+}(\cdot|s_t, a_{t:t+h}^+)} \left[ R_{t:t+h} + \gamma^h \hat{V}_{\text{ac}}^+(s_{t+h}) \right] \\
&= \mathbb{E}_{P_{\tilde{\mathcal{D}}^+}(\cdot|s_t, a_{t:t+h}^+)} \left[ R_{t:t+h} + \gamma^h V^\star(s_{t+h}) \right] + \gamma^h \mathbb{E}_{P_{\tilde{\mathcal{D}}^+}(\cdot|s_t, a_{t:t+h}^+)} \left[ \hat{V}_{\text{ac}}^+(s_{t+h}) - V^\star(s_{t+h}) \right] \\
&\leq \tilde{Q}_{\max}^+(s_t, a_{t:t+h}^+) + \gamma^h \mathbb{E}_{P_{\mathcal{D}}(\cdot|s_t, a_{t:t+h}^+)} \left[ \hat{V}_{\text{ac}}^+(s_{t+h}) - V^\star(s_{t+h}) \right] \\
&\leq \tilde{Q}_{\max}^+(s_t, a_t^+) + \gamma^h \mathbb{E}_{P_{\mathcal{D}}(\cdot|s_t, a_{t:t+h}^+)} \left[ \hat{V}_{\text{ac}}^+(s_{t+h}) - V^\star(s_{t+h}) \right] \\
&\leq Q^\star(s_t, a_t^+) + \vartheta_h^L + \gamma^h \mathbb{E}_{P_{\mathcal{D}}(\cdot|s_t, a_{t:t+h}^+)} \left[ \hat{V}_{\text{ac}}^+(s_{t+h}) - V^\star(s_{t+h}) \right].
\end{aligned}
\tag{243}
$$

Let

$$
\Delta(s_t) := V^\star(s_t) - Q^\star(s_t, a_t^+).
\tag{244}
$$

$$
\hat{\Delta}(s_t) := \hat{V}_{\text{ac}}^+(s_t) - Q^\star(s_t, a_t^+).
\tag{245}
$$

From the inequalities above, we have

$$
\hat{\Delta}(s_t) \leq \vartheta_h^L + \gamma^h \sup_{s_{t+h}} \left[ \hat{\Delta}(s_{t+h}) - \Delta(s_{t+h}) \right],
\tag{246}
$$

$$
0 \leq \Delta(s_t) \leq \frac{\vartheta_h^G}{1 - \gamma^h} + \hat{\Delta}(s_t),
\tag{247}
$$

$$
\hat{\Delta}(s_t) - \Delta(s_t) \leq \min \left\{ \frac{\vartheta_h^G}{1 - \gamma^h}, \hat{\Delta}(s_t) \right\}.
\tag{248}
$$

The minimum operator allows us to obtain two upper-bounds on $\Delta$:

$$
\Delta(s_t) \leq \vartheta_h^L + \frac{(1 + \gamma^h)\vartheta_h^G}{1 - \gamma^h},
\tag{249}
$$

$$
\Delta(s_t) \leq \frac{\vartheta_h^G}{1 - \gamma^h} + \hat{\Delta}(s_t) \leq \frac{\vartheta_h^L + \vartheta_h^G}{1 - \gamma^h}.
\tag{250}
$$

Finally, combining these two upper-bounds together and recursively applying the inequality yields our desired results:

$$
V^\star(s_t) - Q^\bullet(s_t, a_t^+) \leq \frac{\vartheta_h^L}{1 - \gamma} + \frac{\vartheta_h^L}{(1 - \gamma)(1 - \gamma^h)} + \frac{\gamma^h \min(\vartheta_h^G, \vartheta_h^L)}{(1 - \gamma)(1 - \gamma^h)}.
\tag{251}
$$

$\square$

### H.13 PROOF OF THEOREM 6

---

**Theorem 6** (Worst-case Closed-loop AC Policy under BOV) For any $h > 1, \gamma \in (0,1), \vartheta_h^G, \vartheta_h^L \in \left(0, \frac{\gamma - \gamma^h}{4(1-\gamma)}\right], c \in \left[0, \frac{\gamma - \gamma^h}{4(1-\gamma^h)}\right), \sigma \in \left(0, \frac{\min(\vartheta_h^G, \vartheta_h^L)}{1-\gamma}\right)$, there exists $\mathcal{M}$ and $\mathcal{D}$ satisfying the assumptions in Theorem 3 such that there exists $s_t \in \text{supp}(P_{\mathcal{D}^\star}(s_t))$, where

$$V^\star(s_t) - V^\bullet(s_t) = \frac{\vartheta_h^L}{1-\gamma} + \frac{\vartheta_h^G + \gamma^h \min(\vartheta_h^L, \vartheta_h^G)}{(1-\gamma)(1-\gamma^h)} - \sigma, V^\star(s_t) - V_{\text{ac}}^+(s_t) \geq \frac{c}{1-\gamma}. \quad (36)$$

---

To show that our upper bound is achievable, we need to carefully design both the MDP and the data distribution. For clarity of the proof, we divide up the construction into two parts. The first part (Lemma 9) focuses on designing part of the MDP and two data distributions $\mathcal{D}^\star$ and $\mathcal{D}^\diamond$ such that any action chunk that has a value bigger than $V^\star - \frac{\vartheta_h^G}{1-\gamma^h}$ is preferred over the action chunks in $\mathcal{D}^\star$ and $\mathcal{D}^\diamond$. The second part (Lemma 10) focuses on constructing the remaining MDP and the $\mathcal{D}^\triangle$ that contains the action chunk that $\pi_{\text{ac}}^+$ picks where $\hat{V}_{\text{ac}}^+$ overestimates the value of this action chunk by $\vartheta_h^L + \frac{\gamma^h \min(\vartheta^L, \vartheta_h^G)}{1-\gamma^h}$. Finally, we assemble these two results (combining $\mathcal{D}^\star, \mathcal{D}^\diamond, \mathcal{D}^\triangle$) to show that the MDP and the mixture data achieve our upper-bound *exactly*.

---

**Lemma 9** ("The Castle") For $\delta \in (0,1), \vartheta_h^G < \frac{\gamma - \gamma^h}{2(1-\gamma)}$, consider a 2-state, 2-action MDP in Figure 11. Let there be two data distributions, $\mathcal{D}^\star$ and $\mathcal{D}^\diamond$. $\mathcal{D}^\star$ is collected by the following optimal closed-loop policy from $X$ and $Y$:

$$\pi^\star(X) = 0, \pi^\star(Y) = 1. \quad (252)$$

$\mathcal{D}^\diamond$ is collected by the following optimal closed-loop policy from $X$ and $Y$:

$$\pi^\diamond(X) = 1, \pi^\diamond(Y) = 0. \quad (253)$$

Let $\mathcal{D}$ be a mixture of $\mathcal{D}^\star$ and $\mathcal{D}^\diamond$ with

$$P_{\mathcal{D}} = (1 - \varsigma)P_{\mathcal{D}^\star} + \varsigma P_{\mathcal{D}^\diamond}. \quad (254)$$

There exists $c_1 \in (0, 1/2)$ such that

1. $\mathcal{D}^\star$ and $\mathcal{D}^\diamond$ both individually exhibits 0-variability in optimality conditioned on $s_t, a_t$ for all $s_t, a_t \in \text{supp}(P_{\mathcal{D}}(s_t, a_t))$,

2. $\mathcal{D}$ exhibits $\vartheta_h^G$-variability in optimality conditioned on $s_t, a_{t:t+h}$ for all $s_t, a_{t:t+h} \in \text{supp}(P_{\mathcal{D}}(s_t, a_{t:t+h}))$,

and

$$\hat{V}_{\text{ac}}^+(X) = \hat{V}_{\text{ac}}^+(Y) = \frac{1 - \gamma + \varsigma(\gamma - \gamma^h)}{2(1-\gamma^h)(1-\gamma)} - \frac{\varsigma \vartheta_h^G}{1-\gamma^h}. \quad (255)$$

---

*Proof.* Set

$$c_1 = \frac{(1-\gamma)\vartheta_h^G}{\gamma - \gamma^h}. \quad (256)$$

We first check whether $c_1 \in (0, 1/2)$. For the upper-bound, it is clear that $c_1 < 1/2$ because $\vartheta_h^G < \frac{\gamma - \gamma^h}{2(1-\gamma)}$. For the lower-bound, $c > 0$ because all terms in the fraction are positive.

We now check the two optimality variability conditions. The first (local) one is trivial because $\pi^\diamond$ always receives $r = 1/2 - c_1$ and $\pi^\star$ always receives $r = 1/2$, and the optimal value for $X$ and $Y$ are both $V^\star(X) = V^\star(Y) = \frac{1}{2(1-\gamma)}$.

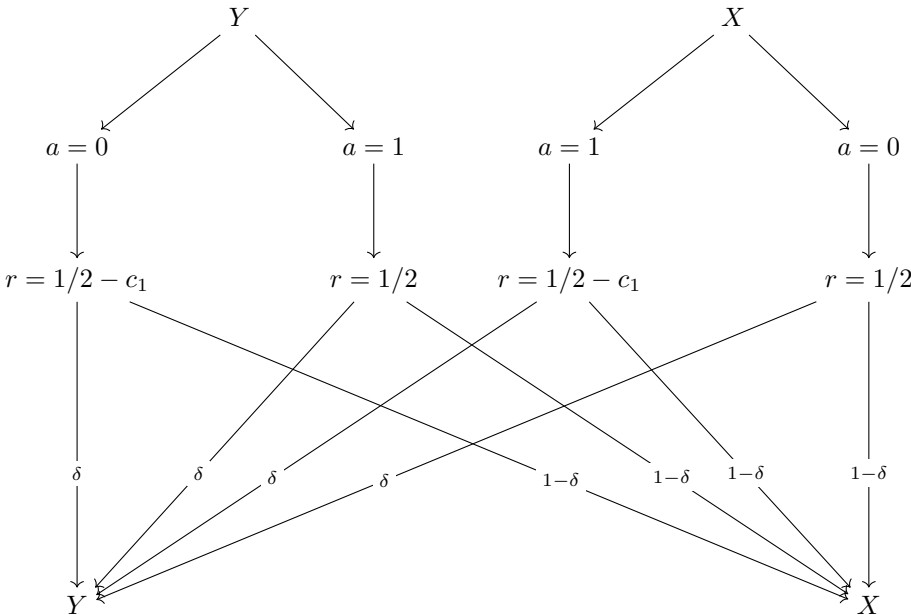

Figure 11: **MDP construction Part 1 for Theorem 6 ("the castle").** This diagram describes state $X$ and $Y$ and how actions $a = 0$ and $a = 1$ transition between them. The main purpose of this construction is to make $\hat{V}_{\mathrm{ac}}^{+}(X)$ underestimate $V^{\star}$ by exactly $\vartheta_{h}^{G}/(1 - \gamma^{h})$. This allows the action chunk that appears in the second part of the construction to be preferred (by $\pi_{\mathrm{ac}}^{+}$) over the action chunks that start with $a = 0$ or $a = 1$.

Next, we check the second (global) condition by analyzing all possible states and action chunks in $\mathcal{D}$. We observe that for any $a_{t:t+h}$ that starts with $a_t = 0$, we have

$$\tilde{Q}_{\min}(X, a_{t:t+h}) = \frac{1 - 2c_1(\gamma - \gamma^h)}{2(1 - \gamma)}, \tag{257}$$

$$\tilde{Q}_{\max}(X, a_{t:t+h}) = \frac{1}{2(1 - \gamma)}, \tag{258}$$

which gives

$$\tilde{Q}_{\max}(X, a_{t:t+h}) - \tilde{Q}_{\min}(X, a_{t:t+h}) = \vartheta_{h}^{G}. \tag{259}$$

By symmetry, we also have

$$\tilde{Q}_{\max}(Y, a_{t:t+h}) - \tilde{Q}_{\min}(Y, a_{t:t+h}) = \vartheta_{h}^{G}. \tag{260}$$

for all $a_{t:t+h}$ that starts with $a_t = 1$.

Now, for any $a_{t:t+h}$ that starts with $a_t = 1$, we have

$$\tilde{Q}_{\min}(X, a_{t:t+h}) = \frac{\gamma - 2c_1(\gamma - \gamma^h)}{2(1 - \gamma)}, \tag{261}$$

$$\tilde{Q}_{\max}(X, a_{t:t+h}) = \frac{\gamma}{2(1 - \gamma)}, \tag{262}$$

which admits the same gap as the case when $a_t = 0$. The same also holds for $Y$ with $a_t = 1$. Thus, $\mathcal{D}$ exhibits $\vartheta_{h}^{G}$-variability in optimality conditioned on $s_t, a_{t:t+h}$ for all $s_t, a_{t:t+h} \in \mathrm{supp}(P_{\mathcal{D}}(s_t, a_{t:t+h}))$.

Finally, we check for the value,

$$
\begin{aligned}
\hat{V}_{\text{ac}}^+(X) = \hat{V}_{\text{ac}}^+(Y) &= (1-\varsigma)/2 + \varsigma(1/2 + (1-2c_1)\frac{\gamma - \gamma^h}{2(1-\gamma)}) \\
&= \frac{1}{1-\gamma^h}\left[1/2 + \varsigma\frac{(1-2c_1)(\gamma - \gamma^h)}{2(1-\gamma)}\right] \\
&= \frac{1}{2(1-\gamma^h)}\left[1 + \varsigma\frac{\gamma - \gamma^h - 2(1-\gamma)\vartheta_h^G}{1-\gamma}\right] \\
&= \frac{1-\gamma+\varsigma(\gamma-\gamma^h)}{2(1-\gamma^h)(1-\gamma)} - \frac{\varsigma\vartheta_h^G}{1-\gamma^h},
\end{aligned}
\tag{263}
$$

as desired. □

---

**Lemma 10** ("The Flower") Assume $\vartheta_h^G \in \left(0, \frac{1-\gamma^h}{8}\right], \vartheta_h^L \in \left(0, \frac{\gamma-\gamma^h}{4(1-\gamma)}\right], \gamma \in (0,1)$, and Consider a 5-state, 3-action MDP in Figure 12 building on top of the transitions that already in Figure 11. Let $\mathcal{D}^\triangle$ be a data distribution induced by a cycling, time-dependent (with a time cycle length of $h$) policy $\pi^\triangle$ (we use the subscript to indicate the time step from 0 to $h-1$):

$$\pi_0^\triangle(s_t = X) = \pi_0^\triangle(s_t = \tilde{X}) = 2, \tag{264}$$

$$\pi_0^\triangle(s_t = Y) = 3 \tag{265}$$

$$\pi_k^\triangle(s_{t+k} = \tilde{C}) = \pi_k^\triangle(s_{t+k} = \tilde{D}) = 2, \quad \forall k \in \{1, 2, \cdots, h-2\}, \tag{266}$$

$$\pi_k^\triangle(s_{t+h-1} = \tilde{C}) = \pi_k^\triangle(s_{t+h-1} = \tilde{D}) = 0, \tag{267}$$

$$\pi_k^\triangle(s_{t+k} = X) = 0, \quad \forall k \in \{1, 2, \cdots, h-1\}, \tag{268}$$

$$\pi_k^\triangle(s_{t+k} = Y) = 1, \quad \forall k \in \{1, 2, \cdots, h-1\}. \tag{269}$$

Let $\hat{V}_{\text{ac}}^+$ be the nominal value of the action chunking policy $\pi_{\text{ac}}^+$ learned from $\mathcal{D}^\triangle$ and let

$$\Delta = \vartheta_h^L + \frac{\vartheta_h^G}{1-\gamma^h} + \frac{\gamma^h \min(\vartheta_h^G, \vartheta_h^L)}{1-\gamma^h}. \tag{270}$$

For any $c \in \left[0, \frac{\gamma-\gamma^h}{4(1-\gamma^h)}\right)$, there exists some $0 < c_2 \leq 1/2, 0 < c_3 \leq 1/2, \delta, \delta_2 \in (0,1)$, such that for every $0 < \tilde{\Delta} < \min\left(\Delta, \frac{2\vartheta_h^G}{1-\gamma^h}\right)$,

1. $\mathcal{D}^\triangle$ exhibits 0-variability in optimality conditioned on $s_t, a_{t:t+h}$ for all $s_t, a_{t:t+h} \in \text{supp}(P_{\mathcal{D}^\triangle}(s_t, a_{t:t+h}))$,

2. $\mathcal{D}^\triangle$ exhibits $\vartheta_h^L$-variability in optimality conditioned on $s_t, a_t$ for all $s_t, a_t \in \text{supp}(P_{\mathcal{D}^\triangle}(s_t, a_t))$,

and

$$\hat{V}_{\text{ac}}^+(X) = \frac{1}{2(1-\gamma)} - \frac{\vartheta_h^G}{1-\gamma^h} + \tilde{\Delta}, \tag{271}$$

$$V^\star(X) - V^\bullet(X) = \frac{\Delta - \tilde{\Delta}}{1-\gamma}, \tag{272}$$

$$V^\star(X) - V_{\text{ac}}^+(X) \geq \frac{c}{1-\gamma}, \tag{273}$$

$$V^\star(X) - V_{\text{ac}}^\star(X) \geq \frac{c}{1-\gamma}. \tag{274}$$

---

*Proof.* Without the loss of generality, we assume we always start from state $X$. Due to symmetry, the same analysis applies to state $Y$ (with the first action being $a_t = 3$ rather than $a_t = 2$).

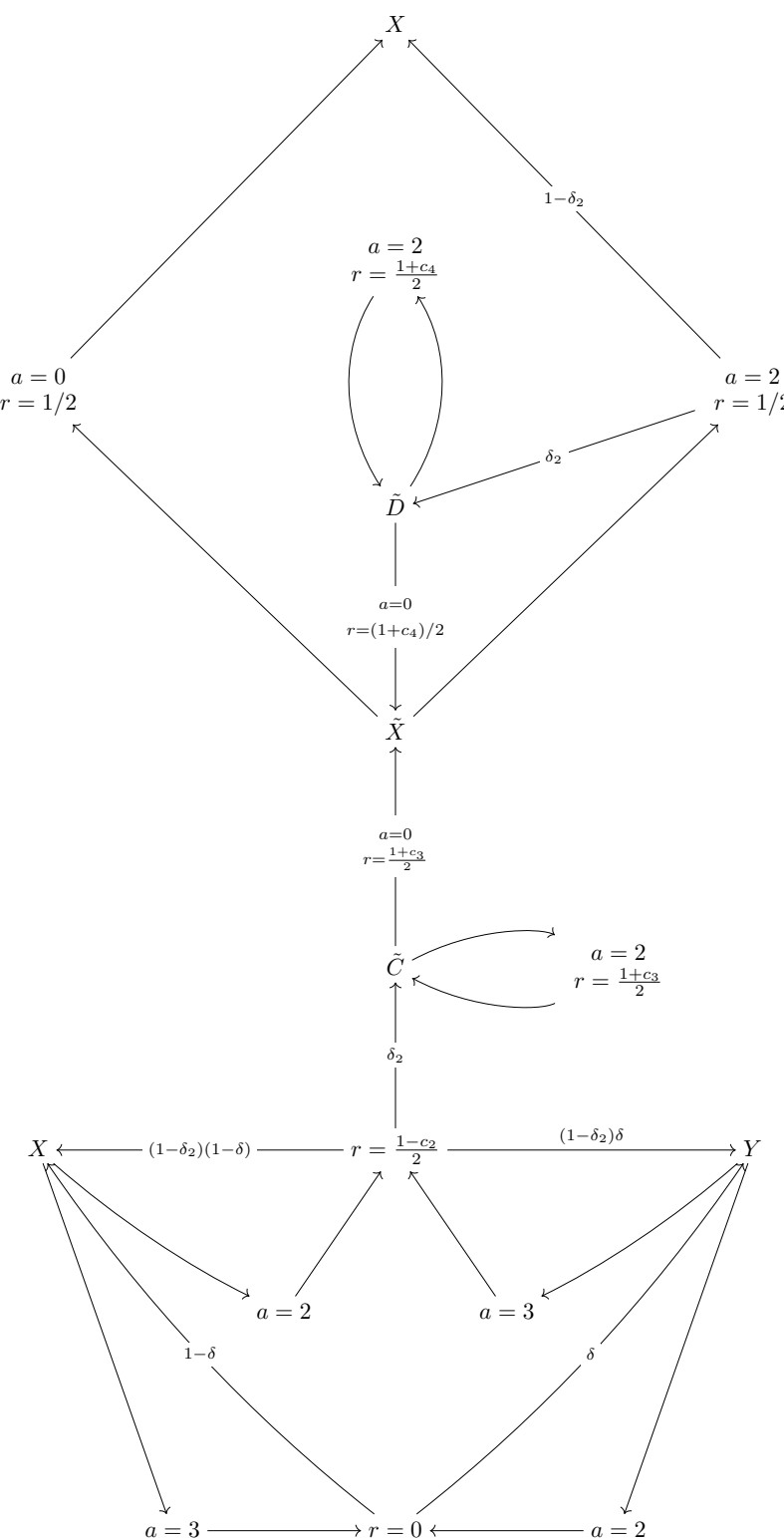

Figure 12: **MDP construction Part 2 for Theorem 6 ("the flower").** This diagram describes the remaining states $\tilde{C}$, $\tilde{D}$ and $\tilde{X}$, and what actions $a = 2$ and $a = 3$ do in state $X$ and $Y$. The main purpose of this construction is to make $\hat{V}_{\mathrm{ac}}^+(X)$ overestimate the optimal value of the action chunks that $\pi_{\mathrm{ac}}^+$, $Q^\star(X, a_t^+)$, by exactly $\vartheta_h^L + \gamma^h \min(\vartheta_h^L, \vartheta_h^G)/(1 - \gamma^h)$.

Due to cycling nature of the data collection policy, we observe that all action chunks starting from $X$ are in the form of $a_{t:t+h} = (2, \underbrace{\cdots}_{\text{0's and 1's}})$ or $a_{t:t+h} = (2, 2, \cdots, 2, 0)$. These two possibilities correspond to two different paths that the data collection policy takes:

- $a_{t:t+h}^{\circ} = (2, \underbrace{\cdots}_{\text{0's and 1's}})$: Stay in either $X$ or $Y$. The agent going on this path receives a constant reward of $1/2$ except the first step where it receives a reward of $(1 - c_2)/2$.

- $a_{t:t+h}^{\triangle} = (2, 2, \cdots, 2, 0)$: Visit $\tilde{C}$ and then stays there for $h - 1$ until it goes out with $a = 0$ to visit $\tilde{X}$. The agent going on this path receives a constant reward of $(1 + c_3)/2$ except the first step where it receives a reward of $(1 - c_2)/2$.

Similarly, all action chunks starting from $\tilde{X}$ are in the form of $a_{t:t+h} = (2, \underbrace{\cdots}_{\text{0's and 1's}})$ or $a_{t:t+h} = (2, 2, \cdots, 2, 0)$. These two possibilities correspond to two different paths that the data collection policy takes:

- $a_{t:t+h}^{\circ} = (2, \underbrace{\cdots}_{\text{0's and 1's}})$: Stay in either $X$ or $Y$. The agent going on this path receives a constant reward of $1/2$.

- $a_{t:t+h}^{\triangle} = (2, 2, \cdots, 2, 0)$: Visit $\tilde{C}$ and then stays there for $h - 1$ until it goes out with $a = 0$ to visit $\tilde{X}$. The agent going on this path receives a constant reward of $(1 + c_4)/2$ except the first step where it receives a reward of $1/2$.

Now, we divide up the problem into two cases depending on the relative values of $\vartheta_h^L$ and $\vartheta_h^G$.

*1. Case $\vartheta_h^L \geq \vartheta_h^G$:*

Set

$$c_2 = 2\left[\vartheta_h^L + \frac{(1 + \gamma^h)\vartheta_h^G}{1 - \gamma^h}\right] - 2\tilde{\Delta} > 0, \tag{275}$$

$$c_3 = \frac{2(1 - \gamma)\vartheta_h^L}{\gamma - \gamma^h} > 0, \tag{276}$$

$$c_4 = \frac{2(1 - \gamma)\vartheta_h^G}{\gamma - \gamma^h} > 0. \tag{277}$$

Next, we check that $c_2, c_3, c_4 \leq 1/2$.

We first observe that

$$(1 - \gamma)(1 - \gamma^h) - 2(\gamma - \gamma^h) = 1 - 3\gamma + \gamma^h(\gamma + 1) \leq 1 - 3\gamma + \gamma(\gamma + 1) = (1 - \gamma)^2 \geq 0. \tag{278}$$

Dividing both sides by $8(1 - \gamma)$ yields

$$\frac{1 - \gamma^h}{8} \geq \frac{\gamma - \gamma^h}{4(1 - \gamma)} \geq \vartheta_h^L \geq \vartheta_h^G. \tag{279}$$

Now, using the inequality above, we have

$$
\begin{aligned}
c_2 &= 2\left[\vartheta_h^L + \frac{(1 + \gamma^h)\vartheta_h^G}{1 - \gamma^h}\right] - \tilde{2}\Delta \\
&\leq 2\left[\vartheta_h^L + \frac{(1 + \gamma^h)\vartheta_h^L}{1 - \gamma^h}\right] \\
&\leq \frac{4\vartheta_h^L}{1 - \gamma^h} \\
&\leq 1/2.
\end{aligned} \tag{280}
$$

Furthermore,

$$c_4 \le c_3 = \frac{2(1-\gamma)\vartheta_h^L}{\gamma - \gamma^h} \le 1/2. \tag{281}$$

Next, we check the data distribution $\mathcal{D}^\triangle$ satisfies both optimality variability conditions. We first note that we only need to check for $s_t \in \{X, \tilde{X}\}$ because all other states are out of the support due to the cycling nature of the data collection policies. The first (global) optimality condition is trivial because the $h$-step reward received is deterministic conditioned on $a_{t:t+h} \in \{a_{t:t+h}^\circ, a_{t:t+h}^\triangle\}$, and the optimal value of $V^\star(s_{t+h})$ is always $\frac{1}{2(1-\gamma)}$. This leads to 0-variability in optimality conditioned on $s_t, a_{t:t+h}$. For the second (local) optimality condition, we check the difference in optimality for two paths from $s_t, a_t = 2$ for both $s_t = X$ and $s_t = \tilde{X}$.

For $s_t = X$, the optimality gap is

$$c_3 \frac{\gamma - \gamma^h}{2(1 - \gamma^h)} = \vartheta_h^L. \tag{282}$$

For $s_t = \tilde{X}$, the optimality gap is

$$c_4 \frac{\gamma - \gamma^h}{2(1 - \gamma^h)} = \vartheta_h^G \le \vartheta_h^L. \tag{283}$$

This concludes that the second (local) optimality condition is also satisfied.

Next, we first analyze which action chunk $\pi_{\mathrm{ac}}^+$ prefers by computing $\hat{Q}_{\mathrm{ac}}^+$'s:

$$\hat{Q}_{\mathrm{ac}}^+(X, a_{t:t+h}^\circ) = \frac{1}{2}\left[(1 - c_2) + \frac{\gamma - \gamma^h}{1 - \gamma}\right] + \gamma^h \hat{V}_{\mathrm{ac}}^+(X), \tag{284}$$

$$\hat{Q}_{\mathrm{ac}}^+(X, a_{t:t+h}^\triangle) = \frac{1}{2}\left[(1 - c_2) + (1 + c_3)\frac{\gamma - \gamma^h}{1 - \gamma}\right] + \gamma^h \hat{V}_{\mathrm{ac}}^+(\tilde{X}), \tag{285}$$

$$\hat{Q}_{\mathrm{ac}}^+(\tilde{X}, a_{t:t+h}^\circ) = \frac{1}{2}\left[\frac{1 - \gamma^h}{1 - \gamma}\right] + \gamma^h \hat{V}_{\mathrm{ac}}^+(X), \tag{286}$$

$$\hat{Q}_{\mathrm{ac}}^+(\tilde{X}, a_{t:t+h}^\triangle) = \frac{1}{2}\left[1 + (1 + c_4)\frac{\gamma - \gamma^h}{1 - \gamma}\right] + \gamma^h \hat{V}_{\mathrm{ac}}^+(\tilde{X}). \tag{287}$$

We first observe that

$$\begin{aligned}
\hat{Q}_{\mathrm{ac}}^+(\tilde{X}, a_{t:t+h}^\triangle) - \hat{Q}_{\mathrm{ac}}^+(X, a_{t:t+h}^\triangle) &= \frac{1}{2}\left[c_2 - (c_3 - c_4)\frac{\gamma - \gamma^h}{1 - \gamma}\right] \\
&= \vartheta_h^L + \frac{(1 + \gamma^h)\vartheta_h^G}{1 - \gamma^h} - \vartheta_h^L + \vartheta_h^G - \tilde{\Delta} \\
&= \frac{2\vartheta_h^G}{1 - \gamma^h} - \tilde{\Delta} \\
&> 0.
\end{aligned} \tag{288}$$

Also,

$$\hat{Q}_{\mathrm{ac}}^+(\tilde{X}, a_{t:t+h}^\circ) - \hat{Q}_{\mathrm{ac}}^+(X, a_{t:t+h}^\circ) = c_2 > 0 \tag{289}$$

Therefore,

$$\begin{aligned}
\hat{V}_{\mathrm{ac}}^+(X) &= \max(\hat{Q}_{\mathrm{ac}}^+(X, a_{t:t+h}^\circ), \hat{Q}_{\mathrm{ac}}^+(X, a_{t:t+h}^\triangle)) \\
&< \max(\hat{Q}_{\mathrm{ac}}^+(\tilde{X}, a_{t:t+h}^\circ), \hat{Q}_{\mathrm{ac}}^+(\tilde{X}, a_{t:t+h}^\triangle)) \\
&= \hat{V}_{\mathrm{ac}}^+(\tilde{X}).
\end{aligned} \tag{290}$$

Now, we can compare the values for the action chunks for $X$ and $\tilde{X}$:

$$\hat{Q}_{\mathrm{ac}}^+(X, a_{t:t+h}^\triangle) - \hat{Q}_{\mathrm{ac}}^+(X, a_{t:t+h}^\circ) = c_3 \frac{\gamma - \gamma^h}{2(1 - \gamma)} + \gamma^h(\hat{V}_{\mathrm{ac}}^+(\tilde{X}) - \hat{V}_{\mathrm{ac}}^+(X)) > 0, \tag{291}$$

$$\hat{Q}_{\mathrm{ac}}^+(\tilde{X}, a_{t:t+h}^\triangle) - \hat{Q}_{\mathrm{ac}}^+(\tilde{X}, a_{t:t+h}^\circ) = c_4 \frac{\gamma - \gamma^h}{2(1 - \gamma)} + \gamma^h(\hat{V}_{\mathrm{ac}}^+(\tilde{X}) - \hat{V}_{\mathrm{ac}}^+(X)) > 0, \tag{292}$$

since $c_3, c_4 > 0$ and $h > 1, 0 < \gamma < 1$ (and thus $\frac{\gamma - \gamma^h}{1 - \gamma} > 0$).

This concludes that $\pi_{\mathrm{ac}}^+(X) = \pi_{\mathrm{ac}}^+(\tilde{X}) = a_{t:t+h}^\triangle = (2, 2, \cdots, 2, 0)$ and thus

$$\hat{V}_{\mathrm{ac}}^+(\tilde{X}) = \frac{1 - \gamma + (\gamma - \gamma^h)(1 + c_4)}{2(1 - \gamma^h)(1 - \gamma)}, \tag{293}$$

and

$$\hat{V}_{\mathrm{ac}}^+(X) = \frac{1}{2}\left[(1 - c_2) + (1 + c_3)\frac{\gamma - \gamma^h}{1 - \gamma}\right] + \frac{\gamma^h}{1 - \gamma^h}\hat{V}_{\mathrm{ac}}^+(\tilde{X})$$

$$= \frac{1}{2(1 - \gamma)} - \frac{\vartheta_h^G}{1 - \gamma^h} + \frac{\tilde{\Delta}}{2}. \tag{294}$$

We can now compute the remaining values as follows:

$$V^\star(X) = \frac{1}{2(1 - \gamma)}, \tag{295}$$

$$Q^\star(X, a = 2) = \frac{(1 - c_2)(1 - \gamma) + \gamma}{2(1 - \gamma)}, \tag{296}$$

$$Q^\bullet(X, a = 2) = \frac{1 - c_2}{2(1 - \gamma)}. \tag{297}$$

Substituting the value of $c_2$ yields

$$V^\star(X) - V^\bullet(X) = \frac{\vartheta_h^L}{1 - \gamma} + \frac{(1 + \gamma^h)\vartheta_h^G}{(1 - \gamma)(1 - \gamma^h)} - \frac{\tilde{\Delta}}{2(1 - \gamma)}. \tag{298}$$

*2. Case $\vartheta_h^L < \vartheta_h^G$:*

Set

$$\Delta = 2\left[\frac{\vartheta_h^L + \vartheta_h^G}{1 - \gamma^h}\right] \tag{299}$$

$$c_2 = 2\left[\frac{\vartheta_h^L + \vartheta_h^G}{1 - \gamma^h}\right] - \tilde{\Delta} > 0, \tag{300}$$

$$c_3 = c_4 = \frac{2(1 - \gamma)\vartheta_h^L}{\gamma - \gamma^h} > 0 \tag{301}$$

where again $\tilde{\Delta}$ is any value that satisfies $0 < \tilde{\Delta} \leq \Delta$.

From the definitions above and the value range of $\vartheta_h^G$ ($\vartheta_h^G \leq \frac{1 - \gamma^h}{4}$), it is clear that

$$c_3 = c_4 < c_2 \leq \frac{4\vartheta_h^G}{1 - \gamma^h} \leq \frac{2(1 - \gamma)}{\gamma - \gamma^h} \leq 1/2. \tag{302}$$

Next, we check the data distribution $\mathcal{D}^\triangle$ satisfies both optimality variability conditions. With the same argument as the previous case, we can quickly conclude that the global optimality condition is satisfied. We just need to show the remaining local optimality condition. We repeat the procedure from the previous case.

For $s_t = X$, the local optimality gap is

$$c_3\frac{\gamma - \gamma^h}{2(1 - \gamma^h)} = \vartheta_h^L. \tag{303}$$

For $s_t = \tilde{X}$ the local optimality gap is the same because $c_4 = c_3$:

$$c_4\frac{\gamma - \gamma^h}{2(1 - \gamma^h)} = \vartheta_h^L. \tag{304}$$

This concludes that the second (local) optimality condition is also satisfied for the second case.

Now, we can follow the same procedure as the previous case to show that $\hat{Q}_{\text{ac}}^+(X, a_{t:t+h}^{\triangle}) - \hat{Q}_{\text{ac}}^+(X, a_{t:t+h}^{\circ}) > 0$ and $\hat{Q}_{\text{ac}}^+(\tilde{X}, a_{t:t+h}^{\triangle}) - \hat{Q}_{\text{ac}}^+(\tilde{X}, a_{t:t+h}^{\circ}) > 0$.

This concludes that $\pi_{\text{ac}}^+(X) = \pi_{\text{ac}}^+(\tilde{X}) = a_{t:t+h}^{\triangle} = (2, 2, \cdots, 2, 0)$, and thus

$$\hat{V}_{\text{ac}}^+(\tilde{X}) = \frac{1}{2} \left[ \frac{1 - \gamma + (1 + c_3)(\gamma - \gamma^h)}{(1 - \gamma)(1 - \gamma^h)} \right], \tag{305}$$

and

$$\hat{V}_{\text{ac}}^+(X) = \frac{1}{2} \left[ (1 - c_2) + (1 + c_3) \frac{\gamma - \gamma^h}{1 - \gamma} \right] + \frac{\gamma^h}{1 - \gamma^h} \hat{V}_{\text{ac}}^+(\tilde{X})$$
$$= \frac{1}{2(1 - \gamma)} - \frac{\vartheta_h^G}{1 - \gamma^h} + \frac{\tilde{\Delta}}{2}. \tag{306}$$

Repeating the same procedure as the previous case, we obtain

$$V^\star(X) - Q^\star(X, a = 2) = \frac{\vartheta_h^L + \vartheta_h^G}{1 - \gamma^h} - \tilde{\Delta}, \tag{307}$$

resulting in an optimality of

$$V^\star(X) - V^\bullet(X) = \frac{\vartheta_h^L + \vartheta_h^G}{(1 - \gamma)(1 - \gamma^h)} - \frac{\tilde{\Delta}}{1 - \gamma}. \tag{308}$$

*3. Sub-optimality of $V_{\text{ac}}^+$:*

Finally, we can use a pretty crude upper-bound on the actual value of the action chunking policy $\pi_{\text{ac}}^+$ (reparameterizing $\tilde{\delta}_2 = 1 - (1 - \delta_2)^h$):

$$V_{\text{ac}}^+(X) \leq (1 - \tilde{\delta}_2) \left[ (1 - c_2)/2 + \frac{\delta(\gamma - \gamma^h)}{2(1 - \gamma)} + \gamma^h V_{\text{ac}}^+(X) \right] + \frac{\tilde{\delta}_2}{1 - \gamma} \tag{309}$$

$$\leq \frac{1 - \tilde{\delta}_2}{2(1 - \gamma^h)(1 - \gamma)} \left[ 1 - \gamma + \delta(\gamma - \gamma^h) \right] + \frac{\tilde{\delta}_2}{1 - \gamma}. \tag{310}$$

Set $\delta = 1/2$, we have

$$V_{\text{ac}}^+(X) \leq \frac{1 - \tilde{\delta}_2}{2(1 - \gamma^h)(1 - \gamma)} \left[ 1 - \gamma/2 - \gamma^h/2 \right] + \frac{\tilde{\delta}_2}{1 - \gamma}. \tag{311}$$

We set

$$\delta_2 = 1 - \left[ 1 - \frac{\gamma - \gamma^h - 4c(1 - \gamma^h)}{2 - 3\gamma^h + \gamma} \right]^{1/h}, \tag{312}$$

which results in

$$\tilde{\delta}_2 = \frac{\gamma - \gamma^h - 4c(1 - \gamma^h)}{2 - 3\gamma^h + \gamma}. \tag{313}$$

It is clear that $0 < \delta_2 < 1$ because $c < \frac{\gamma - \gamma^h}{4(1 - \gamma^h)}$ and $\frac{\gamma - \gamma^h}{2 - 3\gamma^h + \gamma} < 1$.

Substituting $\tilde{\delta}_2$ in the bound of $V_{\text{ac}}^+(X)$ above, we obtain

$$V^\star(X) - V_{\text{ac}}^+(X) \geq \frac{c}{1 - \gamma}. \tag{314}$$

$\square$

*Proof of Theorem 6.* Let

$$\Delta = \vartheta_h^L + \frac{\vartheta_h^G}{1-\gamma^h} + \frac{\gamma^h \min(\vartheta_h^G, \vartheta_h^L)}{1-\gamma^h}. \tag{315}$$

Consider the 5-state, 3-action MDP constructed in Lemma 9 and Lemma 10 and a data distribution consisting of a mixture of three data distributions $\mathcal{D}^\star, \mathcal{D}^\diamond$ (from Lemma 9) and $\mathcal{D}^\triangle$ (from Lemma 10):

$$P_\mathcal{D} = \alpha(1-\varsigma)P_{\mathcal{D}^\star} + \varsigma P_{\mathcal{D}^\diamond} + (1-\alpha)P_{\mathcal{D}^\triangle}. \tag{316}$$

We set $\alpha$ to be any value between $0$ and $1$ (non-inclusive) and set $\varsigma$ as any positive value such that

$$\varsigma < \frac{(\gamma - \gamma^h) - 2\vartheta_h^G(1-\gamma) + 2\tilde{\Delta}(1-\gamma)(1-\gamma^h)}{(\gamma - \gamma^h) - 2\vartheta_h^G(1-\gamma)}, \tag{317}$$

where $\tilde{\Delta} = \sigma(1-\gamma) < \min(\vartheta_h^L, \vartheta_h^G) < \min(\Delta, \frac{2\vartheta_h^G}{1-\gamma^h})$ (satisfying the condition for $\tilde{\Delta}$ in Lemma 10).

The numerator and the denominator are both positive:

$$(\gamma - \gamma^h) - 2\vartheta_h^G(1-\gamma) + 2\tilde{\Delta}(1-\gamma)(1-\gamma^h) > (\gamma - \gamma^h) - 2\vartheta_h^G(1-\gamma) > 0, \tag{318}$$

meaning such $\varsigma$ always exists.

Substituting the inequality to the result of Lemma 9 results in

$$\frac{1 - \gamma + \varsigma(\gamma - \gamma^h)}{2(1-\gamma^h)(1-\gamma)} - \frac{\varsigma\vartheta_h^G}{1-\gamma^h} < \frac{1}{2(1-\gamma)} - \frac{\vartheta_h^G}{1-\gamma^h} + \tilde{\Delta}, \tag{319}$$

which shows that $\pi_{\mathrm{ac}}^+$ will always prefer $a_{t:t+h}^\triangle$ over action chunks in $\mathcal{D}^\star$ and $\mathcal{D}^\diamond$.

This means that the value $\hat{V}_{\mathrm{ac}}^+$ and the action chunking policy $\pi_{\mathrm{ac}}^+$ we learn from $\mathcal{D}$ coincides with these of $\mathcal{D}^\triangle$, allowing us to directly use the results of Lemma 10.

Thus, we can conclude that

$$V^\star(s_t) - V_{\mathrm{ac}}^+(s_t) \geq \frac{c}{1-\gamma}, \tag{320}$$

and

$$V^\star(X) - V^\bullet(X) = \frac{\Delta - \tilde{\Delta}}{1-\gamma} = \frac{\vartheta_h^L}{1-\gamma} + \frac{\vartheta_h^G}{(1-\gamma)(1-\gamma^h)} + \frac{\gamma^h \min(\vartheta_h^L, \vartheta_h^G)}{(1-\gamma)(1-\gamma^h)} - \sigma, \tag{321}$$

as desired.

$\square$

## H.14 PROOF OF PROPOSITION 5

> **Proposition 5** (Worst-case analysis of $n$-step return backup)  For any $n \in \mathbb{N}^+$, $\tilde{\delta}_n \in (0, \gamma - \gamma^n)$ and $\sigma \in \left(0, \tilde{\delta}_n/(1-\gamma)\right)$, there exists an MDP $\mathcal{M}$, and a $\tilde{\delta}_n$-optimal data distribution $\mathcal{D}$ with $\mathrm{supp}(P_{\mathcal{D}}(s_t, a_t)) \supseteq \mathrm{supp}(P_{\mathcal{D}^\star}(s_t, a_t))$ such that for some $s \in \mathrm{supp}(P_{\mathcal{D}^\star}(s_t))$,
>
> $$V_{\mathrm{ac}}^+(s) - V_n^+(s) = \frac{\tilde{\delta}_n}{1-\gamma} - \sigma, \tag{42}$$
>
> and for all $s \in \mathrm{supp}(P_{\mathcal{D}^\star}(s_t))$,
> $$V^\star(s) = V_{\mathrm{ac}}^+(s). \tag{43}$$

*Proof.* Consider an MDP in Figure 13. Let $\mathcal{D}$ be the data collected by the following policy:
$$\pi(a = 0 \mid X) = \pi(a = 1 \mid X) = 1/2, \tag{322}$$
$$\pi(a = 0 \mid Y) = \alpha, \tag{323}$$
$$\pi(a = 1 \mid Y) = 1 - \alpha. \tag{324}$$

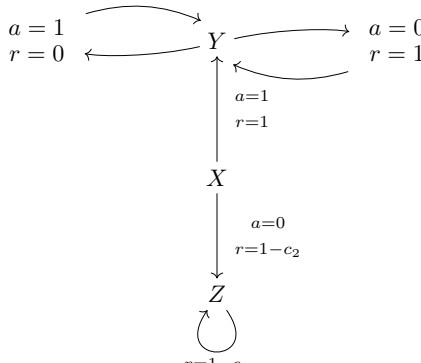

Figure 13: **An MDP where the learned action chunking policies are optimal and the learned $n$-step return policies can be arbitrarily sub-optimal.**

It is clear that the optimal policy is $\pi^\star(X) = 1, \pi^\star(Y) = 0$.

Since the dynamics are deterministic, the data distribution $\mathcal{D}$ is strongly open-loop consistent and thus by Theorem 2, $V_{\mathrm{ac}}(s_t) = V^\star(s_t)$ for all $s_t \in \mathrm{supp}(P_{\mathcal{D}^\star}(s_t))$.

To make sure the data distribution satisfies the $\tilde{\delta}_n$-optimal condition, we set
$$\alpha = 1 - \frac{\tilde{\delta}_n}{\gamma - \gamma^n}, \qquad c_2 = \tilde{\delta}_n - (1-\gamma)\sigma. \tag{325}$$

It is clear that at state $Y$, $\pi_n^+(Y) = 0$. We can then calculate the optimality gap for $\hat{Q}_n^+$ exactly as follows:
$$V^\star(X) - \hat{Q}_n^+(X, a = 0) = \frac{c_2}{1-\gamma} = \frac{\tilde{\delta}_n}{1-\gamma} - \sigma, \tag{326}$$
$$V^\star(X) - \hat{Q}_n^+(X, a = 1) = \frac{1}{1-\gamma} - \frac{1 - \gamma + \alpha(\gamma - \gamma^n)}{(1-\gamma)(1-\gamma^n)} = \frac{\tilde{\delta}_n}{1-\gamma}. \tag{327}$$

Since $\sigma > 0$, $\pi_n^+(X) = 0$. Now, we can compute $V_n(X)$ as follows:
$$V^\star(X) - V_n(X) = \frac{c_2}{1-\gamma} = \frac{\tilde{\delta}_n}{1-\gamma} - \sigma, \tag{328}$$

as desired.

$\square$

### H.15 PROOF OF PROPOSITION 4

> **Proposition 4** (Deterministic Dynamics are Weakly Open-loop Consistent)  If a transition dynamics $\mathcal{M}$ is $\varepsilon$-deterministic, then any data $\mathcal{D}$ collected from $\mathcal{M}$ is weakly $\varepsilon_h$-open-loop consistent with respect to $\mathcal{M}$ for any $h \in \mathbb{N}^+$ as long as $\varepsilon_h \geq 3(1 - (1 - \varepsilon)^{h-1})$.

*Proof.* Since $T$ is $\varepsilon$-deterministic, it can be represented as $T(\cdot \mid s, a) = (1 - \varepsilon)\delta_{f(s,a)} + \varepsilon \tilde{T}(\cdot \mid s, a)$ for some $f : \mathcal{S} \times \mathcal{A} \to \mathcal{S}$ and $\tilde{T} : \mathcal{S} \times \mathcal{A} \to \Delta_\mathcal{S}$. Let $f(s, a_1, \cdots, a_h) = f(\cdots f(f(s, a_1), a_2) \cdots a_h)$.

Let $I \in \{0, 1\}$ a binary indicator variable that is 1 if and only if

$$s_{t+k+1} = f(s_{t+k}, a_{t+k}), \forall k \in \{0, 1, 2, \cdots, h-1\} \tag{329}$$

Intuitively $I = 1$ when the trajectory is generated deterministically until but not including the last state $s_h$ in the trajectory chunk.

From the fact that $T$ is $\varepsilon$-deterministic, we know that

$$P_\mathcal{D}(I_h = 1) \geq (1 - \varepsilon)^{h-1} \tag{330}$$

We also have

$$P_\mathcal{D}(a_{t:t+h} \mid s_t) = P_\mathcal{D}(I_h = 1)P_\mathcal{D}(a_{t:t+h} \mid s_t, I_h = 1) + P_\mathcal{D}(I_h = 0)P_\mathcal{D}(a_{t:t+h} \mid s_t, I = 0) \tag{331}$$

Then we have

$$D_{\mathrm{TV}}(P_\mathcal{D}(a_{t:t+h} \mid s_t) \,\|\, P_\mathcal{D}(a_{t:t+h} \mid s_t, I_h = 1)) \leq (1 - (1 - \varepsilon)^{h-1}) \tag{332}$$

If we transform each distribution of $a_{t:t+h}$ deterministically by $f(s_t, \cdot)$, by data processing inequality (DPI; Lemma 4), we have

$$D_{\mathrm{TV}}\left(\mathbb{E}_{a_{t:t+h} \sim P_\mathcal{D}(\cdot \mid s_t)}\left[\delta_{f(s_t, a_{t:t+h})}\right] \,\middle\|\, \mathbb{E}_{a_{t:t+h} \sim P_\mathcal{D}(\cdot \mid s_t, I_h = 1)}\left[\delta_{f(s_t, a_{t:t+h})}\right]\right) \leq (1 - (1 - \varepsilon)^{h-1}) \tag{333}$$

Similarly, we have

$$D_{\mathrm{TV}}(P_\mathcal{D}(a_{t:t+h+1} \mid s_t) \,\|\, P_\mathcal{D}(a_{t:t+h+1} \mid s_t, I_{h+1} = 1)) \leq (1 - (1 - \varepsilon)^h) \tag{334}$$

which can be also deterministically transformed by taking $a_{t:t+h+1} \mapsto (f(s_t, \cdot), a_{t+h})$ (again with DPI, Lemma 4) to obtain

$$D_{\mathrm{TV}}\Big(\mathbb{E}_{a_{t:t+h} \sim P_\mathcal{D}(\cdot \mid s_t)}\left[\pi_\mathcal{D}^\circ(a_{t+h} \mid s_t, a_{t:t+h})\mathbb{I}_{f(s_t, a_{t:t+h})}\right] \,\|\,$$
$$\mathbb{E}_{a_{t:t+h} \sim P_\mathcal{D}(\cdot \mid s_t, I_{h+1} = 1)}\left[\pi_\mathcal{D}^\circ(a_{t+h} \mid s_t, a_{t:t+h}, I_{h+1} = 1)\mathbb{I}_{f(s_t, a_{t:t+h})}\right]\Big) \leq (1 - (1 - \varepsilon)^h) \tag{335}$$

Now, if we analyze the distribution of $s_{t+h}$ subject to the open-loop execution of the action sequence from $P_\mathcal{D}(\cdot \mid s_t)$ and break it up into the deterministic and the non-deterministic case, we get

$$\mathbb{E}_{a_{t:t+h} \sim P_\mathcal{D}(\cdot \mid s_t)}\left[T_{a_{t:t+h}}(\cdot \mid s_t)\right] = P_T(I = 1)\mathbb{E}_{a_{t:t+h} \sim P_\mathcal{D}(\cdot \mid s_t)}\left[\delta_{f(s_t, a_{t:t+h})}\right] +$$
$$P_T(I = 0)\mathbb{E}_{a_{t:t+h} \sim P_\mathcal{D}(\cdot \mid s_t)}\left[T_{a_{t:t+h}}(\cdot \mid s_t, I_h = 0)\right] \tag{336}$$

Note that $P_T(I = 1)$ denotes the probability that an open-loop executed trajectory using $a_{t:t+h} \sim P_\mathcal{D}(\cdot \mid s_t)$ is deterministic. This is different from $P_\mathcal{D}(I_h = 1)$ because the latter is based on $P_\mathcal{D}(s_{t:t+h+1}, a_{t:t+h})$ whereas $P_T(I_h = 1)$ is based on the open-loop trajectory distribution: $P_\mathcal{D}(\cdot \mid s_t)\prod_{k=0}^{h-1} T(s_{t+k} \mid s_t, a_{t:t+k})$. They both admit the same lower bound of $2(1 - (1 - \varepsilon)^{h-1})$.

Therefore,

$$D_{\mathrm{TV}}\left(\mathbb{E}_{a_{t:t+h} \sim P_\mathcal{D}(\cdot \mid s_t)}\left[T_{a_{t:t+h}}(\cdot \mid s_t)\right] \,\|\, \mathbb{E}_{a_{t:t+h} \sim P_\mathcal{D}(\cdot \mid s_t)}\left[\delta_{f(s_t, a_{t:t+h})}\right]\right) \leq (1 - (1 - \varepsilon)^{h-1}) \tag{337}$$

Similarly for the state-action case, we can multiply both side by the same conditional distribution $\pi_\mathcal{D}^\circ(a_{t+h} \mid s_t, a_{t:t+h})$ which preserves the TV bound. For the left-hand side, we have

$$P_\mathcal{D}^\circ(s_{t+h}, a_{t+h} \mid s_t) = \mathbb{E}_{a_{t:t+h} \sim P_\mathcal{D}(\cdot \mid s_t)}\left[\pi_\mathcal{D}^\circ(a_{t+h} \mid s_t, a_{t:t+h})T_{a_{t:t+h}}(s_{t+h} \mid s_t)\right] \tag{338}$$

Therefore, we get

$$D_{\mathrm{TV}}\big(P_{\mathcal{D}}^{\circ}(s_{t+h}, a_{t+h} \mid s_t) \,\big\|\, \mathbb{E}_{a_{t:t+h} \sim P_{\mathcal{D}}(\cdot \mid s_t)} \big[\pi_{\mathcal{D}}^{\circ}(a_{t+h} \mid s_t, a_{t:t+h}) \mathbb{I}_{f(s_t, a_{t:t+h})}\big]\big)$$
$$\leq (1 - (1-\varepsilon)^{h-1}) \tag{339}$$

We also have

$$P_{\mathcal{D}}(s_{t+h} \mid s_t) = (1-\varepsilon)^{h-1} P_{\mathcal{D}}(s_{t+h} \mid s_t, I = 1) + (1 - (1-\varepsilon)^{h-1}) P_{\mathcal{D}}(s_{t+h} \mid s_t, I_h = 0) \tag{340}$$

Similarly, we have

$$D_{\mathrm{TV}}(P_{\mathcal{D}}(s_{t+h} \mid s_t) \,\|\, P_{\mathcal{D}}(s_{t+h} \mid s_t, I_h = 1))$$
$$= D_{\mathrm{TV}}\big(P_{\mathcal{D}}(s_{t+h} \mid s_t) \,\big\|\, \mathbb{E}_{a_{t:t+h} \sim P_{\mathcal{D}}(\cdot \mid s_t, I_h = 1)} \big[\delta_{f(s_t, a_{t:t+h})}\big]\big) \leq (1 - (1-\varepsilon)^{h-1}) \tag{341}$$

For state-action, we can also get

$$P_{\mathcal{D}}(s_{t+h}, a_{t+h} \mid s_t) = (1-\varepsilon)^h P_{\mathcal{D}}(s_{t+h}, a_{t+h} \mid s_t, I_{h+1} = 1)$$
$$+ (1 - (1-\varepsilon)^h) P_{\mathcal{D}}(s_{t+h}, a_{t+h} \mid s_t, I_{h+1} = 0) \tag{342}$$

which can be turned into the TV distance bound:

$$D_{\mathrm{TV}}(P_{\mathcal{D}}(s_{t+h}, a_{t+h} \mid s_t) \,\|\, P_{\mathcal{D}}(s_{t+h}, a_{t+h} \mid s_t, I_{h+1} = 1))$$
$$= D_{\mathrm{TV}}\Big(P_{\mathcal{D}}(s_{t+h}, a_{t+h} \mid s_t) \,\Big\|$$
$$\mathbb{E}_{a_{t:t+h} \sim P_{\mathcal{D}}(\cdot \mid s_t, I_{h+1} = 1)} \big[\pi_{\mathcal{D}}^{\circ}(a_{t+h} \mid s_t, a_{t:t+h}, I_{h+1} = 1) \mathbb{I}_{f(s_t, a_{t:t+h})}\big]\Big) \tag{343}$$
$$\leq (1 - (1-\varepsilon)^h)$$

Connecting all three total variation inequality (Equations (333), (337) and (341)) together, we get

$$D_{\mathrm{TV}}\big(P_{\mathcal{D}}(s_{t+h} \mid s_t) \,\big\|\, \mathbb{E}_{a_{t:t+h} \sim P_{\mathcal{D}}(\cdot \mid s_t)} \big[T_{a_{t:t+h}}(\cdot \mid s_t)\big]\big) \leq 3(1 - (1-\varepsilon)^{h-1}) \leq \varepsilon_h \tag{344}$$

Connecting all three total variable inequality for state-action (Equations (335), (338) and (343)) together, we get

$$D_{\mathrm{TV}}(P_{\mathcal{D}}^{\circ}(s_{t+h-1}, a_{t+h-1} \mid s_t) \,\|\, P_{\mathcal{D}}(s_{t+h}, a_{t+h} \mid s_t)) \leq 3 - 2(1-\varepsilon)^{h-1} - (1-\varepsilon)^{h-2}$$
$$\leq 3(1 - (1-\varepsilon)^{h-1}) \tag{345}$$
$$\leq \varepsilon_h$$

Therefore, $\mathcal{D}$ is $\varepsilon_h$-open-loop consistent as desired. $\qquad\square$

