# OpenReview forum: "Decoupled Q-Chunking"
_ICLR.cc/2026/Conference — ICLR 2026 Poster_

### Official Review · Reviewer_c39B · 2025-10-28

**Soundness:** 3
**Presentation:** 3
**Contribution:** 2
**Rating:** 6
**Confidence:** 3

**Summary:**

The paper addresses the bootstrapping bias problem by proposing a novel method to decouple the chunk length of the critic from that of the policy, allowing for more efficient multi-step return backups. The proposed algorithm optimizes the policy against a distilled critic, retaining the benefits of multi-step value propagation while avoiding the challenges of open-loop sub-optimality and long action chunk learning.

**Strengths:**

1. The paper is clearly written and easy to follow, with well-structured explanations.
2. The authors effectively discuss the scenarios in which action chunking Q-learning is preferable over standard n-step return learning.

**Weaknesses:**

1. My main concern is that, by comparing the performance and curves of DQC and QC-NS, QC-NS—without decoupling the action chunking of the Q-value from that of the policy—achieves nearly the same performance as DQC, except for the cube-quadruple-100M task.
2. How much does the performance degrade if the expectile update in Eq. (24) is replaced with a quantile update?
3. Why does setting $N=128$ not lead to better performance?
4. In most algorithms, the batch size is typically set to 256, but in this paper, it is set to 4096. Could the authors conduct an ablation study to justify this choice?
5. Could the proposed method be evaluated on standard offline RL benchmarks such as D4RL?
6. There appear to be some minor errors, such as two consecutive "the"s in the first line of page 2, and potential mistakes regarding $\tau_b$ and $\tau_d$ in Algorithm 1. The authors should carefully proofread the manuscript.

**Questions:**

Please see weakness

---

> ### Author Response · Authors · 2025-11-20
>
> Thanks for your detailed review and insightful feedback. Your review has improved our paper quite a bit and we really appreciate it! In this rebuttal, **(1) we addressed your main concern on the performance of DQC and QC-NS being too close by running more seeds and confirmed that QC-NS suffers from more instability compared to DQC, highlighting the importance of learning a separate distilled critic; (2) We addressed your questions regarding the sensitivity of DQC’s performance with respect to the implicit loss type, best-of-N, and batch size with additional sensitivity experiments.**
>
> ---
>
> ## 1. QC-NS is similar to DQC and QC-NS does not use a chunked critic.
>
> We re-tuned our method and all baselines as well as running more seeds ($3 \rightarrow \mathbf{6}$) to consolidate our empirical evaluations of QC-NS and DQC. From our experiments, we found that QC-NS is generally more unstable compared to DQC. There were signs of instability from our experiments in the original submission but were difficult to confirm due to the low number of seeds. We hypothesized that learning a separate critic in DQC helps reduce the variance and resulted in this improvement in stability. We summarize the updated results that compare QC-NS with DQC on all the cube tasks below.
>
> | **Task** | **DQC** | **QC-NS** |
> |-|-|-|
> |cube-triple-100M| **98** [97, 99]| 63 [31, 96]|
> |cube-quadruple-100M| **93** [91, 95]| 73 [44, 89]|
> |cube-octuple-1B|**31** [29, 33]|20 [12, 26]|
>
> ---
>
> ## 2. Performance difference if we use expectile backup instead of quantile backup
>
> Our conclusion is that the performance of our method is relatively not sensitive to the choice of expectile and distillation. We drew this conclusion from the new sensitivity analysis results in Figure 3 (second sub-plot) where we experimented with different combinations of expectile/quantile backup and expectile/quantile distillation. We found that using expectile backup resulted in a slight performance drop. Using quantile distillation instead of expectile distillation leads to a slightly larger performance drop but with a similar final performance.
>
> ---
>
> ## 3. Why the batch size of 4096 and not the standard 256?
>
> We used a large batch size (4096) in our work because we found it to be crucial for good performance on these difficult OGBench tasks. For this rebuttal, in Figure 4, we showed the performance of our approach with smaller batch sizes (e.g., 1024 and 256) on all tasks and found that there is a drastic performance drop from 4096 to 1024 and from 1024 to 256. It is worth noting that all our baselines (e.g., NS, QC, SHARSA, HIQL) are run with the same large batch size of 4096.
>
> ---
>
> ## 4. Why is the larger best-of-N not better?
>
> We first clarified that we followed IDQL [1] to employ Best-of-N during evaluations only and did not use it during training. Combined with implicit value learning, we decouple value learning completely from policy extraction (same as IDQL). While this lowers the training time (as we do not need to run the policy network to compute the Q-target), it also means that there is a limit to how much the policy extraction can improve by increasing $N$ as the performance would eventually be bottlenecked by the quality of our value function.
>
> ---
>
> ## 5. "Could the proposed method be evaluated on standard offline RL benchmarks such as D4RL?"
>
> Thanks for the request! The main reason we did not evaluate our method on D4RL was because the tasks have been increasingly saturated in performance [2] and the horizon length of the tasks in D4RL is not sufficiently long for our proposed method (which primarily focuses on speeding up value learning for long-horizon tasks) to shine. The longest antmaze-large task requires <500 steps to solve. In contrast, the OGBench tasks, especially the more difficult ones, require a significantly large number of steps to complete. For instance, on the cube tasks, it takes ~100 steps to move one cube and thus ~800 to solve some of the harder cube-octuple tasks. On humanoidmaze-giant, the hardest task requires ~3000 steps to travel from the initial state to the goal state. On puzzle tasks, it takes ~40 steps to press a button and solving the hardest puzzle-4x5 task and puzzle-4x6 task requires 20 and 24 button presses respectively (and thus a total number of steps of ~800 and ~960 respectively).
>
> [1] "IDQL: Implicit q-learning as an actor-critic method with diffusion policies." arXiv preprint arXiv:2304.10573 (2023).
>
> [2] "D5RL: Diverse datasets for data-driven deep reinforcement learning." arXiv preprint arXiv:2408.08441 (2024).
>
> ---
>
> *Grammar mistakes and typos:* Thank you for spotting these mistakes. We have fixed these errors, proofread our draft and updated the PDF.
>
> ---
>
> Thanks again for your review as especially your review has already resulted in significant improvements to our paper. **Please also let us know if you have any additional questions or concerns regarding our paper!**

---

### Official Review · Reviewer_WFcb · 2025-10-30

**Soundness:** 3
**Presentation:** 3
**Contribution:** 3
**Rating:** 6
**Confidence:** 4

**Summary:**

This paper introduces DQC, that builds upon Q-chunking but decouples the critic’s chunk length from the policy’s chunk length. It aims to address bootstrapping bias from TD backups and open-loop policy inefficiency in off-policy TD learning. Experiments on the OGBench suite show that DQC outperforms Q-chunking and other baselines.

**Strengths:**

1. This paper derives explicit bias and near-optimality bounds, offering theoretical insights of when action-chunking methods succeed or fail.

2. Empirical performance of DQC is nice.

**Weaknesses:**

1. Theorem 4.6 is somewhat idealized. It depends on the open-loop consistency assumption. It is hard to hold in the practice for realistic offline datasets, especially in long-horizon settings, in such cases $\epsilon_h$ may not be small, and thus the errors scale and then the bound can become vacuous. Could the authors provide an empirical analysis on how big is $\epsilon_h$ in the OGBench datasets?
2. Theorem 4.6 does not seem to cover the decoupled critics. Is this bound still valid when the two critics are trained with different horizons and objectives?

**Questions:**

1. The distilled partical critic $Q_\psi^P$ is not Bellman-consistent with the long-horizon cirtic $Q_\phi$. $Q_\psi^P$ is trained to approximate an optimistic projectiion of $Q_\phi$ rather than its bellman target, thus there's an inherent objective mismatch between the two critics. Will it introduce systematic bias or training instability? Especially, when in non-markovian or sparse-reward settings, the optimistic assumption might not hold. Could the authors clarify why this training remains stable and effective in practice? It would be helpful to see a quantitive analysis showing how closely $Q_\psi^P$ tracks $Q_\phi$.

2. How the key hyperparameters were chosen, for example, $h$ and $h_a$? How were these values selected in practice—through tuning, fixed ratios, or heuristics? And how sensitive is the method to their choice?

3. What is the computational overhead of maintaining two critics with different horizons?

4. In Algorithm, it introduces two expectile parameters, $\tau_d$ and $\tau_b$ for the two critics. However, I do not find how the two factors are selected or tuned in Table 4. Expectile regression is known to be sensitive to $\tau$. It would amplify overestimation bias when using a large $\tau$. Are $\tau_b$ and $\tau_d$ the same across envs, or task-specific? Did the authors observe significant sensitivity in performance when varing $\tau_b$ and $\tau_d$?

---

> ### Author Response · Authors · 2025-11-20
>
> Thanks for your detailed comments and insightful feedback. Your review has greatly improved our paper and we really appreciate it! In this rebuttal, **(1) we addressed the Bellman-inconsistency concern by clarifying/providing an alternative interpretation of the what our critics are estimating, as well as where the mismatch comes in, and finally providing additional empirical analysis that show the stable/effective learning of our algorithm, (2) we addressed your concerns on our theoretical analysis by introducing two new results regarding our algorithm (DQC), and (3) we addressed your concerns on lack of empirical details by including details of our hyperparameter tuning process, additional hyperparameter sensitivity analysis, and computational overhead of maintaining two critics.**
>
> ---
>
> ## 1. “Distilled partial critic is not Bellman-consistent with the long-horizon critic.”
>
> Thanks for the clarification question. We understand that you are coming from the understanding that $Q^P_\psi$ should approximate the value of the policy (with a smaller chunk size) and thus is inconsistent with $Q_\phi$ that approximates the value of the policy (with the same chunk size as that of the critic), and seemingly could cause a stability problem. Here, we offer an alternative view where actually **both $Q^P_\psi$ and $Q_\phi$ are approximating the value of the policy (with a chunk size as large as that of the critic).** In particular, we write down the semantic meaning of each value function exactly as follows:
>
> - $Q\_\phi(s, a\_{t:t+h})$: The expected discounted return if we are at state $s$, execute $a_{t:t+h}$ open-loop for the next $h$-step, and then follow the optimal action chunking policy after.
> - $Q^P\_\psi(s, a_{t:t+h_a})$: The expected discounted return if we are at state $s$, execute $a\_{t:t+h_a}$ open-loop for then next $h\_a$-step, optimally choose and execute $a_{t+h_a:t+h} for the rest ($h-h\_a$ actions) open-loop, and then follow the optimal action chunking policy after.
> - $V\_\xi(s)$: The expected discounted return if we are at state $s$, and follow the optimal action chunking policy.
>
> Intuitively, the optimistic distillation loss from $Q_\phi(s, a_{t:t+h})$ to $Q^P_\psi(s, a_{t:t+h_a})$ allows us to implicitly optimize over the second half of the action chunk $a_{t+h_a:t+h}$ for the action chunking policy without actually predicting them using the policy, but $Q^P_\psi(s, a_{t:t+h_a})$ still estimates the value of executing the full open-loop action chunk. Therefore, **there is no objective mismatch between the two critics and does not introduce systematic bias or training instability**.
>
> To further verify this, we also plotted the three values (averaged over training batches in the dataset) for all our tasks over the course of training in Figure 5 and showed that these three values closely track each other (with the gaps between them induced by the optimistic parameter used in the implicit loss function). This shows that our algorithm exhibits effective training with no instability issues.
>
> However, **the discrepancy pointed out by the reviewer does not simply vanish in our alternative view. It gets transformed into a mismatch between the policy that $Q^P_\psi(s, a_{t:t+h_a})$ is estimating and the policy that we rollout during evaluations**: $Q^P_\psi(s, a_{t:t+h_a})$ estimates the value of the action chunk policy where we execute the full action chunk open-loop whereas we only predict the first $h_a$ actions and execute these actions open-loop during evaluations. Essentially, the discrepancy is the result of us closed-loop executing (i.e., only executing the first or first a couple of actions) the action chunking policy. Luckily, as what we will show next, we can still offer theoretical guarantees of the “closed-loop” execution of an action chunking policy either under the original open-loop consistency condition or under a new set of conditions (where a stronger bound can be obtained).

---

> > ### Author Response · Authors · 2025-11-20
> >
> > ## 2. “Theorem 4.6 does not cover decoupled critics. Does the bound still hold?”
> >
> > Not the same bound but **a similar bound still holds**. For this rebuttal, we included a new subsection in our theoretical analysis (Section 4.4) that is dedicated to the `decoupled’ case. More specifically, we analyze the performance of the learned action chunking policy when we always execute the first action in the predicted action chunk (where denote this closed-loop policy as $\pi^\bullet$ and its performance as $V^\bullet$). This is exactly what DQC is approximating (with policy chunk size $h_a=1$).
> >
> > Proposition 4.9 (new!) shows that with the same strong $\varepsilon_h$-open-loop consistency condition, the closed-loop policy $\pi^\bullet$ is also near-optimal:
> >
> > $$V^\star - V^\bullet \leq \frac{3\vartheta_h}{(1-\gamma)^2(1-\gamma^h)}.$$
> >
> > The proof of this result is built on the observation that in order for an action chunking policy to be near-optimal, the first action that takes cannot be too bad. The only caveat is that we need to pay an extra horizon factor $1/(1-\gamma)$.
> >
> > Fortunately, we can obtain a much stronger bound if our data distribution exhibits bounded optimality variability assumptions (defined in Definition 4.10 and the statement in Theorem 4.11) which we will summarize its intuition below briefly. For a distribution of trajectory chunks of size $h$, optimality variability characterizes how much the $h$-step reward ($R_{t:t+h}$) plus the expected optimal value $h$-step ahead ($\gamma^h V^\star(s_{t+h})$) vary within the support of the distribution (or the support of the distribution that is conditioned on some event which we will describe next).
> >
> > Our conditions leverage this optimality variability concept on trajectory chunk distributions conditioned on the first action and the entire action chunk respectively. More specifically, as long as the following two conditions are satisfied, we can form a much stronger bound on the optimality of $\pi^\bullet$. **(1) local optimality variability condition**: the data distribution is a mixture of a bunch of data sources where the optimality variability conditioned on the *current actions* is bounded within each data source, and additionally **(2) global optimality variability condition**: the optimality variability conditioned on the *current action chunks* is bounded globally across the mixture. We expect these optimality variability conditions to hold in various practical settings. For example, it is common to have a dataset consisting of multiple sources where each data source is collected by either human expert or scripted policy that exhibits a somewhat predictable behavior/optimality (e.g., after a robot arm picks up a cube, it will always move up rather than dropping it right away). Such predictable behavior/optimality leads to small local optimality variability, but not necessarily open-loop consistency because the expert/scripted policy can still close-loop react to the disturbance/noise in the environment. It is also worth noting that the second optimality variability condition is much weaker than the first one because it is conditioned on the event where we observe the entire action chunk $a_{t:t+h}$ (rather than the first action $a_t$ only). For example, for data mixture where each pair of data distributions has non-overlapping support on the action chunks, the second condition is trivially implied by the first condition.
> >
> > Let the local optimality variability to be bounded by $\vartheta^L_h$ and the global optimality variability to be bounded by $\vartheta^G_h$, Theorem 4.11 (new!) shows that the closed-loop policy $\pi^\bullet$’s value function admits the following bound (and it is also tight as we show in Appendix F.4):
> >
> > $$V^\star - V^\bullet \leq \frac{\vartheta^L_h}{1-\gamma} + \frac{\vartheta^G_h + \gamma^h\min(\vartheta^L_h,\vartheta^G_h)}{(1-\gamma)(1-\gamma^h)} \leq \frac{\vartheta^L_h}{1-\gamma} + \frac{2\vartheta^G_h}{(1-\gamma)(1-\gamma^h)}.$$
> >
> > This is a significant result because even in the environment when the strong open-loop consistency condition does not hold, we can still guarantee closed-loop execution of the learned action chunking policy to be near optimal as long as the data distribution exhibits the optimality variability conditions described above. Indeed, in our lower-bound analysis in Appendix F.4, we show that there exists examples where the learned action chunking policies are sub-optimal while the closed-loop execution attains near optimal performance.

---

> > > ### Author Response · Authors · 2025-11-20
> > >
> > > (2. Cont.)
> > >
> > > Combining the two results above together, compared to executing the action chunking policy in open-loop chunks, **closed-loop execution attains a similar bound under the strongly $\varepsilon_h$-open-loop consistent assumption, and excels under the new bounded optimality variability assumptions**. Our main results (including the new bounded optimality variability results) can be summarized in the following table (with $H=1/(1-\gamma), \bar H = 1/(1-\gamma^h)$):
> > >
> > > | |**Value Estimation Error**|**AC Optimality (QC)**| **Closed-loop AC Optimality (DQC with $h_a=1$)** |
> > > |---|---|---|---|
> > > |  | $\| \hat{V}\_{\mathrm{ac}} - V\_{\mathrm{ac}} \|$ | $V^\star - V^+_{\mathrm{ac}}$ | $V^\star - V^\bullet$|
> > > | **Weak $\varepsilon_h$-OLC**      | $O(\varepsilon_h H \bar{H})$ | - | - |
> > > | **Strong $\varepsilon_h$-OLC**    | $O(\varepsilon_h H \bar{H})$| $O(\varepsilon_h H \bar{H})$| $O(\varepsilon_h H^2 \bar{H})$                          |
> > > | **$(\vartheta^L_h, \vartheta^G_h)$-BOV** | -  | - | $O(\vartheta^L_h H + \vartheta^G_h H \bar{H})$|
> > >
> > >
> > > ---
> > >
> > > ## 3. “Theorem 4.6 is idealized. Strong open-loop consistency might not hold in practice. How big is $\varepsilon_h$ in OGBench?”
> > >
> > > We show that our improved bound in Theorem 4.6 is tight (by showing that the upper-bound is attainable with our analysis in Appendix F.3). This shows that the bound is not vacuous in nature as there exists examples that can attain the exact value that our bound suggests. Furthermore, our analysis above suggests that if the bounded optimality variability conditions are satisfied the data distribution does not need to be strongly open-loop consistent in order for our policy to be near-optimal. With that being said, there could exist a better characterization for the degree of “open-loop stability” with other metrics (e.g., Wasserstein distance/KL) that matches the reality more closely which would be exciting future work!
> > >
> > > OGBench uses MuJoCo simulation that is deterministic in nature. If we ignore numerical errors and floating point precision errors (OGBench saves states in float32 rather than float64), the open-loop consistency of the tasks that we evaluate on is exactly zero.
> > >
> > > ---
> > >
> > > ## 4. Computational overhead of maintaining two critics
> > >
> > > We have included a new table in the appendix that reports the training speed of our method compared to baselines (Table 3) on a representative task (cube-quadruple). In general, maintaining the additional distilled critic does introduce additional computational overhead, but only about 34% more (DQC: 0.0271 seconds per step vs. QC: 0.0203 seconds per step).

---

> > > > ### Author Response · Authors · 2025-11-20
> > > >
> > > > ## 5. How are hyperparameters tuned?
> > > >
> > > > In our original submission, we tuned $h, n$ over $\{5, 25\}$ and generally found $h=n=25$ to work well for both DQC and NS. For QC, we found $h=5$ to work the best. For $h_a$, we tuned over two values, $\{1, 5\}$ where we found $h_a=1$ to work well on humanoidmaze, puzzle-4x6 and $h_a=5$ to work well on the other tasks (see the complete results for all combinations of $h, n, h_a$ in Appendix A). We used a uniform backup coefficient $\kappa_b=0.9$ for all environments and tuned the distill coefficient $\kappa_d$ over two values (0.5 or 0.8) for DQC.
> > > >
> > > > We have found that the backup coefficient of $0.9$ to be sub-optimal for both our method and our baselines. For this rebuttal, we performed additional hyperparameter tuning on DQC, NS, QC, OS, QC-NS with two seeds over the hyperparameter ranges specified in Table 8. Then, we ran all these methods with the best hyperparameter configurations (Table 7) with 6 different seeds. As a result, DQC’s performance improves substantially (e.g.,  from 24 to 33 on cube-octuple, from 90 to 93 on cube-quadruple, from 72 to 92 on humanoidmaze-giant, from 72 to 81 on puzzle-4x6-1B, see our updated table below). In our original submission, we also had results where we took directly from prior work (with different batch size and number of training steps), and we omitted some entries due to high computational costs. For this rebuttal, we reran all such baselines ourselves with the setting consistent with ours (e.g., 4096 bath size and 1M training steps). Notably, this improved SHARSA’s performance as well (20 => 33 on cube-octuple and 56 => 62 on puzzle-4x6). Overall, our main conclusion remains similar: **(1) Our method, DQC, outperforms SHARSA, the previous state-of-the-art method on 6 of the hardest OGBench GCRL tasks. (2) While sometimes DQC is slightly worse than NS on some tasks, overall it is more robust across the board compared to NS and consistently outperforms QC**. We include a summary of our new result table below (see Table 1 in the updated PDF for the confidence intervals).
> > > >
> > > > To provide a sense of how sensitive our method is with respect to $h$ and $h_a$, we include additional results for all combinations of $h$ and $h_a$ in Table 2.
> > > >
> > > > | Task                   | FBC | HFBC | IQL | HIQL | SHARSA | OS | NS | QC | DQC-naïve | DQC |
> > > > |------------------------|-----|------|-----|------|--------|----|----|----|-----------|-----|
> > > > | cube-triple-100M       | 53  | 57   | 64  | 36   | 82     | 56 | 56 | 17 | 36        | **98** |
> > > > | cube-quadruple-100M    | 32  | 38   | 53  | 24   | 67     | 0  | 22 | 29 | 36        | **93** |
> > > > | cube-octuple-1B        | 0   | 28   | 0   | 18   | **33** | 0  | 7  | 0  | 2         | **31** |
> > > > | humanoidmaze-giant     | 1   | 4    | 4   | 24   | 18     | 0  | **97** | 34 | 81        | 92  |
> > > > | puzzle-4x5             | 0   | 0    | 20  | 0    | 1      | 18 | 88 | 22 | 31        | **96** |
> > > > | puzzle-4x6-1B          | 0   | 5    | 7   | 10   | 62     | 19 | **95** | 43 | 42        | 81  |
> > > >
> > > > (we bold both DQC and SHARSA because SHARSA’s confidence interval overlaps with the DQC’s mean performance)
> > > >
> > > > ---
> > > >
> > > >
> > > > ## 6. How sensitive is the performance of our method with respect to $\kappa_b, \kappa_d$?
> > > >
> > > > As what we have briefly discussed above, we have found a properly tuned $\kappa_b$ to be crucial for the performance of both our method, DQC, and our baselines (e.g., NS). For this rebuttal, we provide a bit more insights on how $\kappa_b$ interact with $\kappa_d$ in our new sensitivity analysis on cube-quadruple (Figure 3, Implicit Parameter sub-plot). The sub-plot shows that while setting either $\kappa_b=0.5$ or $\kappa_d=0.5$ hurts performance by a little (from around 90% to 80-85% success rate), setting both of them to $\kappa_b=\kappa_d=0.5$ leads to the biggest performance drop (all the way to around 65-70% success rate).
> > > >
> > > > ---
> > > >
> > > > Thanks again for your review especially as your review has already resulted in significant improvements to our paper. **Please also let us know if you have any additional questions or concerns regarding our paper!**

---

> ### Comment · Reviewer_WFcb · 2025-11-25
> **Thanks for your rebuttal!**
>
> Thanks for your rebuttal!

---

> > ### Author Response · Authors · 2025-11-26
> >
> > No problem at all! Thanks again for your review and let us know if there are any remaining concerns or questions that we could address!

---

### Official Review · Reviewer_uJSe · 2025-10-30

**Soundness:** 3
**Presentation:** 2
**Contribution:** 2
**Rating:** 4
**Confidence:** 3

**Summary:**

This paper proposes Decoupled Q-Chunking (DQC), an offline RL algorithm that mitigates bootstrapping bias in long-horizon learning by evaluating long action chunks while predicting only short ones. By decoupling the policy and critic chunk lengths, DQC eases policy learning without sacrificing the benefits of multi-step value propagation. Theoretically, it formalizes the open-loop bias in chunked TD backups and proves when chunked critics outperform standard $n$-step returns. Empirically, DQC consistently outperforms prior state-of-the-art methods—including SHARSA—on the most challenging OGBench tasks.

**Strengths:**

1. The core idea of decoupling the policy and critic chunk sizes is novel, effectively addressing a known trade-off in multi-step Q-learning to get "the best of both worlds."
2. The paper provides deep theoretical backing for Q-learning with action chunking, formally identifying and quantifying bias, and proving the conditions under which the approach is superior.
3. It demonstrates superior, state-of-the-art results on challenging long-horizon tasks, significantly outperforming previous methods on the OGBench benchmark.
4. The work includes comprehensive experiments, comparisons to relevant baselines, and ablation studies that strongly support the authors' claims.

**Weaknesses:**

1. The approach still suffers from the inherent bias of open-loop value evaluation in action chunking and lacks a mechanism to actively correct it.
2. Its theoretical guarantees rely on a strong "open-loop consistency" assumption for the offline dataset, which may not hold in many real-world scenarios, limiting the generality of the claims.
3. The use of a fixed, global chunk size for both the policy and critic is a limitation, as the optimal action horizon might vary depending on the state.
4. The framework introduces additional components (e.g., two Q-networks, best-of-N sampling) and hyperparameters, increasing computational overhead and the practical difficulty of tuning and deployment.

**Questions:**

in weaknesses

---

> ### Author Response · Authors · 2025-11-20
>
> Thanks for your review and your review has already resulted in various improvements to our paper! We are glad to hear that you find our method novel, our experiments comprehensive and demonstrate superior SOTA results on challenging long-horizon tasks, and you find our theoretical insight deep in formally identifying and quantifying biases in Q-learning with action chunking.
>
> In this rebuttal, we **(1) addressed your concern on computational overhead by showing that our method only required ~34% more training time, while significantly improving upon our baselines, (2) introduced a new theory result that shows how our method does not necessarily suffer from the open-loop consistency problem suffered by Q-chunking, and provided theoretical justifications to quantify under what conditions this holds.**
>
> ---
>
> ## 1. Computation overhead of two Q-networks and best-of-N sampling
>
> We have included a new table in the appendix that reports the training speed of our method compared to baselines (Table 3) on a representative task (cube-quadruple). In general, maintaining the additional distilled critic does introduce additional computational overhead, but only about 34% more (DQC: 0.0271 seconds per step vs. QC: 0.0203 seconds per step).
>
> In addition, we would like to highlight that our best-of-N sampling is not used during training and is only used during evaluations. This allows the sampling procedure to be effectively parallelized (in contrast to using best-of-N sampling during training where it would effectively explode the batch size by a factor of $N$). This technique is not unique to our method–the previous state-of-the-art method on these OGBench hard domains, (i.e., SHARSA), also employed the same best-of-N sampling during evaluations.

---

> > ### Author Response · Authors · 2025-11-20
> >
> > ## 2. Open-loop consistency assumption may not hold in many real-world scenarios, limiting the generality of the claims.
> >
> > For this rebuttal, **we developed new theoretical results that provide theoretical guarantees that do not rely on the strong open-loop consistency assumption**. Furthermore, our theoretical results suggest that **the performance of our policy does not necessarily suffer from the inherent bias of open-loop value evaluation in action chunking**.
> >
> > In particular, we can obtain a much stronger bound for the performance of the closed-loop execution of the learned action chunking policy (always execute the first action in the predicted action chunk) leveraging a new bounded optimality variability conditions (defined in Definition 4.10 and the statement in Theorem 4.11) which we will summarize its intuition below briefly. For a distribution of trajectory chunks of size $h$, optimality variability characterizes how much the $h$-step reward ($R_{t:t+h}$) plus the expected optimal value $h$-step ahead ($\gamma^h V^\star(s_{t+h})$) vary within the support of the distribution (or the support of the distribution that is conditioned on some event which we will describe next).
> >
> > Our conditions leverage this optimality variability concept on trajectory chunk distributions conditioned on the first action and the entire action chunk respectively. More specifically, as long as the following two conditions are satisfied, we can form a much stronger bound on the optimality of $\pi^\bullet$. **(1) local optimality variability condition**: the data distribution is a mixture of a bunch of data sources where the optimality variability conditioned on the *current actions* is bounded within each data source, and additionally **(2) global optimality variability condition**: the optimality variability conditioned on the *current action chunks* is bounded globally across the mixture. We expect these optimality variability conditions to hold in various practical settings. For example, it is common to have a dataset consisting of multiple sources where each data source is collected by either human expert or scripted policy that exhibits a somewhat predictable behavior/optimality (e.g., after a robot arm picks up a cube, it will always move up rather than dropping it right away). Such predictable behavior/optimality leads to small local optimality variability, but not necessarily open-loop consistency because the expert/scripted policy can still close-loop react to the disturbance/noise in the environment. It is also worth noting that the second optimality variability condition is much weaker than the first one because it is conditioned on the event where we observe the entire action chunk $a_{t:t+h}$ (rather than the first action $a_t$ only). For example, for data mixture where each pair of data distributions has non-overlapping support on the action chunks, the second condition is trivially implied by the first condition.
> >
> >
> > Let the local optimality variability to be bounded by $\vartheta^L_h$ and the global optimality variability to be bounded by $\vartheta^G_h$, we show that the closed-loop policy $\pi^\bullet$’s value function admits the following bound (and it is also tight as we show in Appendix F.4):
> >
> > $$V^\star - V^\bullet \leq \frac{\vartheta^L_h}{1-\gamma} + \frac{\vartheta^G_h + \gamma^h\min(\vartheta^L_h,\vartheta^G_h)}{(1-\gamma)(1-\gamma^h)} \leq \frac{\vartheta^L_h}{1-\gamma} + \frac{2\vartheta^G_h}{(1-\gamma)(1-\gamma^h)}.$$
> >
> > This is a significant result because even in the environment when the strong open-loop consistency condition does not hold, we can still guarantee closed-loop execution of the learned action chunking policy to be near optimal as long as the data distribution exhibits the optimality variability conditions described above. Indeed, in our lower-bound analysis in Appendix F.4, we show that there exists examples where the learned action chunking policies are sub-optimal while the closed-loop execution attains near optimal performance.
> >
> > ---
> >
> > ## 3. “The use of a fixed, global chunk size for both the policy and critic is a limitation, as the optimal action horizon might vary depending on the state.”
> >
> > A fixed chunk size/backup horizon is a common design choice adopted by prior work [1, 2] because how to best dynamically pick the horizon is still an open problem. We do believe that this is a promising direction for future work and we are looking forward to future work that explores this direction!
> >
> > [1] "Reinforcement learning with action chunking." arXiv preprint arXiv:2507.07969 (2025).
> >
> > [2] "Horizon Reduction Makes RL Scalable." arXiv preprint arXiv:2506.04168 (2025).
> >
> > ---
> >
> > Thanks again for your review as especially your review has already resulted in various improvements to our paper. **If we have successfully addressed all your concerns, could you kindly update the score?** Please also let us know if you have any additional questions or concerns regarding our paper!

---

> > > ### Author Response · Authors · 2025-11-28
> > > **A quick followup**
> > >
> > > As the end of the rebuttal discussion period is approaching quickly, we would like to post a short follow-up to check whether our rebuttals have addressed all your concerns. If there are any remaining questions or concerns, please let us know!

---

### Official Review · Reviewer_WP1s · 2025-11-01

**Soundness:** 4
**Presentation:** 3
**Contribution:** 3
**Rating:** 8
**Confidence:** 3

**Summary:**

This work improves on the concept of chunked critics in the context of off-policy RL.
The chunk length of the critic and policy is decoupled by optimizing the policy against a distilled critic, allowing the policy to output shorter chunks.

**Strengths:**

- Well-written and detailed theoretical investigation.
- Strong and robust results.

**Weaknesses:**

- Some minor grammatical errors (e.g., l. 26, 182)
- There is no discussion of the computation overhead compared to the baseline methods (e.g., from maintaining the additional distilled critic). Since the performance gap to the baselines is substantial, providing a short intuition should suffice.

**Questions:**

- Notably, the only experiment where the proposed method is outperformed is by the $NS$ approach in the puzzle-4x6-1B environment. Can you provide an explanation/intuition of why this is the case? Are there certain environmental characteristics where you would expect your approach to perform worse than computationally simpler methods?

---

> ### Author Response · Authors · 2025-11-20
>
> Thank you for taking the time to review our paper and we appreciate your positive review! We addressed your concerns by providing a discussion of **(1) the computational overhead compared to baselines and (2) why the n-step return baseline performs well in the puzzle-4x6-1B environment.**
>
> **Computational overhead**
>
> We included a new table (Appendix, Table 3) that reports the training speed of our method compared to baselines on a representative task (cube-quadruple). In general, maintaining the additional distilled critic does introduce additional computational overhead, but only about 34% more (DQC: 0.0271 seconds per step vs. QC: 0.0203 seconds per step).
>
> **Why is NS so good on puzzle-4x6-1B?**
>
> This is a great question! We do not have a definitive answer but our hyperparameter tuning process on this task gave some clues. When we tuned hyperparameters for puzzle-4x6, we found that introducing optimism often hurts the performance ($\kappa\_b > 0.5, \kappa\_d > 0.5$) so we used small optimism $\kappa\_b=0.7, \kappa\_d=0.5$ in the end. The same goes for our NS baseline where we simply used $\kappa\_b=0.5$. This observation made us believe that puzzle-4x6 might exhibit a very narrow data distribution, and when combined with the long-horizon nature of the task, any optimism may cause the extracted policy to go out of distribution much more easily compared to other tasks.
>
> **Grammar:** Thanks for catching these mistakes! They have been fixed in the updated PDF.
>
> Thank you again for your review and help in improving our paper. Please let us know if you have any additional questions or concerns regarding our paper!

---

### Author Response · Authors · 2025-11-20
**General Comments**

We would like to thank all reviewers for the detailed and insightful reviews and area chair for facilitating the review discussion. For the updated version of PDF, we have made various improvements to our empirical evaluations and theoretical analysis which we summarize below. All changes in the PDF are highlighted in purple.

## 1. Empirical evaluation improvements with additional tuning and seeds (3 seeds => **6 seeds**):

In our original submission, we used a uniform $\kappa_b=0.9$ for all environments, which turned out to be sub-optimal for both our method and our baselines. For this rebuttal, we performed additional hyperparameter tuning on DQC, NS, QC, OS, QC-NS with two seeds over the hyperparameter ranges specified in Table 8. Then, we ran all these methods with the best hyperparameter configurations (Table 7) with 6 different seeds. As a result, DQC’s performance improves substantially. In our original submission, we also had results where we took directly from prior work (with different batch size and number of training steps). For this rebuttal, we reran all such baselines ourselves with the setting consistent with ours (e.g., 4096 bath size and 1M training steps). Overall, our main conclusion remains similar: **(1) Our method, DQC, outperforms SHARSA, the previous state-of-the-art method on 6 of the hardest OGBench GCRL tasks. (2) While sometimes DQC is slightly worse than NS on some tasks, overall it is more robust across the board compared to NS and consistently outperforms QC**. We include a summary of our new result table below (see Table 1 in the updated PDF for the confidence intervals). On top of these new results, we also included a more comprehensive sensitivity analysis in Figure 3.

| Task | FBC | HFBC | IQL | HIQL | SHARSA | OS | NS | QC | DQC-naïve | DQC |
|--|-|-|-|-|-|-|-|-|-|-|
| cube-triple-100M | 53  | 57   | 64  | 36   | 82 | 56 | 56 | 17 | 36 | **98** |
| cube-quadruple-100M    | 32  | 38   | 53  | 24   | 67     | 0  | 22 | 29 | 36 | **93** |
| cube-octuple-1B  | 0   | 28   | 0   | 18   | **33** | 0  | 7  | 0  | 2 | **31** |
| humanoidmaze-giant | 1   | 4    | 4   | 24   | 18  | 0  | **97** | 34 | 81 | 92  |
| puzzle-4x5   | 0   | 0    | 20  | 0    | 1      | 18 | 88 | 22 | 31 | **96** |
| puzzle-4x6-1B  | 0   | 5    | 7   | 10   | 62  | 19 | **95** | 43 | 42 | 81  |

(we bold both DQC and SHARSA because SHARSA’s confidence interval overlaps with the DQC’s mean performance)

## 2. New theoretical results to provide guarantees for our method, DQC, in Section 4.4.

We developed two new results regarding our DQC method. (1) Under the strong $\varepsilon_h$-open-loop consistency condition, we showed that closed-loop execution of the action chunking policy is near-optimal (Proposition 4.9). (2) Under our new optimality variability conditions with $\vartheta^L_h$ and $\vartheta^G_h$, the closed-loop execution of the action chunking policy is also guaranteed to be near-optimal, independent from the degree of open-loop consistency of the data distribution (Theorem 4.11).

Let $H=1/(1-\gamma), \bar H = 1/(1-\gamma^h)$, our current theoretical results can be summarized as follows:

| |**Value Estimation Error**|**AC Optimality (QC)**| **Closed-loop AC Optimality (DQC with $h_a=1$)** |
|---|---|---|---|
|  | $\| \hat{V}\_{\mathrm{ac}} - V\_{\mathrm{ac}} \|$ | $V^\star - V^+_{\mathrm{ac}}$ | $V^\star - V^\bullet$|
| **Weak $\varepsilon_h$-OLC**      | $O(\varepsilon_h H \bar{H})$ | - | - |
| **Strong $\varepsilon_h$-OLC**    | $O(\varepsilon_h H \bar{H})$| $O(\varepsilon_h H \bar{H})$| $O(\varepsilon_h H^2 \bar{H})$                          |
| **$(\vartheta^L_h, \vartheta^G_h)$-BOV** | -  | - | $O(\vartheta^L_h H + \vartheta^G_h H \bar{H})$|

(OLC: open-loop consistent, BOV: bounded optimality variability)

## 3. Additional baseline: DQC-naïve.

Since the core idea of our method is to decouple the chunk size of the policy from that of the critic, we also explore another common sense baseline where we take QC and simply execute the first couple of actions in the action chunk instead of the full chunk. We include this baseline as part of our main updated result table and plot (Table 1 and Figure 1). While this naïve way of decoupling alone can already provide some benefits over QC, they are generally much worse than DQC. This shows that training a separate distilled critic and a policy with smaller action chunk is crucial for performance improvements in DQC.

## 4. Tight bounds for all our theoretical results in Section 4.

For Theorem 4.4, Corollary 4.5, Theorem 4.6, we improved the bounds such that they match our lower-bounds (available in Appendix F).

**We again thank all reviewers and AC and please let us know if there are any additional questions or concerns regarding our paper.**

---

### Meta-Review · Area_Chair_BSGg · 2026-01-06

**Summary:**

This paper proposes Decoupled Q-Chunking (DQC), improving chunked-critic off-policy/offline RL by decoupling critic chunk length from policy chunk length: the policy outputs shorter chunks while optimizing against a distilled critic that preserves long-horizon value propagation. Reviewers generally praised the strong theory and large gains on difficult long-horizon OGBench tasks, but the main decision-relevant concerns were (i) practicality/overhead of maintaining an additional distilled critic and extra components, (ii) whether the theory relies too heavily on idealized open-loop consistency assumptions and how it extends to the decoupled setting, and (iii) whether the empirical gains are truly from decoupling vs simpler variants like QC-NS, plus questions about hyperparameter sensitivity (batch size, expectiles) and the one environment where a simpler baseline wins.

**Reviewer Concerns:**

The rebuttal addressed most key concerns: authors added measured compute overhead (about 34% slower per step than QC on a representative task), explained the one failure case (puzzle-4x6-1B where optimism hurts due to narrow support + long horizon), and provided additional theory for the decoupled/closed-loop execution case (new results beyond the strong open-loop consistency assumption using “optimality variability” conditions). They also clarified that best-of-N is evaluation-only, added plots showing distilled/long-horizon critics track each other, expanded hyperparameter tuning + sensitivity (including expectile/quantile variants and batch size ablations), and strengthened empirical comparisons showing QC-NS is more unstable with more seeds. Remaining limitations are mostly scope-related (no D4RL; fixed global chunk sizes), but these were framed as out of scope/future work with justification.

**Reviewer Scores:**

WP1s likely stays 8→8 (concerns fully addressed). uJSe likely 4→6 (new theory relaxing open-loop consistency + overhead table and clarifications address core weaknesses). WFcb likely 6→7 (decoupled-theory and tracking analysis plus overhead/tuning details directly answer questions). c39B likely 6→7 (QC-NS instability evidence, batch-size ablation, and sensitivity studies address the main critique). Overall, discussion trends upward.

---

### Decision · Program_Chairs · 2026-01-26

Accept (Poster)